# Time changes during the last 40 years in the Solfatara magmatic-hydrothermal system (Campi Flegrei, Italy): new conceptual model and future scenarios

Luigi Marini[1], Claudia Principe[2,3], Matteo Lelli[2,4]

[1] STEAM srl, Pisa, I-56121, Italy
[2] CNR, Istituto di Geoscienze e Georisorse, Pisa, I-56124, Italy
[3] INGV - Istituto Nazionale di Geofisica e Vulcanologia, Osservatorio Vesuviano, Napoli, I-80124, Italy
[4] INGV - Istituto Nazionale di Geofisica e Vulcanologia, Sezione di Pisa, Pisa, I-56125, Italy

*Correspondence to*: Matteo Lelli (m.lelli@igg.cnr.it)

**Abstract.** We propose a new conceptual model of the Solfatara magmatic-hydrothermal system based on the results of new gas-geoindicators (Marini et al., 2022) and the available geological, volcanological, and geophysical information from surface surveys and deep geothermal wells. Using the new gas-geoindicators, we monitored the temperature and total fluid pressure over a time interval of ~40 years: (i) in the shallow reservoir (0.25-0.45 km depth), where CO equilibrates; (ii) in the intermediate reservoir (2.7-4.0 km depth), where $CH_4$ attains equilibrium; (iii) in the deep reservoir (6.5-7.5 km depth), where $H_2S$ achieves equilibrium. From 1983 to 2022, the temperature and total fluid pressure of the shallow reservoir did not depart significantly from ~220°C and ~25 bar, whereas remarkable, progressive increments in temperature and total fluid pressure occurred in the intermediate and deep reservoirs, with peak values of 590-620 °C and 1200-1400 bar in the intermediate reservoir and 1010-1040°C and 3000-3200 bar in the deep reservoir, in 2020. Our new conceptual model allowed us to explain: (a) the pressurization-depressurization of the intermediate reservoir, acting as the "engine" of bradyseism, and (b) the time changes of total fluid pressure in the deep reservoir, working as temporary "on-off switch" of magmatic degassing. We also used our new conceptual model to infer the only two possible future scenarios, in the lack of external factors, showing that the pressurization of the intermediate reservoir might trigger a hydrothermal explosion and proposing risk mitigation actions.

## 1 Introduction

The Campi Flegrei, Phlegraean Fields in English, are located next to Naples (Fig. 1a), are a very densely populated area with about 500,000 inhabitants, and are considered to be one of the most dangerous volcanic sites worldwide, as they were impacted by several large-scale explosive eruptions. Slow vertical ground movements known as "bradyseism" have affected the Campi Flegrei area since at least Roman times with alternating episodes of uplift or resurgence and deflation or subsidence (Lyell, 1830). The slow ground movements typical of the bradyseism are totally different from (and should not be confused with) the

fast and local uplift preceding the last volcanic eruption in the Campi Flegrei area that began on 29 September 1538, had the

duration of a week, and consisted in a small phreatomagmatic event generating the Monte Nuovo cone (Guidoboni and

Ciuccarelli, 2011). The last resurgence cycle begun in 1950, caused maximum uplifts of 1.77 m in 1969-1972 and 1.79 m in

1982-1984, both followed by deflation periods, the last of which ended in 2005, when the still ongoing inflation phase started

(Del Gaudio et al., 2010; De Martino et al., 2014; Tramelli et al., 2021). Ground deformation was accompanied by seismicity

(e.g., D'Auria et al., 2011; Di Luccio et al., 2015) and changes in the chemistry and emission rate of the fumarolic fluids

released from Solfatara-Pisciarelli (Cioni et al., 1984, 1989; Chiodini and Marini, 1998; Caliro et al., 2007, 2014; Chiodini,

2009; Chiodini et al., 2010, 2011, 2012, 2015, 2016, 2017a, b, 2021; Buono et al., 2023; Fig. 1b), the most impressive

manifestations in the Campi Flegrei.

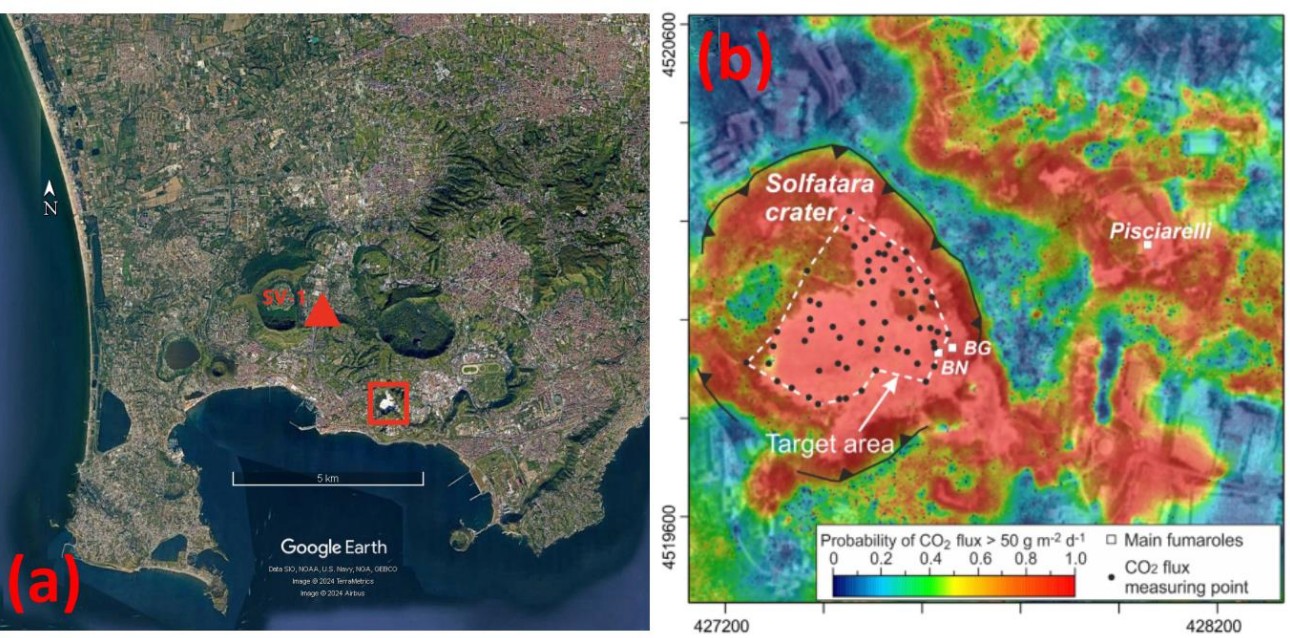

**Figure 1: (a) © Google-Earth image of Campi Flegrei, showing the location of the deep geothermal well San Vito 1 (SV-1). The red square indicates Fig. 1b. (b) Map of the Solfatara-Pisciarelli diffuse degassing structure elaborated from the 1998-2016 dataset of $CO_2$ fluxes also showing the location of the Bocca Grande (BG), Bocca Nuova (BN), and Pisciarelli (Pi) fumarolic vents (from Chiodini et al., 2021; Elsevier license number 5601960223358 on Aug 04, 2023). Kilometric coordinates are reported on the two axes.**

The Solfatara edifice has a sub-circular crater, with a diameter varying between 610 and 710 m and an area of ~0.35 km$^2$ (Isaia

et al., 2015). It was generated by a purely hydrothermal event (Principe, 2024) which possibly occurred about 4200-4400 ka

BP (Isaia et al., 2009). The morphology of the Solfatara edifice and crater as well as the present upflow of deep fluids are

mainly controlled by Apenninic and anti-Apenninic tectonic elements, striking WNW and ENE, respectively (Rosi and Sbrana,

1987).

During the last 40 years, the Solfatara fumarolic fluids were periodically collected and analyzed, in the framework of the

Campi Flegrei volcanic surveillance (Chiodini et al., 2021 and references therein; Buono et al., 2023). Fluids samples were

mainly obtained from the Solfatara vents known as Bocca Grande and Bocca Nuova, discharging $CO_2$-rich superheated steam, with smaller amounts of $H_2S$, $N_2$, $H_2$, $CH_4$, He, CO, and Ar, in decreasing order, at outlet temperatures of 150-165°C and 135-154°C, respectively[1]. Thus, a very large geochemical database, one of the longest record series worldwide, was produced. It comprises the chemical concentrations of several gas species in fumarolic fluids (Table S1) and a congruous number of isotopic data (Cioni et al., 1984, 1989; Chiodini and Marini, 1998; Caliro et al., 2007, 2014; Chiodini, 2009; Chiodini et al., 2010, 2011, 2012, 2015, 2016, 2017a, b, 2021; Buono et al., 2023).

A fundamental tool to understand the behaviour of any hydrothermal-magmatic system is the conceptual model. A general conceptual model of volcano-hosted magmatic-hydrothermal systems was proposed by Fournier (1999), whereas Cumming (2009, 2016) provided the guidelines for elaborating the conceptual model of the hydrothermal domain of these systems. In the case of the Solfatara magmatic-hydrothermal system, the conceptual models proposed so far extend to depths of a few hundred meters (Cioni et al., 1984) or 2.5-3.0 km (Caliro et al., 2007) and refer to the hydrothermal domain. In fact, the gas equilibration temperatures and pressures obtained in previous studies (Cioni et al., 1984, 1989; Chiodini and Marini, 1998; Caliro et al., 2007, 2014; Chiodini, 2009; Chiodini et al., 2010, 2011, 2012, 2015, 2016, 2017a, b, 2021; Buono et al., 2023) were exclusively or chiefly based on geothermometers and geobarometers controlled entirely or mostly by CO, which equilibrates at shallow depth. Among the other gas species, $CH_4$ was considered together with CO in some cases, assuming it has little effects on the obtained results (see section 4.3), or treated as a tracer rather than an indicator, whereas $H_2S$ was considered a gas species of little interest based on the behaviour of the Giggenbach's gas-geothermometer involving pyrite (Caliro et al., 2007).

Recently, we proposed new gas geothermometers and geobarometers, which were suitably calibrated for different plausible expansion paths of the Solfatara fluids, also considering the deviations from the ideal gas behaviour, for the first time. Our results were presented and thoroughly discussed by Marini et al. (2022). In this work, we summarize the main findings of Marini et al. (2022), taking into account the data produced by Buono et al. (2023) for Bocca Grande and Bocca Nuova fumaroles for October 2020 to January 2022. Moreover, in this work, we use our geothermometric and geobarometric results, as well as the information from other disciplines (e.g., surface geo-volcanological surveys, data from geothermal deep wells, and geophysical investigations) to elaborate a new conceptual model of the Solfatara magmatic-hydrothermal system which extends at magmatic depths ($\geq$ 8 km) and represents a step forward with respect to previous conceptual models. The new conceptual model proposed in this work is also able to explain the slow vertical ground movements and other active processes typical of the bradyseism. In its framework, and in the lack of external factors, two possible future scenarios were inferred, showing that the pressurization of the intermediate reservoir might trigger a hydrothermal explosion and proposing risk mitigation actions.

---

[1] In the past, samples were collected sporadically at Pisciarelli while, in the last years, samples were periodically obtained also from these vents due to their increasing flow and temperature with time.

The main reason that pushed us to write this paper is to provide a contribution to the discussion animating, in this period, the international scientific community on the possible evolution of the unrest episode currently affecting the Campi Flegrei. Furthermore, the mitigation of the volcanic hazard in the Campi Flegrei is not a local issue because worldwide volcanologists look at it as an analogue of similar volcanic systems (Marini et al., 2022).

**2 Methods**

Three distinct equilibrium temperatures and related total fluid pressures were computed for each fumarolic gas sample collected and analysed from June 1983 to January 2022. The first one refers to CO equilibrium, which is controlled by the homogeneous reaction:

$$CO_2 + H_2 = CO + H_2O. \tag{1}$$

The second one relates to $CH_4$ equilibrium, which is governed by the homogeneous reaction:

$$CO_2 + 4 H_2 = CH_4 + 2 H_2O. \tag{2}$$

The third one refers to $H_2S$ equilibrium, which is ruled by the heterogeneous reaction:

$$CaSO_{4(s)} + CO_2 + 4 H_2 = CaCO_{3(s)} + H_2S + 3 H_2O, \tag{3}$$

involving anhydrite $[CaSO_{4(s)}]$ and calcite $[CaCO_{3(s)}]$. Of interest is also the heterogeneous redox reaction:

$$CaSO_{4(s)} + CH_4 = CaCO_{3(s)} + H_2S + H_2O. \tag{4}$$

which is assumed to fix the $CH_4$ concentration in the zone of $H_2S$ equilibration, $X_{CH_4 @ T_{H_2S}}$. The equilibrium temperatures of CO, $CH_4$, and $H_2S$ as well as the $X_{CH_4 @ T_{H_2S}}$ were computed by means of the gas-geothermometers of Marini et al. (2022). These were obtained by rearranging the log K of reactions (1), (2), (3), and (4), in order to separate the analytical data from all the other terms (log K, $P_{H_2O}$, and fugacity coefficients, $\phi_i's$), which were expressed as a function of temperature (range

specified below) and $CO_2$ mole fractions ($X_{CO_2}$ range 0.05 to 0.40). The gas- geothermometers are as follows:

$$log\left(\frac{X_{CO}}{X_{CO_2}}\right) - log\left(\frac{X_{H_2}}{X_{H_2O}}\right) = \alpha = log\,K_{(1)} - log\frac{\phi_{CO}}{\phi_{CO_2}} + log\frac{\phi_{H_2}}{\phi_{H_2O}} \tag{5}$$

$$log\left(\frac{X_{CH_4}}{X_{CO_2}}\right) - 4 \cdot log\left(\frac{X_{H_2}}{X_{H_2O}}\right) = \beta = log\,K_{(2)} - log\frac{\phi_{CH_4}}{\phi_{CO_2}} + 4 \cdot log\frac{\phi_{H_2}}{\phi_{H_2O}} + 2 \cdot log\,P_{H_2O} + 2 \cdot log\,\phi_{H_2O} \tag{6}$$

$$log\left(\frac{X_{H_2S}}{X_{CO_2}}\right) - 4 \cdot log\left(\frac{X_{H_2}}{X_{H_2O}}\right) = \gamma = log\,K_{(3)} - log\left(\frac{\phi_{H_2S}}{\phi_{CO_2}}\right) + 3 \cdot log\left(\frac{\phi_{H_2}}{\phi_{H_2O}}\right) + log\,\phi_{H_2} + log\,P_{H_2O} \tag{7}$$

$$log\,X_{CH_4} - log\,X_{H_2S} = -log\,K_{(4)} + log\left(\frac{\phi_{H_2S}}{\phi_{CH_4}}\right) + log\,\phi_{H_2O} + log\,P_{H_2O} \tag{8}$$

As reaction (2), occurring in the intermediate reservoir (see below), consumes $CO_2$ and $H_2$, producing $CH_4$ and $H_2O$, the analytical mole fractions of $H_2$, $CO_2$, and $H_2O$ were corrected, based on the stoichiometry of reaction (2), and the corrected concentrations $X_{H_2,c} = X_{H_2} + 4 \cdot X_{CH_4}$, $X_{CO_2,c} = X_{CO_2} + X_{CH_4}$, and $X_{H_2O,c} = X_{H_2O} - 2 \cdot X_{CH_4}$ were considered in Eqn. (7).

For what concerns the fugacity coefficients, first, the $\phi_{H_2O}$ and $\phi_{CO_2}$ were computed using the GERG-2008 EOS (Kunz and Wagner, 2012), the EOS of Gallagher et al. (1993), and the Peng-Robinson EOS, obtaining comparable values. Second, the

Peng-Robinson EOS was utilized to compute the fugacity coefficients not only of $H_2O$ and $CO_2$, but also of $H_2S$, $H_2$, $CH_4$, and $CO$, for Solfatara gas mixtures of different $X_{CO_2}$ (which was set equal to 0.05, 0.10, 0.20, 0.30, and 0.40), $X_{H_2O}$ (which is equal to $1 - \sum X_i$), and average mole fractions of other gas species. This was found to be an adequate approximation, not surprisingly, because $X_{H_2O}$ and $X_{CO_2}$ constitute together 99.3 to 100 mol % of the Solfatara fluids. Further considerations on fugacity coefficients are found in Appendix A.

Different expansion paths were considered in the calibration of gas geothermometers by Marini et al. (2022), namely: (i) the saturation expansion path involving a vapor phase and a brine containing 33.5 wt% NaCl, which was utilized to calibrate the $CO$-, $CH_4$-, and $H_2S$ geothermometers; (ii) the saturation expansion path comprising a vapor phase and a brine containing 21 wt% NaCl, which was used to calibrate the $CO$- and $CH_4$-geothermometers;  (iii) the linear P-T depressurization path, which was adopted to calibrate the $CO$- and $CH_4$-geothermometers; (iv) the isenthalpic decompression path, which was used to

calibrate the $CO$-geothermometer only. Consequently, for the dataset of interest, there are: (i) four time series of $CO$ equilibrium temperature and total fluid pressure, related to the isenthalpic, linear P-T, saturation (21 wt% NaCl), and saturation (33.5 wt% NaCl) decompression paths; (ii) three time series of $CH_4$ equilibrium temperature and total fluid pressure, related to the linear P-T, saturation (21 wt% NaCl), and saturation (33.5 wt% NaCl) decompression paths; (iii) one time series of $H_2S$ equilibrium temperature and total fluid pressure, related to the saturation (33.5 wt% NaCl) decompression path. The $CO$

equilibrium temperatures and total fluid pressures related to the four distinct decompression paths are not very different from each other with deviations of few °C and few bar, respectively, and do not deserve further considerations. In contrast, it is worth to examine the differences between the $CH_4$ equilibrium temperatures and total fluid pressures calculated for the three distinct decompression paths and to explain why we choose the outcomes of the saturation (21 wt% NaCl) decompression path among the three time-series given by the $CH_4$-geoindicators. These matters are discussed in Appendix B. We recall that, in

this communication, we present and discuss: (i) The $CO$ and $CH_4$ equilibrium temperatures and total fluid pressures computed for the saturation decompression path of Solfatara fluids involving a vapor phase and a brine containing 21 wt% NaCl, and (ii) the $H_2S$ equilibrium temperature and total fluid pressures, as well as the $CH_4$ concentration in the $H_2S$ equilibration zone calculated for the saturation decompression path of Solfatara fluids involving a vapor phase and a brine containing 33.5 wt% NaCl. We also recall that gas species are assumed to attain chemical equilibrium in an almost pure saturated vapor phase coexisting with a very small amount of brine, while other assumptions and limitations related to gas geothermometers are

given in Appendix C. Brine-vapor coexistence is the most probable condition in the Solfatara hydrothermal-magmatic system, because the other possible conditions are highly unlikely. In fact: (i) On the one hand, the occurrence of a single liquid (brine) at comparatively low temperatures and high pressures is at variance with the huge amount of heat released from the magma batch and transferred to the overlying hydrothermal part of the system. (ii) On the other hand, a single vapor phase coexisting

with solid NaCl might occur in depressurized vapor-cored magmatic systems (Reyes et al., 1993), such as Vulcano Island, Italy (Cioni and D'Amore, 1984), and many systems of Indonesia (Abiyudo et al., 2016) and The Philippines (Reyes et al., 1993; Ramos-Candelaria et al., 1995; Apuada and Sigurjonsson, 2008), but it is at variance with the current pressurization and

related ground uplift of the Solfatara hydrothermal-magmatic system. Brine-vapor coexistence is assumed to fix $H_2O$ partial pressure. This is the only role played by the brine, both in our approach and in Giggenbach (1987).

The CO equilibrium temperature is computed using the following relation:

$$T_{CO}(°C) = [0.0008487619 \cdot X_{CO_2}^4 - 0.001191429 \cdot X_{CO_2}^3 + 0.0006858667 \cdot X_{CO_2}^2 - 0.0003799214 \cdot X_{CO_2} + 0.001147017 +$$

$$(-0.005137695 \cdot X_{CO_2}^4 + 0.005929029 \cdot X_{CO_2}^3 - 0.002740698 \cdot X_{CO_2}^2 + 0.0005322879 \cdot X_{CO_2} - 0.00077217) \cdot \alpha +$$

$$(-0.009782143 \cdot X_{CO_2}^4 + 0.01168796 \cdot X_{CO_2}^3 - 0.005585195 \cdot X_{CO_2}^2 + 0.001486423 \cdot X_{CO_2} - 0.0005542923) \cdot \alpha^2 +$$

$$(-0.005205648 \cdot X_{CO_2}^4 + 0.006267498 \cdot X_{CO_2}^3 - 0.003011277 \cdot X_{CO_2}^2 + 0.0008427649 \cdot X_{CO_2} - 0.0002680103) \cdot \alpha^3 +$$

$$(-0.0008588714 \cdot X_{CO_2}^4 + 0.001036726 \cdot X_{CO_2}^3 - 0.000498879 \cdot X_{CO_2}^2 + 0.0001418067 \cdot X_{CO_2} - 0.00004093) \cdot \alpha^4]^{-1} -$$

273.15                    (9)

The $CH_4$ equilibrium temperature is calculated utilizing one of the two following equations:

$$T_{CH_4}(°C) = [-249.7937 \cdot X_{CO_2}^4 + 260.668 \cdot X_{CO_2}^3 - 98.31472 \cdot X_{CO_2}^2 + 16.41854 \cdot X_{CO_2} - 1.147862 + (89.79142 \cdot X_{CO_2}^4 -$$

$$93.69606 \cdot X_{CO_2}^3 + 35.33578 \cdot X_{CO_2}^2 - 5.899114 \cdot X_{CO_2} + 0.4110321) \cdot \beta + (-12.07122 \cdot X_{CO_2}^4 + 12.59559 \cdot X_{CO_2}^3 - 4.749825 \cdot$$

$$X_{CO_2}^2 + 0.7927105 \cdot X_{CO_2} - 0.0550276) \cdot \beta^2 + (0.7193242 \cdot X_{CO_2}^4 - 0.7505385 \cdot X_{CO_2}^3 + 0.2830093 \cdot X_{CO_2}^2 - 0.04721837 \cdot X_{CO_2} +$$

$$0.003268372) \cdot \beta^3 + (-0.01603161 \cdot X_{CO_2}^4 + 0.01672658 \cdot X_{CO_2}^3 - 0.006306754 \cdot X_{CO_2}^2 + 0.001051957 \cdot X_{CO_2} -$$

$$0.00007262834) \cdot \beta^4]^{-1} - 273.15$$                    (10)

$$T_{CH_4}(°C) = [-338.1988 \cdot X_{CO_2}^4 + 324.4917 \cdot X_{CO_2}^3 - 102.6725 \cdot X_{CO_2}^2 + 10.85846 \cdot X_{CO_2} - 0.03389924 + (164.3052 \cdot X_{CO_2}^4 -$$

$$157.9805 \cdot X_{CO_2}^3 + 50.16642 \cdot X_{CO_2}^2 - 5.346373 \cdot X_{CO_2} + 0.02051445) \cdot \beta + (-29.85781 \cdot X_{CO_2}^4 + 28.76961 \cdot X_{CO_2}^3 - 9.16867 \cdot$$

$$X_{CO_2}^2 + 0.9845943 \cdot X_{CO_2} - 0.004369622) \cdot \beta^2 + (2.405338 \cdot X_{CO_2}^4 - 2.322631 \cdot X_{CO_2}^3 + 0.7428868 \cdot X_{CO_2}^2 - 0.08038292 \cdot X_{CO_2} +$$

$$0.0004033068) \cdot \beta^3 + (-0.07247987 \cdot X_{CO_2}^4 + 0.07013834 \cdot X_{CO_2}^3 - 0.02251543 \cdot X_{CO_2}^2 + 0.002454832 \cdot X_{CO_2} -$$

$$0.00001364184) \cdot \beta^4]^{-1} - 273.15$$                    (11)

Equation (10) is valid for $\beta > 9.5$, whereas Eq. (11) holds true for $\beta < 9.5$. Eqn. (9), (10), and (11) can be applied up to 600 °C. For $CH_4$ equilibrium temperatures in the range 150 to 500°C, the $H_2O$ partial pressure ($P_{H_2O}$ in bar), is obtained using the

following relation ($T$ in K):

$$\log P_{H_2O} = 5.3323 - \frac{1986.4}{T}.$$                    (12)

For $CH_4$ equilibrium temperatures in the interval 500 to 600°C, $P_{H_2O}$ (in bar) is calculated utilizing the following polynomial ($T$ in K):

$$\log P_{H_2O} = \frac{-4.2374E+09}{T^3} + \frac{1.4105E+07}{T^2} - \frac{1.6926E+04}{T} + 10.215.$$                    (13)

The total fluid pressure ($P_{tot}$ in bar) is then computed by means of the equation:

$$P_{tot} = P_{H_2O} \cdot \left(1 + \frac{X_{CO_2}}{1 - X_{CO_2}}\right).$$                    (14)

The $H_2S$ equilibrium temperature is computed using the following relation:

$$T_{H_2S}(°C) = [(-0.0000622619 \cdot X_{CO_2}^4 + 0.00006685541 \cdot X_{CO_2}^3 - 0.00002653208 \cdot X_{CO_2}^2 + 0.000003056223 \cdot X_{CO_2} +$$

$$0.000001021845) \cdot \gamma^4 + (0.00280445 \cdot X_{CO_2}^4 - 0.002992959 \cdot X_{CO_2}^3 + 0.001167822 \cdot X_{CO_2}^2 - 0.0001254123 \cdot X_{CO_2} -$$

$0.00004531143) \cdot \gamma^3 + (-0.0470429 \cdot X_{CO_2}^4 + 0.04991906 \cdot X_{CO_2}^3 - 0.01917336 \cdot X_{CO_2}^2 + 0.001920842 \cdot X_{CO_2} + 0.0007510058) \cdot$

$\gamma^2 + (0.3483213 \cdot X_{CO_2}^4 - 0.367639 \cdot X_{CO_2}^3 + 0.1391247 \cdot X_{CO_2}^2 - 0.01297486 \cdot X_{CO_2} - 0.005414418) \cdot \gamma - 0.96068 \cdot X_{CO_2}^4 +$

$1.008786 \cdot X_{CO_2}^3 - 0.3764276 \cdot X_{CO_2}^2 + 0.03258146 \cdot X_{CO_2} + 0.01498928]^{-1} - 273.15$ (15)

The $P_{H_2O}$ (in bar) at the H₂S equilibrium temperature is calculated using the following polynomial ($T$ in K):

$$log\, P_{H_2O} = \frac{-1.0521E+09}{T^3} + \frac{2.3948E+06}{T^2} - \frac{2.7508E+03}{T} + 4.5720.$$ (16)

The total fluid pressure ($P_{tot}$ in bar) is then computed by means of equation (14). The CH₄ concentration in the H₂S equilibration zone is given by the following relation ($T_{H_2S}$ in °C):

$log\, X_{CH_4\,@\,T_{H_2S}} = log\, X_{H_2S} + 0.000000000004960196 \cdot T_{H_2S}^4 - 0.00000001553066 \cdot T_{H_2S}^3 + 0.00001833542 \cdot T_{H_2S}^2 -$

$0.009852548 \cdot T_{H_2S} - 3.379594.$ (17)

Eqn. (15), (16), and (17) can be applied up to 1000 °C. The computed equilibrium temperatures and related total fluid pressures

as well as the CH₄ concentrations in the H₂S equilibration zone are reported in Table S1, in which all previous equations are programmed in EXCEL.

## 3 Results

The computed CO-, CH₄- and H₂S equilibrium temperatures are shown in Fig. 2 for all the gas samples collected from both Bocca Grande, from June 1983 to January 2022, and Bocca Nuova, from March 1995 to January 2022 (data from Buono et

al., 2023 and references therein). The corresponding total fluid pressures are displayed in Fig. 3. To facilitate the comparison between numbers, the CO-, CH₄- and H₂S equilibrium temperatures and related total fluid pressures calculated for Bocca Grande were subdivided in 24 discrete time intervals and the average and standard deviation for each time interval were computed and reported in Tables D1 and D2 in Appendix D. The following observations can be drawn from Figs. 2 and 3 and Tables D1 and D2:

(i) Reaction (1) indicates low and nearly constant CO-equilibrium temperature and total fluid pressure from 1983 to 2022, 217±9°C, 24.5±4.0 bar, for Bocca Grande, and 219±6°C, 25.4±2.9 bar, for Bocca Nuova[2].

(ii) In contrast, the CH₄-equilibrium temperature and total fluid pressure obtained from reaction (2) increase gradually and significantly with time, from 246±8°C, 38.5±5.6 bar in June 1983-July 1984 to 589±8°C, 1226±46 bar in 2020, for Bocca Grande, and attained 622±12°C, 1401±68 bar in 2020, for Bocca Nuova. Slightly lower values are estimated for the samples

collected in 2021-2022, namely, 580±8°C, 1186±52 bar, for Bocca Grande, and 615±9°C, 1350±54 bar, for Bocca Nuova. Nevertheless, the 2020 values compare with those of 2021-2022 considering short-time changes.

---

[2] Somewhat higher temperatures and total fluid pressures were computed for the 2010-2021 period by Chiodini et al. (2021) using the $CO/CO_2$ geothermometer and the redox buffers of either D'Amore and Panichi (1980) or that of the Campanian Volcanoes (Chiodini and Marini, 1998), namely, 218-267°C and 27-60 bar and 238-287°C and 37-78 bar, respectively.

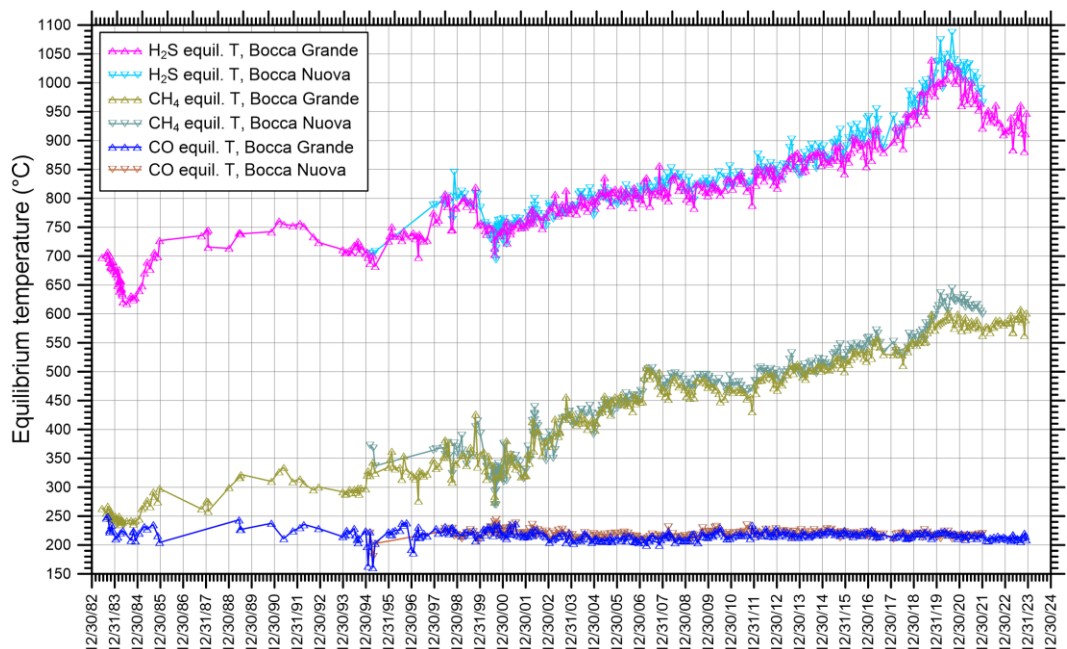

**Figure 2: Chronogram of CO-, CH₄- and H₂S equilibrium temperatures for Bocca Grande, from June 1983 to January 2022, and from Bocca Nuova, from March 1995 to January 2022, calculated from the data of Cioni and coworkers and Chiodini and coworkers (Buono et al., 2023 and references therein).**

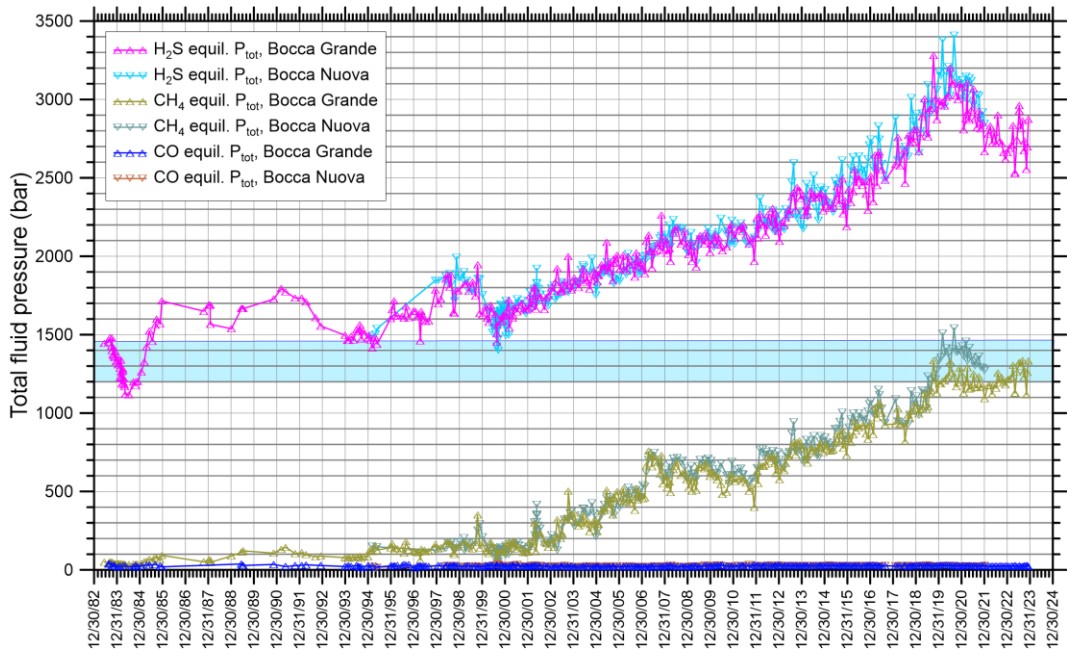

**Figure 3. Chronogram of CO-, CH₄- and H₂S total fluid pressures for Bocca Grande, from June 1983 to January 2022, and for Bocca Nuova, from March 1995 to January 2022, calculated from the data of Cioni and coworkers and Chiodini and coworkers (Buono et al., 2023 and references therein). The strip of sky-blue colour indicates the external pressure expected at a depth of 6.5 – 7.5 km, where reaction (3) is assumed to attain the equilibrium condition.**

(iii) The $H_2S$-equilibrium temperature and total fluid pressure related to reaction (3) experienced a progressive and considerable increment with time as well, from 667±24°C, 1308±102 bar in June 1983-July 1984 to 1010±14°C, 3039±73 bar in 2020, at Bocca Grande, and achieved 1039±25°C, 3162±129 bar in 2020, at Bocca Nuova. Weakly smaller values are obtained for the samples collected in 2021-2022, namely, 975±23°C, 2901±120 bar, for Bocca Grande, and 1008±20°C, 3012±97 bar, for Bocca Nuova. Nevertheless, the 2020 values overlap those of 2021-2022 taking into account short-time variations. To be noted that $H_2S$-equilibrium temperatures are in satisfactory agreement with those of 880-1020°C obtained extrapolating the geothermal gradient of ca. 134°C/km measured from 2 to 3 km depths in the San Vito 1 well (Marini et al., 2022).

The increasing $CH_4$ and $H_2S$ equilibrium temperatures with time implies that fluids come from either (1) progressively deeper zones of the Solfatara magmatic-hydrothermal system, with gradually higher temperatures and fluid pressures with time, or (2) the same deep permeable zones which underwent a progressive increment in temperature and fluid pressure with time. The second implication was adopted in the following discussion because it explains the pressurization of the system and the consequent ground uplift, on which there is a consensus in the scientific literature, whereas the first implication was dismissed as it does not explain these on-going processes.

## 4 Discussion

### 4.1 Noteworthy geo-volcanological aspects and findings of deep geothermal wells

The Campi Flegrei are within the Campanian Plain, a Neogene tectonic graben filled by a sequence of clastic and volcanoclastic sediments and volcanic rocks covering the Mesozoic carbonate basement which has been lowered to depths of some kilometres (Cassano and La Torre, 1987; Zollo et al., 2008). At least two large-scale explosive eruptions occurred in the Campi Flegrei. The most important one, known as Campanian Ignimbrite (CI) eruption, occurred 39.85 ± 0.14 ka BP (Giaccio et al., 2017), and generated either a single caldera (e.g., Rosi et al., 1983) or a nested caldera (e.g., Barberi et al., 1991; Acocella, 2008). The second most significant eruption is known as Neapolitan Yellow Tuff (NYT) eruption, occurred 14.9 ± 0.4 ka BP (Deino et al., 2004) from several vents and either reactivated both compartments of the nested CI caldera (e.g., Acocella, 2008) or reactivated the inner CI caldera (e.g., Barberi et al., 1991) or produced a new caldera (e.g., Lirer et al., 1987). In the time span between the CI and the NYT eruptions, volcanic activity was submarine, whereas the post-NYT activity was mainly subaerial (Rosi et al., 1983).

The early inference of Rittmann (1950) on the occurrence of a large caldera collapse in the Campi Flegrei was confirmed by the volcano-stratigraphic and structural investigations of Rosi et al., (1983) who recognized that the Campi Flegrei caldera formed as a consequence of the CI eruption, described the geological evolution of Campi Flegrei and, *inter alias*, mapped the caldera rim portions identifiable in the field (see also the "Geological and gravimetric map of Phlegrean Fields at the 1:15,000 scale" of Principe et al., 1987). A few years later, Lirer et al., (1987) recognized a smaller caldera which they attributed to the NYT eruption. A remarkable step forward was made by Barberi et al. (1991) who carried out a synthesis work by merging gravimetric and aeromagnetic data, both on-land and offshore, with the findings of surface geological and volcanological

surveys and those of the deep geothermal wells drilled by AGIP-ENEL in the '70s and '80s. In this way, they redefined the geometry of both the outer and inner calderas, suggesting that both were produced as a direct consequence of the CI eruption.

In more detail, according to Barberi et al. (1991): (i) the outer caldera rim (the southern portion of which was reconstructed based on offshore data) is indicated by the outer series of gravity highs (Fig. 4) distributed along a subcircular structure of ~ 13 km in diameter, whereas (ii) the inner caldera rim is marked by the inner circular belt of gravity highs and delimits a more collapsed central zone of ~11-12 km in diameter with a total drop of ~1.6 km. A circular sector of ~1-2 km in width (increasing northwards) and 0.7-0.8 km as maximum drop separates the inner caldera structure from the outer one.

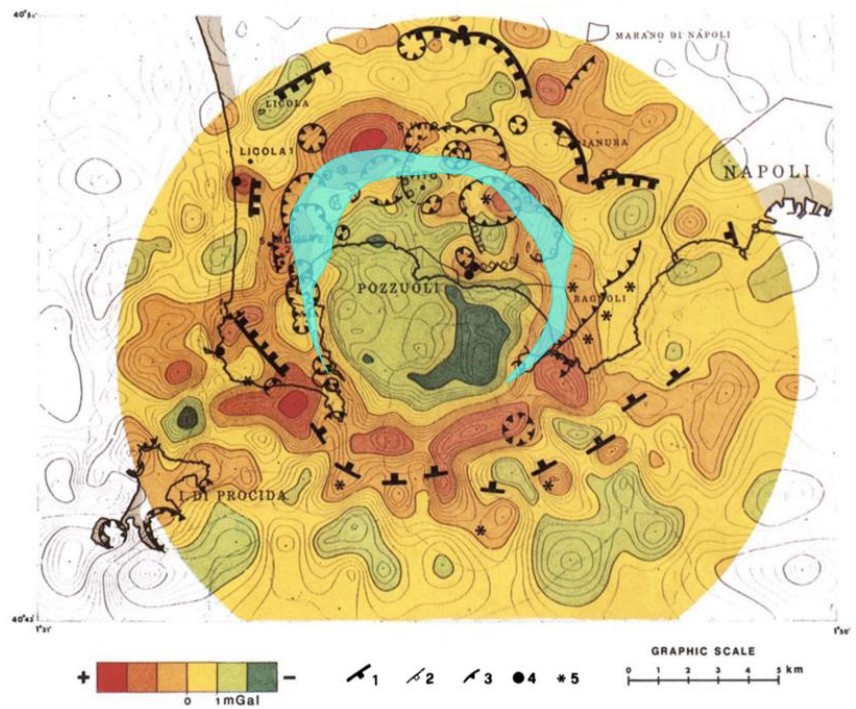


**Figure 4. Map of Bouguer anomaly high-pass filter, based on 770 ground and 500 sea-bottom homogeneously distributed gravity stations (measurements by the Italian Geological Survey and AGIP), with a mean density of ~ 5 stations per km² (from Barberi et al., 1991, modified; Elsevier licence number 5601951169171 on Aug 04, 2023). Also shown are volcanic structures (1=caldera rim; 2=minor, post NYT, volcano tectonic collapsed areas; 3=crater rim; 4=lava dome; 5=eruptive centre) and the boundary of vertical ground movement computed by Bevilaqua et al. (2020; area of cyan colour). The Solfatara is the crater immediately to the east of Pozzuoli.**


The gravimetric low present in the central part of the Bouguer anomaly map of Fig. 4, coinciding with the inner caldera block, deserves further attention because it is circumscribed by the boundary of vertical ground movements computed by Bevilaqua et al. (2020) applying the Radial Interpolation Method to real geodetic data collected at the Campi Flegrei caldera in selected

time intervals during the last 39 years. The implication is that ground deformation affects the inner caldera block only. The gravimetric low may be controlled by different factors, such as: (i) the high thickness of the low-density pyroclastic filling, (ii) the lack of dense lava bodies, (iii) the lowering of the isotherms and of the iso-density lines related to the hydrothermal

circulation, and (iv) the greater depth of the thermometamorphic complex (Barberi et al., 1991). Nevertheless, considering that the circular-shaped gravity minimum corresponds with the area affected by ground movements and accepting the hypothesis that ground deformation is controlled by pressurization-depressurization of a supercritical[3] gas phase, mainly constituted by $H_2O$ and $CO_2$, which saturates the pore spaces of a relatively deep reservoir covered by a caprock, as recognized in several seismic studies (see section 4.4), we propose that gas saturation is the cause or the main contributing cause of the decrease in density and gravity.

The occurrence of a gas-saturated relatively deep reservoir is supported by the findings of the vertical geothermal well San Vito 1 (SV-1 for short), which is located within the area of the gravimetric low and ground deformation (Fig. 4). The SV-1 geothermal well was drilled by AGIP-ENEL in the early '80s and reached a total depth of 3046 m, where a temperature >419°C was estimated to be present (Bruni et al., 1985). At depths of 2500-2800 m b.g.l. and temperatures of 360-385°C, the SV-1 geothermal well encountered a level of altered rocks with abundant hydrothermal quartz (Chelini and Sbrana, 1987). The first short-term production test of well SV-1 was ended by killing the well due to the rapid temperature increase at the well head, not rated for temperatures >300°C (Baron and Ungemach, 1981). During the test, a well-head pressure $P_{tot}$ of 69.6 bar-a and a well-head temperature of 222°C were measured, at the same time. At 222°C and pure water saturation, $P_{H_2O}$ is 24.1 bar, from the Steam Tables (Lemmon et al., 2023), and $P_{tot} - P_{H_2O} \cong P_{CO_2}$ is 45.5 bar, assuming that $CO_2$ is by far the main non-condensable gas constituent. Furthermore, the $X_{H_2O}$ and $X_{CO_2}$ of the gas phase at well head are 24.1 bar/69.6 bar = 0.35 and 45.5 bar/69.6 bar = 0.65, based on the Dalton's law. Summing up, previous considerations mainly based on the results of the AGIP-ENEL geothermal exploration suggest that a supercritical gas phase rich in $H_2O$ and $CO_2$ accumulates in volcanic and marine deposits strongly affected by thermometamorphic alteration below a quartz-rich caprock. The top of this relatively deep reservoir of supercritical fluids is found at ~2.8 km depth, while its areal extension coincides with the inner caldera block, marked by the gravimetric low, and the ground deformation area (Fig. 4). Therefore, it is possible that the pressurization of the ~2.8-km-deep reservoir of supercritical fluids is the "engine" of the ground uplift begun in 2006, also accompanied by anomalous shallow seismicity and increase in fumarolic emission. Since the Solfatara-Pisciarelli fumarolic area is found near the center of the inner caldera block, it might be a sort of "exhaust valve" of the ~2.8-km-deep reservoir. This discussion is resumed in section 4.4.

## 4.2 The zone of CO equilibration

Considering that the total fluid pressure of CO equilibrium remained ~25 bar during the last 38 years and assuming that it is balanced by an external pressure of the same value, it can be inferred that CO equilibrium is attained in a shallow reservoir whose top is located at ~250 m depth. This depth agrees with that of the bottom of the low-resistivity, clay-rich caprock present below the Solfatara crater as indicated by audiomagnetotellurics (Siniscalchi et al., 2019). The shallow reservoir is assumed

---

[3] The adjective supercritical is used to indicate temperatures and pressures higher than those of the critical point of pure water, i.e., 374 °C and 222 bar.

to correspond with the relatively conductive unit (10-30 $\Omega \cdot m$) which is situated below the caprock and has an average thickness of ~200 m. The areal extension of the shallow reservoir corresponds to that of the Solfatara diffuse degassing structure, ~1 km$^2$, as indicated by high $CO_2$ fluxes (Cardellini et al., 2017; Fig. 1b) and occurrence of advanced argillic alteration (Piochi et al., 2015).

## 4.3 Equilibrium versus disequilibrium between CO and CH$_4$

The disequilibrium between CO and CH$_4$ is a very likely condition in hydrothermal-magmatic environments, especially when the residence time of the fluid in the system is relatively short, because CO is a fast-reacting species and CH$_4$ is one of the slowest species to react (e.g., Giggenbach 1987). This does not exclude that equilibrium between CO and CH$_4$ can be reached, if the residence time of the fluid in the system is long enough. Actually, both conditions occurred, at different times, in the shallow reservoir below the Solfatara. In fact, CO- and CH$_4$ equilibrium temperatures and total fluid pressures were similar to each other, within uncertainties, until July 1984, while the difference between the CO- and CH$_4$ equilibrium temperatures and total fluid pressures increased more and more in the following years (Figs. 2 and 3; Tables D1, D2 in Appendix D, and S1). Since the attainment of CO-CH$_4$ equilibrium requires a long time, the similarity between CO- and CH$_4$ equilibrium temperatures and total fluid pressures, in the period June 1983 - July 1984, indicates that the residence time of fluids in the shallow reservoir was long enough to allow the attainment of CO-CH$_4$ equilibrium and that the inflow of deep gases from below was nil to negligible. In other words, the shallow reservoir behaved as a closed system or nearly so, at that time, as proposed in the conceptual model of Cioni et al. (1984). The shallow reservoir opened in July-September 1984 and was affected, in the following years, by a time-increasing inflow of deep fluids, mostly coming from a degassing magma batch, as postulated by the conceptual model of Caliro et al. (2007) and adopted in the subsequent studies of Chiodini and coworkers (Caliro et al., 2014; Chiodini, 2009; Chiodini et al., 2010, 2011, 2012, 2015, 2016, 2017a, b, 2021; Buono et al., 2023). The change from closed to open state of the Solfatara magmatic-hydrothermal system explains the differences between the two models, that are both valid because they refer to two distinct time lapses.

Therefore, it is advisable to use reaction (1), involving CO but not CH$_4$, and reaction (2), including CH$_4$ but not CO, for geothermometric-geobarometric purposes. In other terms, it is better to consider CO and CH$_4$ separately, rather than using the following reaction:

$$CO + \tfrac{1}{2} H_2O = \tfrac{3}{4} CO_2 + \tfrac{1}{4} CH_4, \tag{18}$$

as it involves both gas species with different stoichiometric coefficients, 1 for CO and ¼ for CH$_4$. Thus, in case of CO-CH$_4$ disequilibrium, the equilibrium temperature is meaningless as it is the weighted average of the CO- and CH$_4$ equilibrium temperatures given by reactions (1) and (2), respectively. Similar considerations apply to equilibrium pressures. Irrespective of these issues, reaction (18), together with reaction (1), was taken into account in several studies of the Solfatara hydrothermal-magmatic system (Cioni et al., 1984, 1989; Chiodini and Marini, 1998; Caliro et al., 2007, 2014; Chiodini, 2009; Chiodini et al., 2010, 2011, 2012, 2015, 2016, 2017a, b, 2021; Buono et al., 2023).

## 4.4 The zone of CH$_4$ equilibration

Owing to its sluggish behavior, CH$_4$ was often considered to be a tracer instead of an indicator, although in some studies, e.g., Moretti et al. (2017), CH$_4$ was treated as a reactive species. The latter approach is corroborated by the good agreement between the CH$_4$ equilibrium temperature given by reaction (2) and the temperature of isotopic CH$_4$-CO$_2$ equilibrium, in spite of the limited number of isotope data available for the Solfatara fluids (Caliro et al., 2007; Fiebig et al., 2013, 2015). Assuming that this agreement is not fortuitous, it is legitimate to conclude that reaction (2) provides meaningful geothermometric results, at least for the Solfatara magmatic-hydrothermal system. Nevertheless, the depth of the reservoir where CH$_4$ and CO$_2$ equilibrate chemically and isotopically remains a matter of discussion. Since the attainment of this condition requires a time interval long enough, that is, the residence of the fluid into a sufficiently large reservoir, a possible candidate is the ~2.8-km-deep reservoir, whose areal extension coincides with that of the inner caldera block (Fig. 4; see section 4.1). Further information on this reservoir of supercritical fluids is provided by modeling of both the active seismic reflection data of the SERAPIS survey (Zollo et al., 2003; Judenherc and Zollo 2004; Zollo et al., 2008) and the passive seismic data of the 1982-1984 bradyseismic crisis (Vanorio et al., 2005; Chiarabba and Moretti 2006; Battaglia et al., 2008; De Siena et al., 2017), indicating that it extends from 2.7 to 4 km depth.

Recalling the available geological knowledge (section 4.1), the 2.7-4 km deep reservoir is covered by an impermeable layer generated by self-sealing, chiefly quartz deposition, at temperature of ~400°C, in line with the general conceptual model of volcano-hosted magmatic-hydrothermal systems of Fournier (1999). According to the Fournier's model, the quartz-rich layer separates: (a) the underlying deep-magmatic domain, where hypersaline brines and gases exsolved from the underlying crystallizing magma accumulate at lithostatic pressure within a volume of plastic rocks, from (b) the overlying shallow-hydrothermal domain, where hydrothermal fluids of meteoric and/or marine origin circulate through brittle rocks at hydrostatic pressure. Fournier (1999) also recognized that the quartz-rich self-sealed layer is broken, from time to time, by an uprise of magma or other processes determining the fast spill of hypersaline brines and gases from the plastic-magmatic domain into the brittle-hydrothermal domain, at smaller pressure and temperature. The resultant increase in fluid pressure and temperature within the brittle-hydrothermal domain triggers faulting and fracturing, with an ensuing increase in permeability and the discharge rate of magmatic–hydrothermal fluids. The Fournier's model is perfectly applicable to the Solfatara magmatic-hydrothermal system, as already recognized by other authors (e.g., Lima et al., 2009, 2021; Smale 2020; Kilburn et al., 2023), and allows us to understand both its structure and its evolution over time.

## 4.5 The zone of H$_2$S equilibration

The occurrence of reaction (3) is supported by the widespread presence of calcite and anhydrite veins in the carbonate-evaporite geothermal systems of Central Italy (Marini and Chiodini 1994), such as Latera, where both anhydrite and calcite are very abundant authigenic minerals also in the contact-metasomatism paragenesis (Cavarretta et al., 1985), that is, near the magma chamber, which was penetrated by >350 m by deep geothermal drilling, being positioned at ~ 2 km depth (Turbeville, 1993).

The Mesozoic carbonate sequence crops out all around the Campanian Plain, but it is found at depths greater than ~4 km in the Campi Flegrei, based on the evidence provided by seismic data (Zollo et al., 2008), whereas a melt zone occurs at depths of ~8.0 to ~8.5 km. Assuming the presence of a level of skarn and marble separating the two units, the carbonate sequence is expected to be situated from ~ 4 km to ~7.5 km depth. Thus, the $H_2S$ equilibrium temperature is assumed to mark the base of the carbonate sequence, where the acidic fluids released from the underlying degassing magma are quickly neutralized through interaction with carbonate minerals, at depths of 6.5–7.5 km, to be considered as an educated guess.

Incidentally, magmatic $SO_2$ reacts to form $H_2SO_4$ and $H_2S$ through the following disproportionation reaction (Holland, 1965):

$$4\ SO_{2(aq)} + 4\ H_2O_{(l)} \rightarrow H_2S_{(aq)} + 3\ H_2SO_{4(aq)}, \tag{19}$$

upon dissolution either in groundwater or in the liquid phase formed through condensation of magmatic gases. Then, $H_2SO_4$ is neutralized through the following reaction:

$$CaCO_{3(s)} + H_2SO_{4(aq)} \rightarrow CaSO_{4(s)} + CO_{2(aq,g)} + H_2O_{(l)}, \tag{20}$$

causing the conversion of calcite into anhydrite, the two minerals controlling the $H_2S$ geothermometer [see reaction (3)].

The external (overburden) pressure at depths of 6.5–7.5 km is expected to be of 1330±135 bar, as indicated by the strip of sky-blue color in Fig. 3. Interestingly, in June 1983–July 1984, total fluid pressure at 6.5–7.5 km depth balanced the external pressure and the fluids present at the base of the carbonate sequence could not flow upward, in line with the conceptual model of Cioni et al. (1984; see above). Then, since September 1984, fluid pressure started to exceed the external pressure, although the difference between the fluid pressure and the overburden pressure remained relatively small until 2000-2001. A continuous and considerable increase in fluid pressure occurred afterwards and it was particularly important since 2016. This growth in the fluid pressure at 6.5–7.5 km depth with time explains the provenance of fumarolic fluids from the deep-magmatic portion of the Solfatara magmatic-hydrothermal system since September 1984, in agreement with the conceptual model of Caliro et al., (2007; see above), as well as the increasing degassing process observed at the surface (Chiodini et al., 2021 and references therein).

The $H_2S$ equilibrium temperature can approach but cannot overcome 1120 °C, the temperature of the trachybasaltic magma present at depth (Caliro et al., 2014). Consistent with this expectation, the maximum computed $H_2S$ equilibrium temperatures are 1040°C, for the Bocca Grande sample collected on October 8, 2019, and 1087°C, for the Bocca Nuova fluid sampled on September 1, 2020.

The $CH_4$ concentration at the $H_2S$ equilibrium temperature has average of 0.00571±0.00194 (1σ) μmol/mol and range of 0.00355-0.0137 μmol/mol for Bocca Grande and average of 0.00429±0.00163 (1σ) μmol/mol and range of 0.00211-0.0171 μmol/mol for Bocca Nuova. These very small $CH_4$ concentrations are not surprising for magmatic fluids somewhat modified by absorption in deep brines and interaction with carbonate rocks. Consequently, it can be inferred that most $CH_4$ discharged at the surface is generated through different reactions, such as (2) and (18), upon cooling-depressurization of the gas mixture leaving the zone of $H_2S$ equilibration, and adjustment to the final equilibrium value in the reservoir of supercritical fluids situated at depths of 2.7-4 km.

## 4.6 Our new conceptual model of the Solfatara magmatic-hydrothermal system

Based on previous discussion, we propose a new conceptual model of the Solfatara magmatic-hydrothermal system, which is consistent with the geological-geophysical context of the Campi Flegrei and the hydrothermal mineralogy, both present at the surface in the Solfatara crater (Piochi et al., 2015) and encountered by deep geothermal wells (Chelini and Sbrana 1987). It includes the following units (from top to bottom, Fig. 5):

(1) The cap-rock of the shallow reservoir (0-0.25 km depth) constituted by volcanic deposits affected by advanced argillic alteration, near the surface, and by argillic alteration far from it. Its areal extension corresponds to that of the Solfatara diffuse degassing structure (Chiodini et al., 2001).

(2) The shallow reservoir (0.25-0.45 km depth) hosted in volcanic deposits. It is a steam and gas pocket with areal extension of ~1 km$^2$, matching that of the Solfatara diffuse degassing structure, and volume of ~0.2 km$^3$.

(3) An impermeable sequence (0.45-2.7 km depth) comprising pre- and post-caldera volcanic and marine deposits, affected by phyllitic alteration, in the upper part, and by propylitic alteration, in the lower portion, with an impermeable quartz-rich layer produced by self-sealing at the base. Since the propylitic alteration causes an extensive lithification of the primary materials, the lower portion of this sequence has brittle behavior, is prone to fracture, and could locally host small aquifers. The areal extension of this unit corresponds to the inner caldera block.

(4) The intermediate reservoir (2.7-4 km depth) hosted in volcanic and marine deposits affected by thermometamorphic alteration, determining a broad textural rearrangement of primary lithotypes. The intermediate reservoir is the source of ground uplift and associated shallow seismicity due to the presence of over-pressurized supercritical fluids. The areal extension of the intermediate reservoir matches the inner caldera block. Nevertheless, it could be compartmentalized rather than a single aquifer, as suggested by the piecemeal collapse mechanism of the inner caldera (Capuano et al., 2013) and the distribution of seismic events during the ongoing unrest (https://terremoti.ov.ingv.it/gossip/flegrei/index.html last access 15th December 2024).

(5) A thick carbonate pile (4-6.5 km depth) acting as aquiclude probably due to nil to negligible fracturing and dissolution, by analogy with the geothermal well Nisyros-1, which crossed an 830-m thick sequence of carbonate rocks behaving as aquiclude separating the two permeable zones and ending into the apophyses of a dioritic intrusion and related thermometamorphic rocks (Ambrosio et al., 2010).

(6) The deep reservoir (6.5-7.5 km depth) hosted in carbonate rocks affected by fracturing and dissolution-precipitation processes driven by magmatic fluids. Its lateral extension is expected to reflect that of the underlying melt zone.

(7) A deep aquiclude (7.5-8 km depth) constituted by skarn and marble, produced by metasomatic and thermometamorphic processes. The aquiclude behavior of skarn and marble is supported by their nil porosity (e.g., Kerrick, 1977).

(8) The melt zone (depths ≥ 8 km) storing a trachybasaltic magma (Caliro et al., 2014) and extending over the whole outer caldera.

In principle, our new conceptual model of the Solfatara magmatic-hydrothermal system can be used as reference to compute the residence time spent by the fluids in each reservoir. In practice, it is not possible to obtain reliable results, even for the shallow reservoir, due to the considerable uncertainties affecting the calculations (see Appendix E).

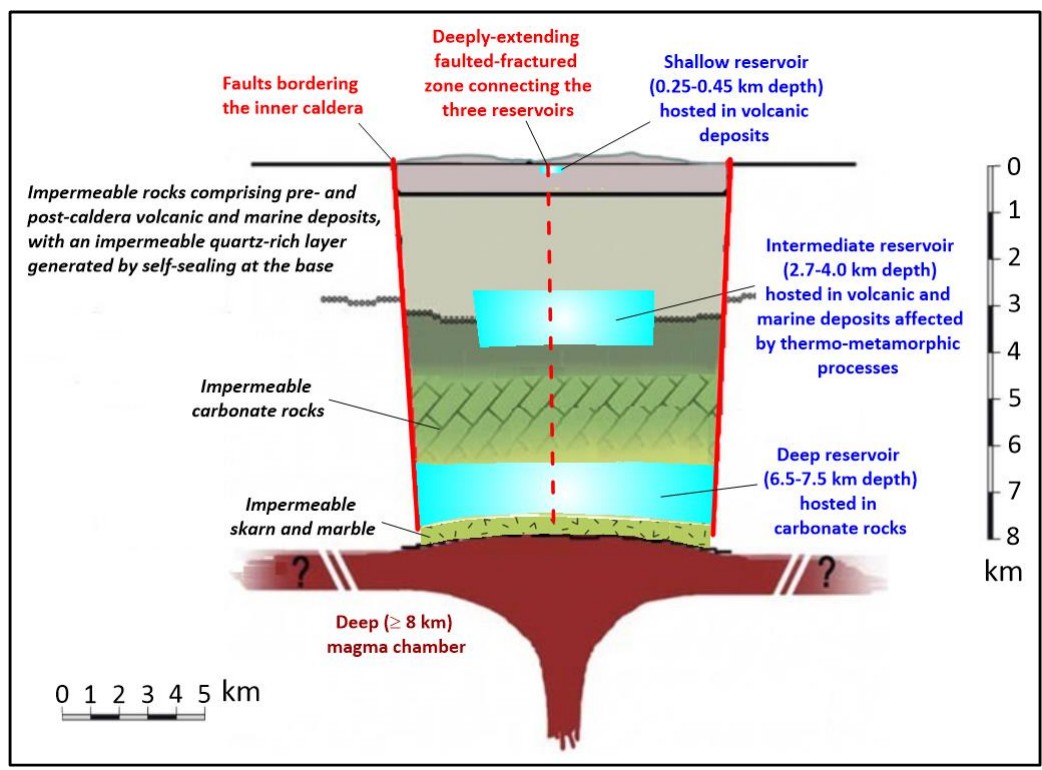

**Figure 5. Conceptual model cross-section of the Solfatara hydrothermal-magmatic system at Campi Flegrei caldera (modified from Moretti et al., 2020; Elsevier license number 5234800567871 on Jan 23, 2022), showing the shallow, intermediate, and deep reservoirs.**

### 4.7 Time changes of temperature and total fluid pressure below the Solfatara

A schematic graphical presentation of the evolution with time, between September 1984 and January 2022, of the temperatures and pressures present at different depths in the Solfatara magmatic-hydrothermal system is given by the temperature and total fluid pressure profiles of Figs. 6 and 7, respectively. The two graphs were prepared assuming that CO-, CH$_4$-, and H$_2$S-equilibrium temperatures (Table D1 in Appendix D) and related total fluid pressures (Table D2 in Appendix D) below the Solfatara refer to depths of 0.25-0.45 km, 2.7-4 km, and 6.5-7.5 km, respectively (see previous section), whereas the

temperatures and pressures in the impermeable units are assumed to vary linearly with depth. Total fluid pressure at the top of the intermediate and deep reservoir, P$_{tot,T}$ (bar), was computed as a function of temperature, whereas the total fluid pressure at the bottom of both reservoirs, P$_{tot,B}$ (bar), was calculated using the relation:

$$P_{tot,B} = P_{tot,T} + \rho \cdot g \cdot 0.01 \tag{21}$$

where ρ (kg/m³) is the density of the H₂O-CO₂ gas mixture and g = 9.80665 m/s² is the conventional standard value of the gravity acceleration. Densities were computed as a function of temperature and $X_{CO_2}$, using polynomials obtained from the molar volumes reported in Marini et al. (2022). The temperature and pressure at the surface were set equal to the measured Bocca Grande outlet temperature and the atmospheric value, respectively, whereas a constant temperature of 1120 °C and a constant pressure of 2879 bar were imposed at 8 km depth based on the characteristics of the trachybasaltic magma present below the Campi Flegrei (Caliro et al., 2014) and the results of magmatic degassing modeling performed using the model of Papale et al. (2006) on H₂O-CO₂ solubility in magmas (Marini et al., 2022).

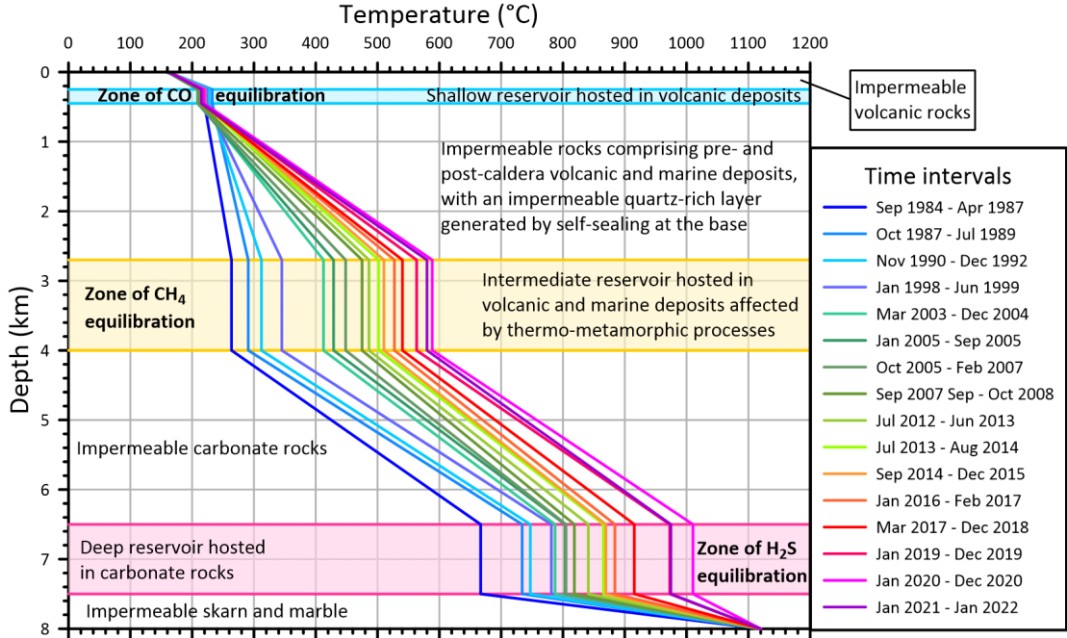

**Figure 6. Time changes in September 1984-January 2022 of the temperature vs. depth profile along a hypothetical borehole drilled in the Solfatara crater. Also shown are the main lithological characteristics of the three reservoirs, where CO, CH₄, and H₂S equilibrate, and of the impermeable zones interposed between the three reservoirs, positioned above the shallow reservoir, and situated below the deep reservoir (see text for details).**

The temperature and total fluid pressure profiles of Figs. 6 and 7, respectively, are expected to be encountered along a hypothetical vertical borehole drilled in the Solfatara crater to a total depth of 8 km. However, it must be recalled that the three reservoirs, where CO, CH₄, and H₂S equilibrate, are connected to each other by a deeply-extending faulted-fractured zone which attains a total depth of ~8 km and acts as conduit for the uprise of the fluids discharging at Solfatara-Pisciarelli. This faulted-fractured zone was activated during the final phase of the 1982-1984 seismic crisis, along tectonic trends already active in the past (Rosi and Sbrana, 1987), as suggested by the occurrence of low-magnitudo earthquakes at depths of 0-8 km (D'Auria et al., 2011). This fluid flow permits a very efficient advective heat transport from the magma to the surface. Nevertheless, the assumed temperature profile along the hypothetical well implies conductive heat transfer in the impermeable zones between

the three reservoirs, as well as above the shallow reservoir and below the deep reservoir, and by convection, which keeps the
temperature constant, in the three reservoirs.

Fig. 6 shows that the temperature of the shallow reservoir, where CO equilibrates, remained nearly constant with time, whereas a considerable temperature increase affected the intermediate and deep reservoirs, where $CH_4$ and $H_2S$ equilibrate, respectively, as already noted above. A possible temperature decrease occurred in the intermediate and deep reservoirs in 2021, after the peak value of 2020. In spite of the remarkable temperature increment in the deep reservoir, this parameter remained well below

the temperature of the underlying magma, at all times, indicating that the heat transfer from the magma to the overlying rocks has never been interrupted during the last 38 years.

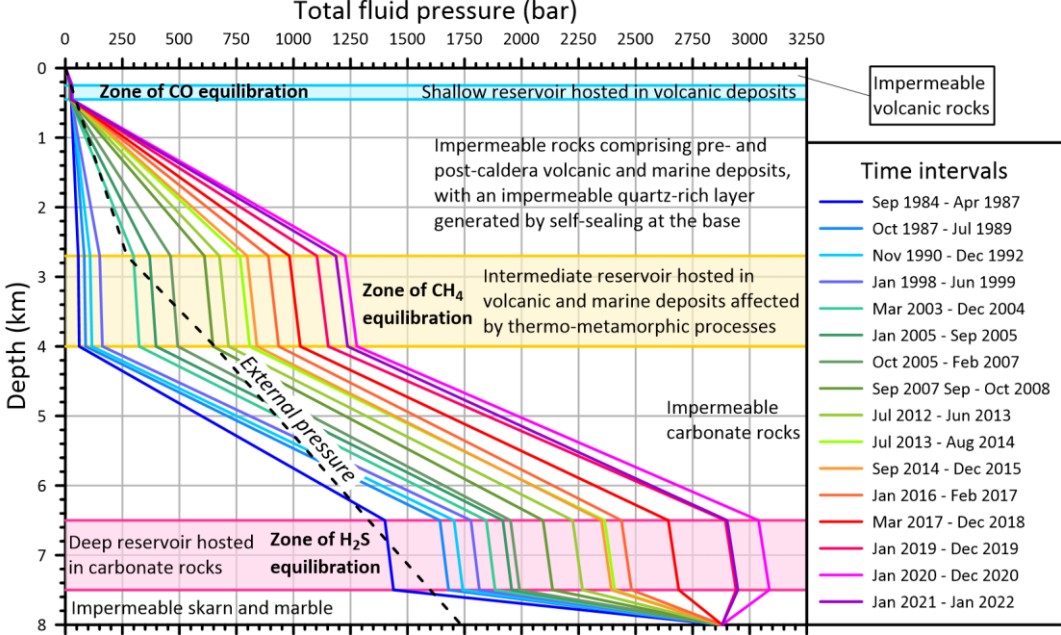

**Figure 7. Time changes, between September 1984 and January 2022, of the total fluid pressure vs. depth profile along a hypothetical borehole drilled in the Solfatara crater. Also shown are (i) the main lithological characteristics of the three reservoirs, where CO,**
**$CH_4$, and $H_2S$ equilibrate, and of the impermeable zones interposed between the three reservoirs, positioned above the shallow reservoir, and situated below the deep reservoir (see text for details), as well as (ii) the external pressure gradient which is assumed to follow the hydrostatic regime above 2.7 km depth and the lithostatic regime below 4 km, with a transition zone between 2.7 and 4 km.**

The following observations can be drawn from Fig. 7: (a) As already noted above, the total fluid pressure of the shallow
reservoir experienced nil to negligible changes with time whereas a remarkable pressurization, progressively increasing with time, impacted the intermediate and deep reservoirs. (b) The zero-pressure gradient typical of vapor-dominated geothermal systems (e.g., White et al., 1971; Truesdell and White, 1973; Grant and Bixley, 2011) occurred in the shallow reservoir, at all times, and in the intermediate reservoir, as long as T < 330°C and P < 150 bar, whereas a non-zero pressure gradient was present in the intermediate reservoir, upon heating and pressurization, and in the deep reservoir, at all times. (c) In particular,

the total fluid pressure in the deep reservoir became slightly higher than that of the underlying magma in 2019, increased further in 2020, and decreased to the 2019 values in 2021. So, the inflow of magmatic gases into the deep reservoir stopped in the period 2019-2021, being prevented by pressures in the deep reservoir greater than or equal to the values of the underlying magma. The question is: how is it possible that total fluid pressure in the deep reservoir became equal to or even higher than that of the underlying magma? A possible explanation is the increasing rate, in 2019-2021, of the gas-producing reactions

occurring in the deep reservoir, because of its continuous heating, with a consequent increase in the partial pressures of relevant gas species and in total fluid pressure. The most important of these gas-producing reactions is the decomposition of impure carbonate rocks, which is generally exemplified by the conversion of calcite and quartz to wollastonite and $CO_2$:

$$CaCO_3 + SiO_2 = CaSiO_3 + CO_2, \tag{22}$$

although other decarbonation reactions are possible (Marini, 2007) and probably occur in the considered system. Thus, the

balance between the total fluid pressure in the deep reservoir and that of the underlying magma appears to be the temporary "on-off switch" of magmatic degassing. (d) The total fluid pressure at the top of the shallow reservoir remained nearly equal to the external pressure in the entire time interval of interest (see section 4.2). In contrast, the total fluid pressure at the top of the intermediate reservoir became greater than the external pressure in the period March 2003-December 2004.

Disregarding isolated spikes and the frequent short-term fluctuations, the chronogram of the overpressure (the difference

between total fluid pressure and external pressure) at the top of the intermediate reservoir (2.7 km depth; Fig. 8a) shows that the intermediate reservoir was not overpressurized before March 2003, whereas a general increase in the overpressure occurred afterwards, with a very weak decrease in 2007-2011 and a moderate decrease in 2020-2021.

The comparison of the chronogram of overpressure at 2.7 km depth with that of the vertical displacement at the center of the inner caldera (Fig. 8b) shows that there is a general correspondence between the two graphs and suggests that an overpressure

of 200-250 bar is necessary to begin the ground uplift, the rate of which increases for higher overpressure values. Nevertheless, the two chronograms decoupled in 2021, when the overpressure decreased moderately but remained very high, with values of either 900-1000 bar or 1000-1150 bar based on Bocca Grande and Bocca Nuova data, respectively, whereas the positive vertical movement continued, although with a somewhat lower rate, evidently because the overpressure was much higher than the initial threshold needed to push up the overlying rocks (see above). Moreover, in the biennium 2020-2021, the ground

uplift was accompanied by an appreciable increment in the frequency of the shallow earthquakes (mostly of low-magnitudo) occurring below the Solfatara area and in the adjacent sector of the Pozzuoli Gulf (Figs. 8c, 8d). These seismic events were accompanied by a remarkable increase in the $CO_2$-rich gas flow from the Solfatara-Pisciarelli degassing structure, indicating the opening of new fractures in the rocks overlying the intermediate reservoir (Chiodini et al., 2021 and references therein) and the consequent increase in degassing from it. This, in turn, might be responsible of the moderate transient decrease in the

overpressure at the top of the intermediate reservoir in 2020-2021. As expected on the basis of our new conceptual model, the hypocenters of these earthquakes are found above the top of the intermediate reservoir or inside it (apart from a few cases) and within the inner caldera (see https://terremoti.ov.ingv.it/gossip/flegrei/index.html/ last access 15[th] December 2024).

A considerable a general increase in the overpressure occurred also at the top of the deep reservoir, as discussed in Appendix F.

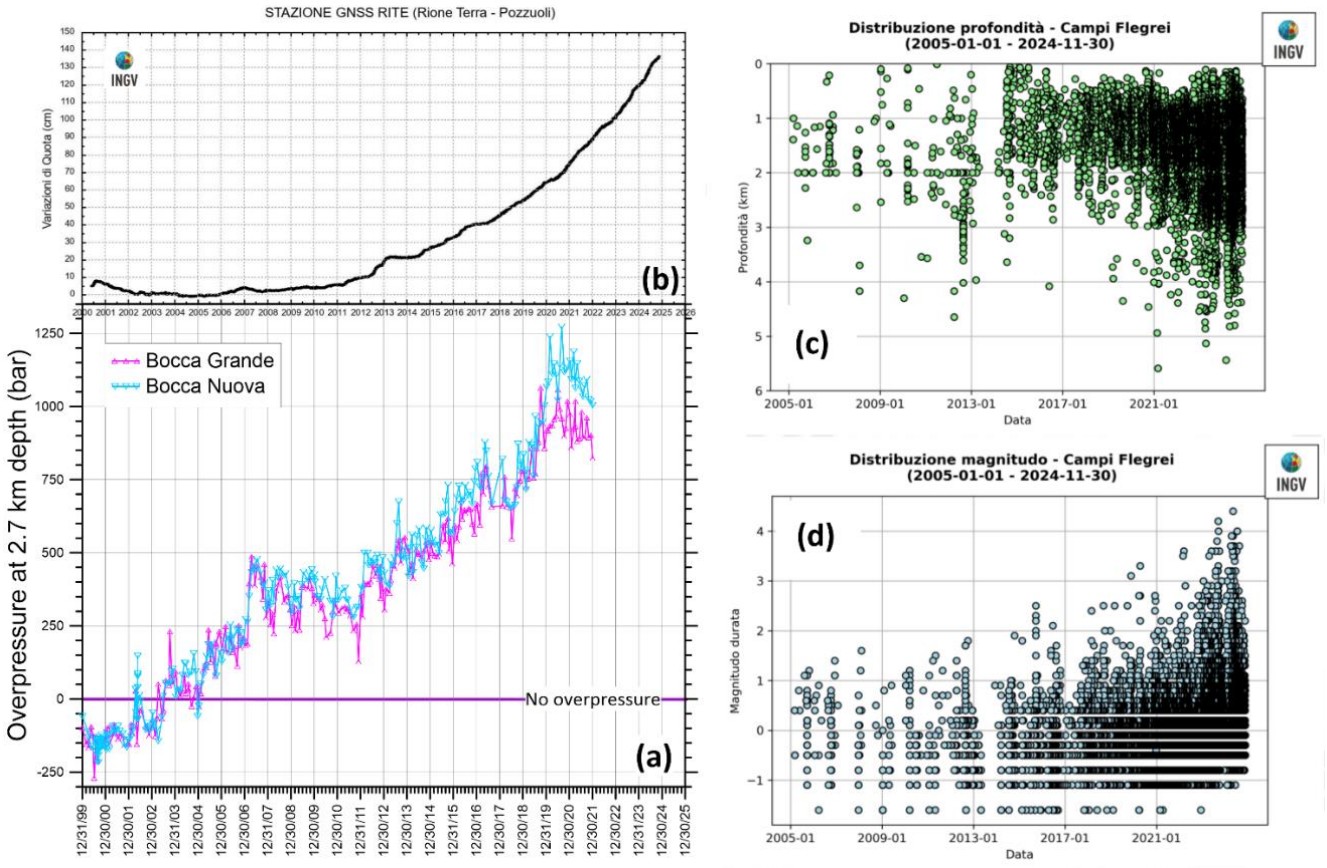

Figure 8. (a) Chronogram of the overpressure at the top of the intermediate reservoir (2.7 km depth), that is, the difference between the time-dependent total fluid pressure (computed from the chemistry of Bocca Grande and Bocca Nuova fumarolic fluids) and the constant external pressure, equal to 270 bar assuming a hydrostatic regime. Also shown are (b) the time series of weekly variations in elevation of the RITE station (Pozzuoli – Rione Terra, near the center of the inner caldera) from 2000 to November 2024, and the chronograms of (c) earthquake depth, and (d) earthquake magnitudo from 2005 to November 2024 (from INGV-Osservatorio Vesuviano, 2024).

## 4.8 Future scenarios and risk mitigation actions

Based on our new conceptual model of the Solfatara magmatic-hydrothermal system described in section 4.6, two future scenarios can be envisaged for the evolution of the current bradyseismic crisis, assuming either (1) the decline of magmatic degassing and heat transfer from the magma to the overlying rocks or, alternatively, (2) the persistence of sustained magmatic degassing and heat transfer from the magma to the overlying rocks, in the next future. In this exercise, we assume the lack of external factors, such as the occurrence of one or more regional earthquakes and the input of fresh magma in the reservoir

positioned at 8 km depth, as well as the uprise of magma at shallower levels[4].. We recall that the three reservoirs considered in our new conceptual model are connected to each other by the deep-reaching fault-fracture zone that opened in the final
phase of the 1982-1984 unrest episode (D'Auria et al., 2011).

In the first scenario, when magmatic degassing will decline to low values, similar to those of the late 1980's-early 2000's, a progressive decrease in the temperature and total fluid pressure will occur both in the deep and in the intermediate reservoirs. Then, the inversion of ground movement will also occur, when total fluid pressure at the top of the intermediate reservoir will be lower than external pressure. Anomalous local seismicity will also terminate.

In our second scenario, as long as sustained magmatic degassing persists and heat is transferred from the magma reservoir at 8 km depth to the overlying rocks, the temperature and total fluid pressure in the deep and in the intermediate reservoirs are expected to increase until the uppermost thresholds are possibly attained. The temperature of the deep reservoir cannot exceed 1120 °C (see section 4.5), whereas the temperature of the intermediate reservoir cannot exceed 700°C approximately, assuming a geothermal gradient similar to those reported in Figure 6 for the impermeable rock sequence interposed between the deep
and the intermediate reservoirs. The corresponding upper thresholds in total fluid pressure are more difficult to be defined, because they depend on fluid chemistry (whose future changes are unpredictable) but, by the same token, they are in the order of 3.5 kbar for the deep reservoir and 1.8 kbar for the intermediate reservoir. Focusing on the intermediate reservoir, its fluid pressure and temperature are regulated by the balance between the flow of magmatic-thermometamorphic fluids entering it from below and the flow of fluids leaving it from the top. The seismic events occurring in the volume of rocks above the
intermediate reservoir, progressively increasing in number and maximum magnitudo during the last years, might determine a gradual weakening of these rocks and the consequent opening of a new fracture zone extending from the intermediate reservoir to the surface, triggering a hydrothermal (phreatic) explosion.

It must be underscored that there is no need to invoke magma movements to trigger a hydrothermal explosion at Campi Flegrei, in that this event might be caused by the pressurization of the intermediate reservoir. This, in turn, is regulated by the time-
increasing inflow of magmatic gases and heat entering it from below, although the pressure threshold triggering the hydrothermal explosion is unknown.

For what concerns the prediction of hydrothermal explosions, in some cases, they are not preceded by precursors and, in other cases, the precursors are few and too close to the event (Barberi et al., 1992; Montanaro et al., 2022). As described in this work, it is possible to mitigate the risk of hydrothermal explosion in the Solfatara magmatic-hydrothermal system monitoring
the pressurization state of the intermediate reservoir using the geo-indicators of Marini et al. (2022).

---

[4] Astort et al. (2024) proposed that the ongoing unrest at Campi Flegrei is governed by both the input of magma into the reservoir at 8 km depth and the transfer of 0.06 to 0.22 km$^3$ of magma from the deep reservoir to shallower levels. It should be considered, however, that while the magma ascent is able to explain the ground uplift, the reverse phenomenon, observed at Campi Flegrei, cannot be explained by magma movements, at least not at the time scale documented for the Campi Flegrei bradyseism.

Of course, it would be much simpler to monitor the state of the intermediate reservoir using one or more geothermal wells, such as the San Vito 1, which unfortunately has been cemented from bottom to top. However, it is possible, not to say advisable, to drill new geothermal wells as also proposed by Lima et al. (2024). Even more important, this action allows one to manage the bradyseism by zeroing the inflation of the intermediate reservoir depressurizing it and consequently cancels the hazard posed by hydrothermal explosions. It requires a considerable initial investment to drill a suitable number of geothermal wells to ~4 km depth[5] and to construct both a geothermal power plant and a mineral recovery plant. However, it provides a considerable economic return, thanks to the exploitation of geothermal energy for electrical production and the recovery of raw materials of utmost interest such as lithium, whose concentration was in the range 146-217 mg/kg in the reservoir liquids of well Mofete-5 (Marini et al., 2022). The feasibility of geothermal exploitation was proven by AGIP-ENEL activities carried out in the '70s and '80s (see above). The obstacles that existed at that time and caused the end of geothermal activities no longer exist today, thanks to the improvements in drilling materials and technologies, as demonstrated by ongoing drilling in several supercritical geothermal systems (e.g., Reinsch et al., 2017).

## 5 Conclusions

The results of the new geothermometers and geobarometers (Marini et al., 2022) and the available geological, volcanological, and geophysical information allowed us to elaborate a new conceptual model of the Solfatara magmatic-hydrothermal system. Our new conceptual model adds further details to previous conceptual models and extends at magmatic depth. Based on our new conceptual model, it was possible: (1) to monitor the temperature and total fluid pressure over a large time interval in the reservoir present at shallow depth below the Solfatara, in the intermediate reservoir present at 2.7-4 km depth in the inner Campi Flegrei caldera, at least in the compartment below the Solfatara, and in the deep reservoir probably extending over the whole outer Campi Flegrei caldera, (2) to explain the evolution of pressurization-depressurization in the intermediate reservoir, acting as the "engine" of bradyseism, and the time changes of total fluid pressure in the deep reservoir, acting as temporary "on-off switch" of magmatic degassing, and (3) to infer the two possible future scenarios of the Solfatara magmatic-hydrothermal system in the lack of external factors, such as the occurrence of regional earthquakes and the input of fresh magma in the reservoir at 8 km depth, as well as the uprise of magma at shallower levels. We showed that the pressurization of the intermediate reservoir might trigger a hydrothermal explosion and we proposed possible actions to mitigate the risk related to events of this type.

We underscore that the achievement of these results has been possible thanks to the availability of a large geochemical database, extending over 40 years, generated in the framework of volcanic surveillance, and a very large multidisciplinary geological, volcanological, and geophysical information obtained not only from surface investigations but also from the deep geothermal wells drilled by AGIP-ENEL in the '70s and '80s.

---

[5] It is unnecessary to drill geothermal wells to 5 km depth or more, as proposed by Lima et al. (2024), because the target of geothermal drilling is the 2.7 to 4.0 km deep intermediate reservoir.

In future studies, the CO, $CH_4$, and $H_2S$ temperatures and total fluid pressures at different times could be used to calibrate the numerical models for simulating the coupled transport of fluids and heat in the porous and fractured media present under the Solfatara crater, improving those developed in previous studies (e.g., Todesco, 2009). The temperatures and total fluid pressures in the shallow, intermediate and deep reservoirs at different times should also be considered to predict the rheological behavior of relevant rocks in the system of interest, thus ameliorating the results of previous investigations (e.g., Kilburn et al., 2023). Thermo-poro-elastic models (e.g., Nespoli et al., 2023) could also be improved taking into account the time changes of the temperatures and total fluid pressures in the intermediate reservoir computed by means of the gas-geoindicators of Marini et al. (2022). Last but not least, we hope that the $CH_4$- and $H_2S$-geoindicators will be used as geochemical monitoring tools, because the current utilization of the sole $CO$-$CO_2$ geo-indicators allows one to monitor the state of the shallow reservoir (0.25-0.45 km depth), whose P, T values are too low to explain the ongoing bradyseism.

**Appendix A. Further considerations on fugacity coefficients**

The importance of considering deviations from the ideal gas behavior in geothermometric calculations is too often overlooked in the geochemical literature. For example, according to Henley and Fischer (2021), "*high temperature volcanic gas mixtures may be considered as ideal because values of the reduced temperature of water (T/Tc where $T_C$ is the critical temperature) in the pressure-temperature range of interest are >1*". In this way, the authors forget that deviations from ideality depend not only on temperature, but also on pressure. It is true that gases approach ideal behavior as temperature increases, but it is also true that gases move away from ideality as pressure increases (Marini, 2007). Therefore, instead of the T/Tc ratio, it is advisable to use the residual volume:

$$\Delta V = V - (RT)/P, \tag{A.1}$$

whose difference from zero measures the deviation from the ideal behavior, or the compressibility factor:

$$Z = (PV)/(RT), \tag{A.2}$$

whose difference from unity is a measure of non-ideality (Marini et al., 2022).

Interestingly, the fugacity coefficients of non-polar gases ($CO_2$, $CH_4$, $CO$, $H_2S$, and $H_2$) increase with increasing P and T, deviating gradually from unity, whereas the fugacity coefficient of $H_2O$ decreases with increasing P and T, departing progressively from one, for all decompression paths considered by Marini et al. (2022). The ensuing implication is that the decimal logarithm of the fugacity coefficient ratios $\varphi_{CO}/\varphi_{CO_2}$, $\varphi_{CH_4}/\varphi_{CO_2}$, and $\varphi_{H_2S}/\varphi_{CO_2}$ are expected to be close to zero and could be disregarded in the geothermometric functions (5), (6), and (7), respectively, without incurring excessive errors, but it is also expected that the decimal logarithm of the fugacity coefficient ratio $\varphi_{H_2}/\varphi_{H_2O}$ deviates significantly from zero and that, therefore, significant errors in the calculated equilibrium temperatures and pressures may occur if it is neglected.

The effects of mutual solubilities on fugacity coefficients can be treated, as proposed by Søreide and Whitson (1992) and Li et al. (2015), considering the binary interaction parameters brine-$CO_2$, brine-$H_2S$, and brine-$CH_4$, which depend on T, P, NaCl molality, and Tc of gas species (Li et al., 2015). Preliminary calculations for the system $CO_2$-$H_2S$-brine (21 wt% NaCl) up to 600°C, 1154 bar show that, with increasing T, P: (i) the $\varphi_{H_2O}$ for the brine system deviates from the $\varphi_{H_2O}$ for the pure water system by $\leq 0.024$ units; (ii) the $\varphi_{CO_2}$ for the brine system increases more steeply than the $\varphi_{CO_2}$ for the pure water system,

departing from it by $\leq 0.658$ units; (iii) the $\varphi_{H_2S}$ for the brine system growths more gently than the $\varphi_{H_2S}$ for the pure water system, departing from it by $\leq 0.339$ units. Due to the lack of the binary interaction parameters brine-$H_2$ and brine-CO, the effects of mutual solubilities on fugacity coefficients were disregarded by Marini et al. (2022).

**Appendix B. Comparison of the CH₄ equilibrium temperature and total fluid pressure computed for different decompression paths**

As shown in Figure B1, the $CH_4$ equilibrium T and P for the saturation decompression path involving a 21 wt% NaCl brine are systematically higher than those computed for the saturation decompression path comprising a 33.5 wt% NaCl brine, with average difference of 27°C and 150 bar, median difference of 25°C and 164 bar, maximum difference of 56 °C and 285 bar, and minimum difference of 7°C and 14 bar.

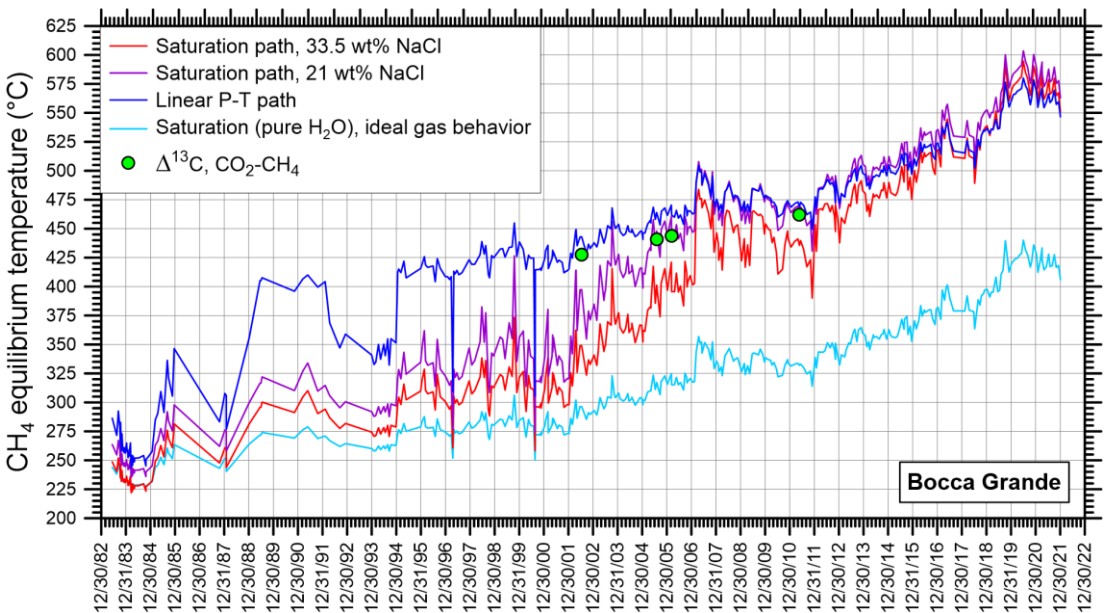

**Figure B1. Chronogram of the CH₄ equilibrium temperature calculated from the data of Cioni and coworkers and Chiodini and coworkers (Buono et al., 2023 and references therein) for different decompression paths of the Bocca Grande fluids. The temperature indicated by the exchange of C isotopes between CO₂ and CH₄ is also shown (data from Caliro et al., 2007 and Fiebig et al., 2013).**

Since the differences between the $CH_4$ equilibrium T and P for the saturation decompression path involving a 21 wt% NaCl brine and those calculated for the linear P-T decompression path are both positive and negative, we have considered the absolute value of these differences, resulting in average values of 32°C and 152 bar, median values of 16°C and 132 bar, maximum values of 109°C and 411 bar, and minimum values of 0.02°C and 0.17 bar. All these figures refer to the whole dataset of 664 data, from both Bocca Grande and Bocca Nuova fumaroles, whereas the chronogram of Figure B1 refer to Bocca Grande data. Since the differences between distinct $CH_4$ equilibrium temperatures and total fluid pressures became <25°C (Figure B1) and <200 bar in 2013-2021, respectively, the computed $CH_4$ equilibrium temperatures and total fluid pressures are almost independent of the considered decompression path in the last years and the error in the estimated overpressure (Fig. 8a) is similar to or even less than short-term fluctuations.

To simplify the discussion, in this paper we have considered the $CH_4$ equilibrium T and P for the saturation decompression path involving a 21 wt% NaCl brine, because of the good agreement between this chemical equilibrium temperature and that indicated by the exchange of C isotopes between $CO_2$ and $CH_4$ for Bocca Grande fumarolic effluents although the available isotopic data (Caliro et al., 2007; Fiebig et al., 2013) are limited in number (as recalled also in section 4.4).

The chronogram of Figure B1 also shows that use of the $CH_4$ geothermometer calibrated for ideal gas behavior and saturation with pure water was acceptable in the 1980s, due to the relatively low values of the $CH_4$ equilibrium temperature, whereas use of the geothermometric functions of Marini et al. (2022), which consider the deviations from the ideal gas behavior, became mandatory in the following years due to the considerable heating of the intermediate reservoir (and even more so for the deep reservoir).

## Appendix C. Assumptions and limitations of gas-geothermometers

### C.1. General assumptions

The main assumptions on which are based the gas-geothermometers specifically calibrated by Marini et al. (2022) for the Solfatara fluids are the same of all water-, gas- or, in a single word, fluid-geothermometers, both chemical and isotopic. They are (Fournier et al. 1974): (1) temperature-dependent reactions occur at depth; (2) all constituents involved in a temperature-dependent reaction are sufficiently abundant (that is, supply is not a limiting factor); (3) fluid-rock equilibration occurs at the reservoir temperature; (4) little or no re-equilibration or change in composition occurs at lower temperatures as the fluid flows from the reservoir to the surface; (5) the fluid coming from deep in the system does not mix with cooler shallow fluids. Here below, other assumptions and limitations are discussed separately for each decompression path modeled by Marini et al. (2022).

## C.2. The saturation decompression paths: assumptions and limitations

Saturation conditions, that is, equilibrium coexistence of liquid water and vapor are usually assumed in all fluid geothermometers, to treat pressure as a function of temperature. In the Solfatara hydrothermal-magmatic systems, the temperature of the melt present at depth $\geq 8$ km is 1120°C and total fluid pressure is 2879 bar (see section 4.7). Since this P, T condition is far beyond the critical point of pure water, i.e., Tc = 373.946 °C and Pc = 220.640 bar (Wagner and Pruß 2002), use of the boiling curve of pure water, that is, the unary system $H_2O$, is not an option to link P and T.

The critical curve of the $H_2O$-$CO_2$ binary system (Tödheide and Franck 1963; Mather and Franck 1992) connects the critical point of pure water to the minimum Tc of 266 °C, where the Pc is 2450 bar and the critical $X_{CO_2}$ is 0.415. Then, the critical curve bends back to higher temperatures, as the pressure continues to rise. Furthermore, the dew curves of $CO_2$-$H_2O$ mixtures of fixed composition exhibit relatively limited changes in temperature and larger variations in pressure (Tödheide and Franck 1963; Mather and Franck 1992). Therefore, also use of the $H_2O$-$CO_2$ binary system is not an option to link P and T.

In the $H_2O$-NaCl binary system, along the critical curve linking the critical point of water to the critical point of NaCl ($\sim$ 3568 °C, $\sim$ 182 bar, $X_{NaCl} = 1$, Anderko and Pitzer, 1993), pressure and $X_{NaCl}$ increase with increasing temperature, at all conditions present in the earth crust (Driesner and Heinrich 2007 and references therein), Therefore, use of the $H_2O$-NaCl binary system is a viable option to link P and T.

A much better option would be to use the $H_2O$-NaCl-$CO_2$ ternary system, but unfortunately it is not feasible because: (1) the studies of the $H_2O$-NaCl-$CO_2$ ternary system, although numerous, have investigated limited P-T-X intervals, as underscored in many studies (e.g., Anovitz et al. 2004; Mao et al. 2015) and (2) there is no reliable EOS for the $H_2O$-NaCl-$CO_2$ ternary system extending up to the P-T-X conditions of interest. These two points are strictly linked because the lack of experimental data obviously complicates, not to say prevents, the derivation of reliable EOS for the $H_2O$-NaCl-$CO_2$ ternary system, in spite of the efforts made in recent years (e.g., Duan et al., 1995, 2008; Sun and Dubessy, 2012; Dubacq et al., 2013; Mao et al., 2015). Among these EOS, only those by Duan et al. (1995) and Mao et al. (2015) extend up to the P-T-X conditions of interest, but the reliability of EOS by Duan et al. (1995) was questioned by other authors (e.g., Schmidt and Bodnar 2000), whereas the predictive thermodynamic model by Mao et al. (2015) is of little interest because it was developed to calculate the molar volume or density of $H_2O$-NaCl-$CO_2$ fluid mixtures and to carry out isochore calculations (pressure-temperature relations) of fluid inclusions. Summing up, new experiments for the $H_2O$-NaCl-$CO_2$ ternary system are needed.

Based on previous discussion on the possible options to link P and T, the $H_2O$-NaCl binary system was finally adopted by Marini et al. (2022) to describe the equilibrium coexistence of the vapor phase with a brine containing either 21 wt% NaCl (data from Tanger and Pitzer 1989) or 33.5 wt% NaCl (data from Driesner and Heinrich 2007), although the $CO_2$ concentration attains values higher than 30 mol % in the Solfatara fumarolic effluents. To be noted that 21 wt% and 33.5 wt% are the uppermost NaCl concentrations considered by Tanger and Pitzer (1989) and Driesner and Heinrich (2007), respectively.

Actually, based on the early experimental data of Sourirajan and Kennedy (1962) for the $H_2O$-NaCl binary system, Giggenbach (1987) adopted a simple relation linking $H_2O$ fugacity, $f_{H_2O}$, to the inverse of the absolute temperature to describe the equilibrium coexistence of a NaCl brine with a vapor phase up to a temperature of 600 °C and a $f_{H_2O}$ of 654 bar. The possible shifts of the brine-vapor co-occurrence condition to lower temperatures or higher $f_{H_2O}$ values due to the presence of $CO_2$ were considered negligible for a vapor phase containing up to 20 mol % $CO_2$ at 6 wt % NaCl and disregarded by Giggenbach (1987).

The other side of the coin is that the simplified approach of Marini et al. (2022) based on the $H_2O$-NaCl binary system, in principle, cannot be applied in the range of conditions of liquid-vapor immiscibility and cannot be used to evaluate mutual partitioning of NaCl and $CO_2$ between fluid phases. However, in practice, both aspects are poorly known.

The saturation condition can be applied up to the Tc and Pc of the considered $H_2O$-NaCl binary system, that is: (i) up to 600°C, 923 bar according to Tanger and Pitzer (1989) or 592°C, 892 bar according to Driesner and Heinrich (2007) for the $H_2O$-NaCl binary system involving a 21 wt% NaCl brine, and (ii) up to 1000°C, 2162 bar according to Driesner and Heinrich (2007) for the $H_2O$-NaCl binary system including a 33.5 wt% NaCl brine. The difference of 8°C in the Tc and 31 bar in the Pc, for the $H_2O$-NaCl binary system comprising a 21 wt% NaCl brine, suggests that both the critical properties and, consequently, the limits of applicability of saturation conditions are affected by some uncertainties.

The $CH_4$ equilibrium temperatures computed using the equation for 21 wt% NaCl are greater than Tc = 600°C (up to 645°C) for 12 of the available 664 samples of the Solfatara fumarolic fluids, whereas the corresponding $CH_4$ equilibrium temperature calculated utilizing the equation for 33.5 wt% NaCl are lower by 7 to 14°C and well below the Tc. Similarly, the $CH_4$ equilibrium $P_{H_2O}$ computed using the equation for 21 wt% NaCl are greater than Pc = 923 bar (up to 1088 bar) for the same 12 samples, whereas the corresponding $CH_4$ equilibrium $P_{H_2O}$ calculated utilizing the equation for 33.5 wt% NaCl are lower by 85 to 108 bar and well below the Pc. Therefore, the question is not if these 12 $CH_4$ equilibrium temperatures > 600°C and $P_{H_2O}$ > 923 bar are meaningful or not, but what is their uncertainty. The uncertainty on the temperature is certainly much less than the difference between the two $CH_4$ equilibrium temperatures, which varies between a minimum of 7°C and a maximum of 56°C, with an average of 27°C and a median of 25°C. In a similar way, the uncertainty on the $P_{H_2O}$ is surely much lower than the difference between the two $CH_4$ equilibrium $P_{H_2O}$, which ranges between a minimum of 12 bar and a maximum of 216 bar, with an average of 117 bar and a median of 125 bar. In fact, it is possible to choose a suitable NaCl concentration, between 21 and 33.5 wt%, whose Tc is > 645°C by a few °C and whose Pc is > 1088 bar by a few bar. A possible solution is the $H_2O$-NaCl binary system at 24.5 wt% NaCl for which Tc = 653°C and Pc = 1090 bar.

**C.3. The linear P-T decompression path: assumptions and limitations**

Another decompression path considered by Marini et al. (2022) is the so-called "linear P-T expansion path", which was suggested by Stevenson (1993). It assumes that both total fluid pressure and temperature increase linearly from the critical point of water to the T, P of the melt present at depth ≥ 8 km, that is 1120°C, 2879 bar (see section 4.7). Below the critical

point of water, $P_{H_2O}$ and temperature are assumed to be constrained by the saturation condition for pure water, while above the critical point of pure water total fluid pressure is a function of both temperature and $X_{CO_2}$, according to the relation ($P_{tot}$ in bar; T in °C):

$$P_{tot} = (-1.07619 \cdot X_{CO_2}^4 + 0.09285714 \cdot X_{CO_2}^3 - 0.3716667 \cdot X_{CO_2}^2 - 0.2898571 \cdot X_{CO_2} + 3.563117) \cdot T$$

$$+ (1152.829 \cdot X_{CO_2}^4 - 57.5119 \cdot X_{CO_2}^3 + 403.51 \cdot X_{CO_2}^2 + 325.7244 \cdot X_{CO_2} - 1111.676). \qquad (C.1)$$

This assumption, although extremely crude, allows to "approach the magma" in a way alternative to the saturation decompression paths, although also the linear P-T expansion path may be controlled by saturation conditions in a system comprising a high-salinity brine. Due to its crude derivation, the limitations and uncertainties of the linear P-T expansion path cannot be estimated.

### C.4. The isenthalpic decompression path: assumptions and limitations

Although the $H_2O$-$CO_2$ binary system is not an option to link P and T (see section B.3), it was used by Marini et al. (2022) to model the isenthalpic decompression path of Solfatara fluids, making a step forward with respect to previous studies (e.g., Cioni et al., 1984; Moretti et al., 2018, 2020), in which the unary system $H_2O$ was adopted.

First, the enthalpy of the considered gas mixture was obtained at atmospheric pressure and outlet temperature. Second, the enthalpy was kept constant and temperature, molar volume, and entropy were computed at increasing pressure values using the GERG-2008 (Kunz and Wagner, 2012) thermodynamic module of the CONVAL® 11 software package, up to maximum pressures and temperatures of: (i) 2750 bar and 609.8 °C for $X_{CO_2}$ of 0.05, (ii) 2500 bar and 575.8 °C for $X_{CO_2}$ of 0.10, (iii) 2000 bar and 514.1 °C for $X_{CO_2}$ of 0.20, (iv) 1750 bar and 460.2 °C for $X_{CO_2}$ of 0.30, and (v) 1500 bar and 414.1 °C for $X_{CO_2}$ of 0.40, because temperature does not change appreciably with pressure, above these upper pressure values. The applicability of the isenthalpic expansion model is further limited by the nonmonotonic behavior of the sum of log-ratios, β, of reaction (2), see Eqn. (6). Due to these limitations, the isenthalpic path was only used to calibrate the CO geoindicators by Marini et al. (2022).

### Appendix D

**Table D1. Average and standard deviations values of the outlet temperature, and CO-, CH₄-, and H₂S equilibrium temperatures for 24 selected time intervals, from June 1983 to January 2022, for the fumarolic samples collected at Bocca Grande by Cioni and coworkers and Chiodini and coworkers (data from Buono et al., 2023 and references therein).**

| Time interval | Outlet T (°C) | | CO equil. T (°C) | | CH₄ equil. T (°C) | | H₂S equil. T (°C) | |
|---|---|---|---|---|---|---|---|---|
| | average | std.dev. | average | std.dev. | average | std.dev. | average | std.dev. |
| 1983 Jun - 1984 Jul | 156.9 | 0.6 | 227 | 13 | 246 | 8 | 667 | 24 |
| 1984 Sep - 1987 Apr | 158.4 | 2.5 | 220 | 11 | 264 | 21 | 667 | 35 |

| Time interval | Outlet T (°C) | | CO equil. T (°C) | | CH$_4$ equil. T (°C) | | H$_2$S equil. T (°C) | |
|---|---|---|---|---|---|---|---|---|
| | average | std.dev. | average | std.dev. | average | std.dev. | average | std.dev. |
| 1987 Oct - 1989 Jul | 162.0 | 0.0 | 233 | 10 | 291 | 26 | 734 | 12 |
| 1990 Nov - 1992 Dec | 162.0 | 0.0 | 228 | 10 | 312 | 13 | 747 | 13 |
| 1993 Dec - 1995 Apr | 162.2 | 0.6 | 208 | 21 | 302 | 17 | 709 | 9 |
| 1995 May - 1997 Dec | 162.0 | 0.0 | 219 | 13 | 328 | 18 | 732 | 18 |
| 1998 Jan - 1999 Jun | 161.8 | 0.9 | 224 | 5 | 345 | 21 | 781 | 21 |
| 1999 Aug - 2001 Mar | 157.4 | 4.0 | 222 | 8 | 339 | 33 | 750 | 25 |
| 2001 Mar - 2003 Feb | 159.8 | 1.2 | 219 | 7 | 356 | 29 | 759 | 14 |
| 2003 Mar - 2004 Dec | 162.0 | 1.2 | 212 | 6 | 413 | 15 | 787 | 12 |
| 2005 Jan - 2005 Sep | 159.9 | 2.2 | 209 | 3 | 429 | 17 | 803 | 14 |
| 2005 Oct - 2007 Feb | 160.1 | 3.4 | 209 | 4 | 448 | 7 | 805 | 9 |
| 2007 Mar - 2007 Jul | 161.0 | 1.4 | 208 | 6 | 498 | 8 | 815 | 18 |
| 2007 Sep - 2008 Oct | 162.4 | 1.1 | 212 | 6 | 475 | 13 | 819 | 17 |
| 2008 Nov - 2010 Sep | 163.0 | 0.9 | 215 | 7 | 470 | 12 | 813 | 11 |
| 2010 Oct - 2012 Jun | 163.1 | 1.4 | 218 | 6 | 468 | 14 | 830 | 15 |
| 2012 Jul - 2013 Jun | 163.2 | 1.1 | 220 | 5 | 486 | 10 | 841 | 12 |
| 2013 Jul - 2014 Aug | 163.7 | 0.9 | 217 | 3 | 502 | 7 | 866 | 11 |
| 2014 Sep -2015 Dec | 162.6 | 0.7 | 221 | 3 | 510 | 8 | 869 | 13 |
| 2016 Jan - 2017 Feb | 163.6 | 0.6 | 220 | 3 | 527 | 8 | 884 | 16 |
| 2017 Mar - 2018 Dec | 163.3 | 0.7 | 215 | 3 | 540 | 12 | 915 | 23 |
| 2019 Jan - 2019 Dec | 161.9 | 0.9 | 218 | 3 | 563 | 16 | 973 | 29 |
| 2020 Jan - 2020 Dec | 161.9 | 1.2 | 219 | 3 | 589 | 8 | 1010 | 14 |
| 2021 Jan - 2022 Jan | 162.4 | 1.2 | 215 | 3 | 580 | 8 | 975 | 23 |

**Table D2. Average and standard deviations values of the CO-, CH$_4$-, and H$_2$S equilibrium pressures for 24 selected time intervals, from June 1983 to January 2022, for the fumarolic samples collected at Bocca Grande by Cioni and coworkers and Chiodini and coworkers (data from Buono et al., 2023 and references therein).**

| Time interval | CO equil. P (bar) | | CH$_4$ equil. P (bar) | | H$_2$S equil. P (bar) | |
|---|---|---|---|---|---|---|
| | average | std.dev. | average | std.dev. | average | std.dev. |
| 1983 Jun - 1984 Jul | 27.9 | 6.8 | 38.5 | 5.6 | 1308 | 102 |
| 1984 Sep - 1987 Apr | 25.7 | 5.2 | 56.7 | 19.8 | 1401 | 184 |
| 1987 Oct - 1989 Jul | 31.4 | 5.6 | 82.8 | 30.0 | 1642 | 58 |
| 1990 Nov - 1992 Dec | 29.0 | 4.8 | 108 | 21 | 1704 | 81 |
| 1993 Dec - 1995 Apr | 20.3 | 6.5 | 92.0 | 24.2 | 1491 | 36 |
| 1995 May - 1997 Dec | 24.4 | 5.6 | 128 | 24 | 1619 | 73 |
| 1998 Jan - 1999 Jun | 25.8 | 2.4 | 151 | 26 | 1778 | 82 |
| 1999 Aug - 2001 Mar | 24.7 | 3.6 | 142 | 58 | 1645 | 106 |
| 2001 Mar - 2003 Feb | 23.8 | 3.2 | 167 | 52 | 1696 | 64 |
| 2003 Mar - 2004 Dec | 21.1 | 2.2 | 299 | 66 | 1845 | 57 |
| 2005 Jan - 2005 Sep | 20.0 | 1.0 | 370 | 75 | 1918 | 69 |
| 2005 Oct - 2007 Feb | 20.4 | 1.5 | 461 | 34 | 1951 | 42 |

| Time interval | CO equil. P (bar) | | CH$_4$ equil. P (bar) | | H$_2$S equil. P (bar) | |
|---|---|---|---|---|---|---|
| | average | std.dev. | average | std.dev. | average | std.dev. |
| 2007 Mar - 2007 Jul | 20.5 | 2.4 | 708 | 39 | 2042 | 71 |
| 2007 Sep - 2008 Oct | 22.5 | 2.5 | 610 | 67 | 2094 | 76 |
| 2008 Nov - 2010 Sep | 24.1 | 3.1 | 586 | 59 | 2073 | 57 |
| 2010 Oct - 2012 Jun | 25.2 | 2.7 | 582 | 67 | 2157 | 67 |
| 2012 Jul - 2013 Jun | 26.7 | 2.3 | 675 | 50 | 2224 | 59 |
| 2013 Jul - 2014 Aug | 25.5 | 1.2 | 765 | 41 | 2364 | 57 |
| 2014 Sep -2015 Dec | 27.1 | 1.2 | 797 | 44 | 2350 | 70 |
| 2016 Jan - 2017 Feb | 27.0 | 1.5 | 890 | 40 | 2438 | 74 |
| 2017 Mar - 2018 Dec | 25.5 | 1.3 | 982 | 65 | 2643 | 114 |
| 2019 Jan - 2019 Dec | 27.1 | 1.3 | 1103 | 98 | 2893 | 160 |
| 2020 Jan - 2020 Dec | 27.7 | 1.5 | 1226 | 46 | 3039 | 73 |
| 2021 Jan - 2022 Jan | 25.6 | 1.3 | 1186 | 52 | 2901 | 120 |

**Appendix E. Considerations on the residence time of fluids in each reservoir**

In principle, assuming steady-state, our conceptual model can be used as reference to compute the residence time spent by the

fluids in each reservoir, specifying the total volume (rocks + fluids), the effective porosity, $\eta$, and the T, P conditions of each

reservoir, as well as the flow of fluids through the system. For example, in 2020, considering a CO$_2$ flow of 5000 ton/day

(Chiodini et al. 2021), a $X_{H_2O}/X_{CO_2}$ ratio of fumarolic fluids of 2.57 (Buono et al. 2023) and the equilibrium T, P reported in

Tables C1 and C2, respectively, the total gas flow through the shallow reservoir was 0.000599 km$^3$/day, at the T, P conditions

of this reservoir. Based on this value and the total volume of the shallow reservoir, ca. 0.2 km$^3$ (with an area of 1 km$^2$ and a

thickness of 0.2 km, see section 4.2), it turns out that the residence time of fluids could be either 33 days for $\eta = 0.1$ or 3.3

days for $\eta = 0.01$ or 0.33 days for $\eta = 0.001$. Thus, the calculation of the residence time of fluids is challenging, not to say

impossible, even for the shallow reservoir since its effective porosity is poorly constrained, not to say unknown. The situation

is worse for the intermediate and the deep reservoirs because the volume of these two reservoirs contributing to the fluid

discharge is also unknown, in addition to the effective porosity.

Considering the shallow reservoir in November 2008 – September 2010, when the CO$_2$ flow was 1000 ton/day (Chiodini et al.

2021) and the $X_{H_2O}/X_{CO_2}$ ratio of fumarolic was 3.35 (Buono et al. 2023), the same calculations show that the residence time

of fluids could be either 120 days for $\eta = 0.1$ or 12 days for $\eta = 0.01$ or 1.2 days for $\eta = 0.001$. Thus, it could be concluded

that the five-fold increase in gas flow from 2008-2010 to 2020 caused a nearly 3.6-fold reduction in residence time, but it

cannot be excluded that different factors (e.g., ground uplift, seismicity, mineral dissolution/precipitation) caused a change in

the effective porosity of the shallow aquifer, invalidating this conclusion. All in all, owing to the uncertainties stated above, it

is not possible to made reliable evaluations of the residence time of fluids, even for the shallow reservoir.

## Appendix F. Considerations on the overpressure at the top of the deep reservoir

The overpressure at the top of the deep reservoir, calculated on the basis of the saturation decompression path involving a 21 wt% NaCl brine, attained the maximum value of 1700 bar in 2020. This overpressure value is unacceptable if the medium were in the elastic regime since the strength of crustal rocks is less than a few hundred bar (e.g., Brace and Kohlstedt, 1980). However, the rocks hosting the deep reservoir and the underlying rocks capping the magma chamber are expected to have a viscoelastic behavior under the conditions present therein. To understand if this overpressure value is acceptable or not in the viscoelastic regime, we refer to the works of Bonafede and coworkers (e.g., Bonafede et al., 1986; Trasatti et al., 2005; Bonafede and Ferrari, 2009) who modeled, adopting different approaches, the two ground deformation episodes occurred in the Campi Flegrei caldera in 1968–72 and 1982–84, resulting in 1.7 m and 1.8 m maximum uplift, respectively. In particular, Bonafede and Ferrari (2009) generalized the popular model of Mogi (1958) to viscoelastic rheology and showed that the overpressure needed within the source to reproduce the ground deformation is strongly dependent on the relaxation time, assuming a conceptual model in which magma is supplied at constant rate to the source. If the relaxation time is short (e.g. < 1 month), the overpressure attains an asymptotic value lower than a few hundred bar after 2 years whereas, for a relaxation time close to 3 months, the overpressure reaches an asymptotic value of 1700 bar after 2 years.

Although other conceptual models, perhaps even more realistic, can be adopted to explain ground deformation in the Campi Flegrei caldera, the model recalled here is reasonable because magma supply did occur during the bradyseismic crisis of 1982–84 (Caliro et al., 2014) and ground deformation was evidently controlled by the deep magma source. In fact, both the intermediate reservoir at 2.7-4.0 km depth and the deep reservoir at 6.5-7.5 km depth were not pressurized at that time (Figure 7) and the deeply-extending faulted-fractured zone connecting the three reservoirs had not formed yet. Today, the situation, as outlined in our conceptual model, is completely different. Therefore, we hope that the $CH_4$- and $H_2S$-geoindicators will be adopted as geochemical monitoring tools, because the current use of the $CO$-$CO_2$ geoindicators allows to monitor only the shallow reservoir (0.25-0.45 km depth), whose P,T conditions are too low to explain the ongoing bradyseism.

## Data availability

All raw data and results of geothermometers and geobarometers developed by Marini et al. (2022) are reported in Table S1.

## Author contribution

LM wrote the geochemical aspects of the manuscript draft; CP wrote the geological aspects of the manuscript draft; ML reviewed and edited the manuscript.

## Competing interests

The authors declare that they have no conflict of interest.

## Acknowledgements

Our manuscript was sent to three scientific journals before the submission to Solid Earth. The Editors of the first and second journals are warmly thanked for quickly reaching the decision that our manuscript was of no interest to their journals, in contrast with the Editor of the third journal who was unable to reach any conclusions in seven months, thus prompting us to

withdraw our manuscript and submit it to Solid Earth. Sincere thanks are due to the Executive Editor of Solid Earth, Prof. Andrea Di Muro, and the Topic Editor of Solid Earth, Prof. Massimo Coltorti, for the editorial assistance. We also want to thank two anonymous reviewers, for their comments and criticisms, even very severe, which anyhow helped us to improve our manuscript.

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
