# Peer review of "Time changes during the last 40 years in the Solfatara magmatichydrothermal system (Campi Flegrei, Italy): new conceptual model and future scenarios"

_EGUsphere, 2024_

## Referee Comment (RC2)

Thank you for the reply, but I must admit that I feel that the contents of my previous review rather got confirmed than refuted. In the following, after a personal introductory remark, I will provide a second, in-depth review of several aspects of the study that will unambiguously demonstrate that key calculations are deficient with the consequence that key parts of the results on pressure lack scientific validity and that likely the temperatures as well are questionable. I will conclude with an incomplete list of answers to the authors rebuttal on points where I find that it may be beneficial for clarification.

**Introductory remark**

Let me start with a personal remark on the reviewing effort. A significant part of the authors' rebuttal is in essence "we explained this in the book Marini et al., 2022". And that was my very point in my previous commentary: a lot of information that is essential to allow a competent, science-focused review (or simply: a read) of the manuscript is not in there. It was submitted as a *research* article and as such should contain sufficient information about the concept, methods and assumptions to allow the reviewer an informed preparation of checking for, quoting the authors, "scientific validity and ... errors of any kind". I estimate that a few manuscript pages on Methods with carefully chosen and explained content would have guided the reviewer (or just reader) well enough to decide whether and what to follow up or not by reading the book or parts of it and they would have much increased the value of the manuscript. The manuscript condensed ca. 150 *pages* in the book to ca. 15 *lines* for the "Methods" and made not a single statement on important underlying concepts, assumptions, limitations, etc. This would have been even more important as the authors claim to have introduced novel tools.

In the version that I reviewed I was essentially forced to explore the book in quite some depth to just get the basics of what was performed in the study, not even to then follow up on the science done. I take peer reviewing serious and I am happy to spend two or three full working days for an in-depth review of an interesting manuscript (as I did then). But, here, this was not leading me anywhere close to understanding the full line of arguments, approaches, assumptions etc. for this study. The book does not make it easy for the reader to detect that information, which is often hidden in very long technical discussions that frequently get as detailed as instructing the reader about how to copy values around in Excel spread sheets (I am not criticizing that this is documented, it is great it is, but it just makes the book a very hard read). For me this was the first time in more than three decades of reviewing that I encountered a paper that expects reading a book in-depth before it can be understood. In essence, this meant reviewing the (published!) book, not the manuscript. I hope the authors may understand that this felt like them making my reviewer job as hard as possible and that I did not feel positive about the extra, in my opinion unnecessary, time effort.

**Additional Review**

Now, although my duty was to review the manuscript and not the book, I felt challenged by the style and content of parts of the rebuttal (some more words on that in the answer list in the third part of this reply) and dived rather deeply into some parts of the "thermometry" and "barometry", which I already felt look suspicious when browsing the book before for understanding what was done in the study. After that new exercise (another two full working days including writing this reply), I make my recommendation even stronger:

***The manuscript should be rejected because (at least) the pressure calculations for the middle and deep "reservoirs" (and likely the associated thermometry as well, at least for the deep and likely also for the***

*middle "reservoir") are based on a fundamentally invalid assumption and are, therefore, scientifically invalid in their current form. Even if the calculations were formally valid, the pressures would be based on an arbitrary, assumed (without constraints from data or theory) temperature-pressure path, i.e., again scientifically not constrained. As these invalid results form the very core of the study's main contribution – the "revised conceptual model" and the important conclusions resulting from it – the manuscript cannot be accepted.*

Notabene (after the experience how the previous review was perceived): I do acknowledge that a lot of honest effort and care were put into the underlying calculations in the book; nevertheless, I am convinced that a key problem escaped the authors' attention, which, unfortunately, renders the "saturation decompression path" invalid, no matter how much effort was put into the calculation concept and its execution and no matter how exciting the results might look to the authors. That I use frank words should not be confused with disrespect but as a way to keep the discussion clear about the problem and keep it focused on the science; and not be shifted again to a scientifically meaningless competition on how experienced and qualified either reviewer or authors might be in different sub-areas of magmatic-hydrothermal systems research or in applying the respective community's wording sufficiently accurately to be accepted (on that: I regret that the omission of "thermometry" in "new gas geochemical *thermometry* data" escaped my attention when proof-reading my review).

**Now to the core problem**. The calculations of gas equilibration temperatures involve thermodynamics-based corrections for strong pressure effects. The pressures and resulting temperatures rely on an *arbitrary, assumed temperature-pressure path* along which fluids would have to flow on their way from the magmatic source to the fumaroles 8 km above where the samples were taken. The study tries to pinpoint where on that generic path individual parts of the actual Solfatara system are located.

According to the book, this assumed "saturation decompression path" is an arbitrary selection. It takes the vapor saturation pressure for a constant composition liquid (21 wt.%) in the pure $H_2O$-NaCl binary and adding a simplistic pressure correction for the presence of $CO_2$ in the vapor phase (as a side remark I wonder how one could call that "in *equilibrium* with a brine"; but see discussion of $CO_2$ solubility below). I like to emphasize again that this is an *assumed, arbitrary* path and not supported by any data; that "Giggenbach did that as well in 1987" is not a good geology- or physics-based justification. Pressures for the "reservoirs" are then "computed" by taking the computed gas equilibration temperatures and locating them on that path (I assume this was an iterative procedure).

In a nutshell, the key problems of that approach are (more, illustrated detail then further below):

- A vapor-saturated 21 wt% NaCl brine that the calculations are based on simply doesn't exist above ca. 590 °C (the critical temperature for that composition in the binary $H_2O$-NaCl system[1]), so the calculation is meaningless for all inferred temperatures for the deep "reservoir" and for the hottest of the middle one.
- Instead, above that temperature, an $H_2O$-NaCl fluid on the twophase liquid-vapor surface in that system is a vapor itself.
- As a major consequence, and as the critical line for 21wt% NaCl (with respect to water) in the ternary $H_2O$-NaCl-$CO_2$ system is nearly isothermal at those 590 °C: if $CO_2$ is present, the fluid at such high temperature and pressure conditions would be a homogeneous, "supercritical", dense vapor-
* * *
[1] btw., Giggenbach (1987) also stated that the validity is limited to ca. 600°C

- like fluid for which no unique P-T coordinates can be constrained without serious further assumptions.
- For this dense vapor-like fluid, due to the presence of significant NaCl and even if the pressures were correct, the fugacity coefficients calculated by the authors via a Peng-Robinson equation of state for a system without NaCl likely are in error (and, as far as I recall, standard P-R equations of state are not very well suited to include electrolytes) and so would be the gas thermometry.
- These points render the temperature-pressure calculations for the deep "reservoir" in their present form invalid. How wrong they are in terms of numbers cannot be estimated in a simple way.
- Moreover, the approximation of a carbonic vapor phase pressure in "equilibrium" with a vapor-saturated $H_2O$-NaCl liquid is only acceptable if the mutual solubilities of $CO_2$ in the aqueous solution and of water in the carbonic vapor phase are small such that the saturation condition spreads only over a small pressure interval and remains close to the saturation line in the aqueous solution binary.

  This holds true for typical geothermal systems at less than ca. 300ish °C and has, therefore, indeed be applied to such systems. As soon as the mutual solubilities are non-negligible (starting at T > ca. 300ish °C, some people might claim 350ish °C, it really doesn't matter much) this is not possible anymore as "saturation pressure" of the liquid is not uniquely defined and phase compositions will change with phase proportions for a given bulk composition.

  Rather, the liquid saturation pressure will now also be a function of the additional $CO_2$ content of liquid, i.e., there is a whole saturation surface over very wide ranges of pressure and for temperatures up to the near-isothermal (the above 590 °C) critical curve of the pseudo-binary "21 wt% aqueous NaCl solution + $CO_2$" instead of a single saturation line. Side remark: so much about the authors' statement "there is no need for an *adequate solubility model for the gases in such a brine*."
- Given the temperature range mentioned in the previous point, also both the temperatures and pressures in the middle "reservoir" are likely highly questionable.

Now this in some more detail, illustrated by published figures of the respective phase diagrams.

Let's start with the issue of 21% vapor-saturated brine with the diagram by Driesner and Heinrich (GCA, 2007). I have highlighted the 20 wt.% curve (let's stay with that as a convenient proxy for the author's choice for rest of the discussion) in yellow and one can see how it intersects the critical curve at ca. 590 °C, i.e., at lower temperatures it is on the liquid side of the twophase vapor+liquid surface, at higher temperatures on the vapor side. So, there is no such thing as a 20 wt% saturated liquid at temperatures higher than this. One might now argue "ok, let's simply take a higher salinity such as 33.5% used elsewhere in the manuscript and the problem is gone".

[Figure]

Well, before we come to that point let's first look at a relevant P-T projection of parts of the $H_2O$-NaCl-$CO_2$ phase diagram, taken from Schmidt and Bodnar (GCA, 2000). The most relevant feature in that diagram is the near-vertical line labeled "20 wt% NaCl". This is the critical line for the pseudo-binary "20 wt% NaCl/80wt% water + $CO_2$". At temperatures higher than that curve any fluid in the ternary $H_2O$-NaCl-$CO_2$ system that has a 20:80 wt% ratio of NaCl and water and has some $CO_2$ content will exist as a single-phase, "supercritical" fluid (highlighted yellow, extent to bottom not well known and not well

understood). Red added curve is the approximate location of 20 wt% NaCl on V+L surface (solid curve: liquid, dashed curve: vapor), added blue curve is threephase vapor+liquid+halite, both in the $H_2O$-NaCl binary. In that binary, below the dashed red curve, there would be liquid with >20% NaCl in coexistence with a vapor <20% NaCl but no fluid phase with 20wt% can exist there. What happens below the red curve in the ternary is not well known, in particular if and where it comes to halite saturation as that is now a divariant surface in T-P rather than a univariant curve and may actually start at much higher pressures than in the binary (see, e.g., Anovitz et al., GCA, 2004).

*I think it is obvious from the above that the authors' pressure calculation method has no foundation at temperatures above ca. 590 °C and that, therefore,* **the pressure calculations for the deep reservoir are simply invalid***. An additional, very important point now is that, at the higher temperature conditions, appreciable amounts of NaCl would be present in the vapor phase (in different concentrations for different T-P) rendering the used Peng-Robinson e.o.s. version non-applicable. For example, already at 500°C and 500 bar, Anovitz et al. (GCA, 2004) found up to several mole percent NaCl in the vapor and this should be expected to increase with higher temperature for the simple thermodynamic requirement that it needs to converge with the aqueous liquid's concentration towards the upper critical end curve.* ***Therefore, even the thermometry is seriously questionable for temperatures of the late part of the time series for the middle reservoir and for all temperatures of the deep reservoir.***

The full implications of the diagram above are non-trivial to comprehend as it is a 2D projection of elements of a 4D diagram. For me, the most intuitive and plausible image emerges when taking the $H_2O$-$CO_2$ phase diagram in 3D as a reference, e.g., from Diamond (Lithos, 2001). What is labelled there "Upper critical curve of binary" is the "0 wt.% NaCl curve" in the previous diagram, i.e., this diagram could roughly be imagined as an extension of the previous diagram to the left with $X_{CO2}$ as an axis going into the diagram. The "Upper critical curve of binary" limits the chimney-like carbonic-aqueous immiscibility region to high temperatures. We could imagine this as a crude guide for how the "immiscibility chimney" expands and shifts

to higher temperature when moving to the pseudo-binary "20 wt% NaCl/80wt% water + $CO_2$" (most elements in the previous diagram): also there, it'd be a "chimney" with significant mutual solubility on the steep saturation surfaces (again: not lines) on the aqueous and carbonic sides and limited to high temperatures by the "20 wt% NaCl" line in the second diagram.

Now, this also explains why choosing saturation of a 33.5% "brine" would not do the job: although the upper critical line of that pseudo-binary would be at higher temperatures (about 1000°C) the mutual solubility would still eliminate the choice of a single saturation pressure and the saturation surfaces would still be steep, i.e., saturation pressure ranges are very large. Even worse (see also Anovitz et al., 2004, their Figure 4): at a given temperature and pressure varying $X_{CO2}$ of the saturated vapor will make the NaCl content of the saturated liquid change significantly. ***So, the authors barometry approach also breaks down at temperatures relevant for the middle reservoir.***

Unfortunately, for the lack of more experimental data, we don't know to how low temperatures this may be the case but I think it would be good scientific practice to cautiously assume that Anovitz' 500°C is not a fortuitous hit of the lower temperature limit of the problem. The community has not yet fully explored and understood all the phase relations in the $H_2O$-NaCl-$CO_2$ ternary, namely on the carbonic side or at the low pressure end. Yet, with the above (which necessarily was highly simplified and incomplete) I think we have understood enough that the negative assessment about the (in)validity of the authors' approach is robust.

In terms of the manuscript's main results this means graphically:

[Figure]

To add on this: the authors could have suspected a serious problem already from their pressure diagram. Fluid pressures as high as 2.4 times lithostatic (or, in absolute values overpressured by up to 1800 bar) and fluid pressure gradients similarly excessive, in particularly in the hot, ductile regime, would be considered unrealistic by many (if not most) people dealing with such problems for reasons laid out, e.g., by Cox (Geofluids, 2010) or discussed in the Weis et al. (2012) paper (or follow-ups on that) that the authors considered not relevant in their rebuttal. In essence, (ductile) rocks cannot sustain such high fluid overpressure, the values are very unrealistic. Since a lot of the implications of the model hinge on that, the author may want to acquaint themselves more with rock failure related to fluid pressure (the Cox paper is a good introduction).

Now, to be "constructive" after all this: The phase relations discussed above also propose a possible way how the "middle reservoir" can indeed be the source of bradyseism. If the fluid released at depth were a "supercritical" $H_2O$-NaCl-$CO_2$ mixture then upon decompression it may hit the "immiscibility chimney" displayed in the $CO_2$-related phase diagrams above. For a suitable parameter combination this should result in sufficient pressurization.

Similarly, the overpressure problem goes back to the assumed temperature-pressure "saturation decompression path" and the fact that it is simply arbitrary, not founded on any data or defendable theory, and leads to too high pressures. Given the above discussions on the phase diagram it seems likely to me that a plausible decompression path can be constructed that implies lower, ore realistic pressures at depth. Then, "lithostatic plus a defendable overpressure" in the deep "reservoir" would be a good starting point for revising the decompression path (from the source upward rather than vice versa) of the "revised conceptual model".

*Up to here, I consider this reply as essential information for the editor. I hope the explanations are not "too nerdy" to be evaluated for the decision process (if they are, please request an explanation in simpler words) and I hope that also the authors can accept my criticism given their own statement "every author should be allowed to publish her/his results, provided that they have scientific validity and are not affected by errors of any kind". I think I clearly showed above that the study contains serious problems of both kinds.*

In the following I take the freedom to share some reflections with the authors as on what other aspects I think they may want to consider and/or re-think when re-visiting their model. This will also reply to some selected parts of their rebuttal. I will not answer to all points as I think that the discussion has already reached a point of getting lost in unnecessary back-and-forth. I don't claim to own the truth here but I would expect my thoughts to be considered a bit more seriously (1) as possibly valuable when revisiting the model and (2) for what to pay attention to beyond the gas chemistry point of view.

*Various points*
* * *
- As an add-on to the discussion above, I would like to suggest to the authors to also think a "decompression path" indeed from the deep source upward and not top down from the surface (which is rather a "pressurization path") although the gas data were collected there. Phase changes will modify $CO_2$-$H_2O$ ratios upon ascent and the degree and cause of this modification cannot be quantified in the top down way; e.g., for your shallow isenthalpic paths, for example, when going down you hit the $H_2O$-$CO_2$ saturation on the (carbonic) vapor side (on a rather flat surface that forms the "chimney's" bottom; such vapor can be equally well result from temporary saturation and condensing out some minor liquid (due to the "bulge" in the vapor enthalpy curve as you discussed) or from the (boiling-like) exsolution of a mass-wise minor vapor phase from an $CO_2$-poor but mass-wise dominant aqueous liquid. What happens below that saturation depth is, therefore, ambiguous and cannot be constrained from surface data as presented. Also, for this problem, I missed a clear statement of what your assumptions were for the top down approach regarding the fluid phase

evolution with depth. Such a statement would have allowed readers/reviewers to test your hypothesis; leaving it out – or not at least discussing this / not formulating a hypothesis – is not good practice.

- Again on top down vs. bottom up: that was also the main reason for recommending Einaudi et al. and others for sulfur or Weis et al. for thermo-hydrology. I did not assume that the former were up to date with respect to the latest in volcanic gas chemistry but these papers think the chemical (or fluid flow, resepctively) process from the source to the surface and this clearly adds value. Namely, Einaudi et al. highlight that the fluid passes through different redox and $f_{S2}$ conditions along its path as exemplified by successive mineral assemblages observed, which will, among others, also alter sulfur speciation and fugacity/concentration (and therefore the values of your $H_2S$ thermometer) on the way up. To me this looks geologically and geochemically much more logically and advanced than the assumption of a single mineral-fluid reaction fixing it right away at 7 or so km depth. Whether the Weis model was inspired by porphyry deposits doesn't play a role; to me, it models a generic magmatic fluid release process that was then interpreted by those authors for its relevance for pophyry-Cu deposits.
- A bit more about the reactive path of sulfur: for the deep parts, the main window of action for $SO_2$ + $H_2O$ reacting are believed to be below 500ish °C or so (gas redox buffer followed by disproportionation along cooling path, if I remember correctly). This is at lower temperatures than your $H_2S$ thermometer equilibration and should therefore be assessed for a possible impact on $H_2S$ concentrations on the way to the surface.
- Fournier vs. Weis et al.: the Weis model replaced the Fournier concept in that it explains the lithostatic to hydrostatic transition as a natural consequence of degassing magmatic fluids having the dual role of heating the rock overlying the magma to ductile temperatures and impermeable behaviour and, in turn, of transiently breaking those heated rocks due to pressure build-up to allow the temporary release of fluids; this magmatic-hydrothermal domain than naturally transitions into a classical geothermal system further up; there is no need for self-sealing by silica, just heat + fluids do the job already, matching many features of natural systems (fossil or active). Let Occam's razor do its job here.
- Lupi/Weis: well, that was quite a cheap trick referring to what I did not point at (the trigger mechanism, which indeed can be questioned) and then trying to make me look ridiculous by criticizing that. However, I appreciate that you didn't fully loose humor over my review. I was pointing at the overpressure waves rising in the magmatic-hydrothermal plume and these – if I remember that correctly from in-depth discussions with Weis – happen on time scales of years or 10s of years; i.e., they are highly relevant. Furthermore, you could learn from Weis et al. how $H_2O$-NaCl phase relations (unfortunately, no $CO_2$) evolve with space and time in such a system to come back to the main problem of your calculations.
- It were these last few points that made me make the comment on "apparently from a volcanology background" as the way the manuscript is written it reads like a naive "what comes out of the fumaroles is what is at depth". There is nothing about "second class scientists" implied but rather should highlight the impression of a surface data-biased view on fluid processes in the deep parts of magmatic-hydrothermal systems (btw. understood as systems dominated by magmatic fluids). A lot of valuable information on the latter is available (mostly from the economic geology community) that could have informed the conceptual model design with quite some advantage but was not

considered. In hindsight, I admit that the statement could be misinterpreted and hope these remarks clarify that.

- Convective zero temperature & zero gradient pressure profiles: another one that was apparently intended to make me look like a beginner by referring to an introductory book. My point was that your "convective" vertical temperature profiles in the reservoirs are an ad hoc invention based on no data and should, therefore be declared as such or be justified. BTW: zero pressure gradient vaporstatic columns would not convect. Regarding that zero pressure gradient, for the fun of it, let's take you own data, for example the Oct10-Jun12 $H_2S$ equilibration conditions: a pure water vapor at 830 °C and 2157 bar would have a density of 432 kg/m$^3$. In the CH4 equilibration reservoir for the same period one would have 412 kg/m$^2$. This is >0.4 times a cold hydrostatic gradient, far away from zero gradient. So, no point to make me look ridiculous when your own data proof you wrong. In my group it is a routine process to perform such obvious checks before adding conceptual figures to a publication.
- "inferred geology". Let me cite the book, page 40: "In particular, according to Zollo et al. (2008), the **inferred** schematic stratigraphy comprises, from top to bottom (Fig. 10):". So, please don't bash me if I use your words.
- Calcite-$H_2S$: you are right, geology rules. I should have expressed much clearer that I was referring to the effect of reactive transport on sulfur content. There, I don't agree that the absolute concentrations don't matter because small concentrations may easier experience massive relative modification (reactive transport is always a competition between equilibrium constant and actual masses present) than bigger ones. Side remark, lines 69-71 of rebuttal: there is no such thing as a "strong acid" at those conditions; acids known to be strong at ambient conditions become weak in the low dielectric constant aqueous solvent at those conditions. As illustration: according to Supcrt, if one trusts it, the logK for $HSO_4^- = SO_4^{2-} + H^+$ in the temperature range of 500 to 800°C and densities from 0.4 to 0.6 g/cm$^3$ is in the range of -8 to -11 ...
- Finally, I think my doubts about the compatibility of "reservoir" and equilibration and a structural transport highway remain valid but there is no point discussing this further here. This applies also for all other points I may not have responded here, too.

I would like to conclude with stating that I start to appreciate the egusphere discussion format. Although I was annoyed by the extra effort compared to what it could have been if the manuscript were properly prepared, I think such discussions can be very helpful and help bridging gaps between different communities, one of which became very obvious here.

---

## Referee Comment (RC3)

[referee-annotated manuscript omitted]

---

## Author Comment (AC1)

*Reply to the General comments of the reviewer*

In the first sentence of the General comments [*Marini et al. attempt to integrate new gas geochemical data and existing information into a revised "conceptual model" of the Solfatara magmatic-hydrothermal system at Campi Flegrei*], there is something wrong, in that we did not produce any new gas geochemical data, whereas we used the Solfatara geochemical database, produced in forty years, first by Cioni and coworkers and second by Chiodini and coworkers, with the aim to monitor the temperatures and total fluid pressures in the three reservoirs which are present at different depths in the Campi Flegrei caldera. As part of our work, we also revised the *conceptual model "of the Solfatara magmatic-hydrothermal system"*, integrating relevant geological data, both collected at the surface and obtained by the deep geothermal wells drilled by Agip-ENEL in the late '70s early '80s, and pertinent geophysical data.

For what concerns the second sentence of the General comments [*I found the manuscript very difficult to review as a lot of important information, namely regarding the methods, the underlying assumptions, uncertainty assessment, known vs. inferred geology, as well as the exact lines of reasoning from data to interpretation are missing*], we recall that, as written in the paper, the methods are the gas-geothermometers and gas-geobarometers specifically derived and calibrated for the Solfatara fumaroles. These gas-geothermometers and gas-geobarometers as well as their underlying assumptions are thoroughly described and discussed in the book of Marini et al. (2022), to whom the interested reader is referred because it would be out of place to repeat all these matters in our manuscript. The uncertainty of gas-geothermometers and gas-geobarometers cannot be assed because this is an impossible exercise for fumarolic fluids' geo-indicators, such as those elaborated and/or used by Cioni et al. (1984, 1989), Taran (1986), Giggenbach (1987), Tedesco and Sabroux (1987), Chiodini and Cioni (1989), Cioni and Marini (1990), Chiodini and Marini (1998), Caliro et al. (2007), Chiodini et al. (2011, 2015, 2016, 2017, 2021), Fiebig et al. (2013), Moretti et al. (2017, 2018, 2020), Buono et al. (2023) and many others. There is no inferred geology in our work. Again, we considered only known geological data, both collected at the surface and obtained by deep geothermal boreholes, as well as known geophysical data to revise the "conceptual model" of the Solfatara magmatic-hydrothermal system. Let us skip, for the moment, the sentence *exact lines of reasoning from data to interpretation are missing*; we will return on this point below.

For what concerns the following sentence of the General comments [*This renders the work in its current form essentially irreproducible and very difficult to review because essentially almost every paragraph can be challenged*]. We do not see any problem to reproduce our results, namely the temperatures and total fluid pressures in the three reservoirs which are present at depth in the Campi Flegrei caldera, in that anyone can use our gas-geothermometers and gas-geobarometers to obtain equilibrium temperature and total fluid pressure values. Thus, we do not see the reason for such a strong statement and the reviewer's mostly non-constructive comments.

Let us now consider the following sentence of the General comments [*Even some (central) elements of the discussion seem to contradict each other; e.g., the concept of separate "reservoirs" as locations of equilibration of different gas*

*geothermometers is obsolete with the assumption of flow through a vertically connected structure*]. Let us try to explain how it is possible that distinct gases (i.e., $H_2S$, $CH_4$, and CO) equilibrate at different temperatures and total fluid pressures in three reservoirs different but connected to each other by a fault zone along which a continuous compressible fluid flow occurs. To this purpose, let us assume that the branches of the fault zone extending from the magma chamber to the surface and connecting the three reservoirs to each other are small-diameter pipelines, the shallow reservoir volume contributing to the fumarolic discharge is a medium-diameter pipeline, and the intermediate and deep reservoirs' volumes contributing to the fumarolic discharge are large-diameter pipelines. The continuity equation written in a simple way for a compressible fluid flow is: $Q = \rho \cdot A \cdot v$, where Q is the mass flow rate (M/T), $\rho$ is the density of the compressible fluid $(M/L^3)$, A is the cross-sectional area $(L^2)$, and v is the velocity of the fluid (L/T). Even if $\rho$ is a function of temperature and pressure, the continuity equation implies that the fluids will move very fast along the fault zone, very slowly in the deep reservoir (in which fluids resides long enough to permit hydrogen sulfide re-equilibration) and in the overlying intermediate reservoir (in which fluids resides long enough to allow methane re-equilibration), and not so fast but not so slowly in the shallow reservoir (in which fluids resides long enough to allow carbon monoxide re-equilibration).

Let us now take into account the subsequent sentence of the General comments [*Also, regarding the core chemistry, the authors (apparently coming from a volcanology background) seem to rely on shallow fumarole gas equilibria and ignore stablished knowledge from the magmatic-hydrothermal systems community for the deeper and higher pressured parts of the system (e.g., H2S generation through the well-known SO2 disproportionation reaction)*]. First of all, the reviewer's words

"*apparently coming from a volcanology background*" are out of place, not to say offensive. The reviewer's words implicitly implies that she/he believes that volcanologists are second-class scientists, compared to the first-class scientists to which she/he belongs. What can we say? We are proud, all three of us, of our common background in geology. Concerning our specialization, two authors (LM and ML) are geochemists and the other author (CP) is volcanologist, unfortunately or fortunately for her. By the way, our scientific profile can be found at the following web pages:

https://sites.google.com/view/luigimarini/home-page https://scholar.google.it/citations?user=gBITrwsAAAAJ&hl=it https://www.igg.cnr.it/organizzazione/personale-igg/matteo-lelli

We do not rely on "*shallow fumarole gas equilibria*", because no gas species equilibrates at surface (fumarolic) conditions. Again, we used the chemistry of the Solfatara fumarolic fluids and the gas-geothermometers and gas-geobarometers of Marini et al. (2022), which were specifically derived and calibrated for the Solfatara fumarolic fluids, taking into account the deviations from the ideal gas behavior (an unprecedented exercise with respect to previous papers on gas-geothermometers and gas-geobarometers), to monitor the temperature and total fluid pressure, over a time interval of ~40 years: (i) in the shallow reservoir (0.25-0.45 km depth), where CO equilibrates; (ii) in the intermediate reservoir (2.7-4.0 km depth), where $CH_4$ attains equilibrium; (iii) in the deep reservoir (6.5-7.5 km depth), where $H_2S$ achieves equilibrium.

Moreover, the reviewer is wrong on our alleged ignorance on the *stablished knowledge from the magmatic-hydrothermal*
*systems community for the deeper and higher pressured parts of the system*, because we are aware of the "*$H_2S$ generation*

*through the well-known SO$_2$ disproportionation reaction*", which is actually thoroughly discussed in Chapter 9 - The Redox Potential and Sulfur Gas Species of Marini et al. (2022) where it is numbered Eqn. (10). We take this opportunity to clarify that the SO$_2$ disproportionation reaction produces not only H$_2$S (as written by the reviewer), which is a weak acid, especially at the high temperatures and pressures of interest, but also sulfuric acid, which is a strong monoprotic acid, as it is totally dissociated in bisulfate ion and hydrogen ion in the considered conditions. This means that the SO$_2$ disproportionation reaction produces a strongly acidic aqueous solution. This aqueous solution reacts promptly with the carbonate rocks which are present above the Campi Flegrei magma chamber, according to the following reaction:

$$CaCO_{3(s)} + H^+ + HSO_4^- = CaSO_{4(s)} + CO_{2(aq,g)} + H_2O.$$

This is the same reaction explaining the widespread occurrence of anhydrite in the geothermal systems of Central Italy, such as Latera (Cavarretta et al. 1985; Marini and Chiodini 1994), even as part of the contact-metasomatism paragenesis, as well as in the geothermal system of Nisyros (Ambrosio et al. 2010), where again carbonate rocks and marbles are present above the magma chamber. The same reaction evidently occurs at the base of the deep reservoir which is present at depths of 6.5-7.5 km in the Campi Flegrei caldera. Anhydrite is the partner of calcite in the heterogeneous reaction:

$$CaSO_{4(s)} + CO_2 + 4\ H_2 = CaCO_{3(s)} + H_2S + 3\ H_2O,$$

representing the theoretical foundation of the H$_2$S gas-geothermometer and gas-geobarometer of Marini et al. (2022).

For what concerns the following sentence of the review [*I think that for these reasons the manuscript should not be published in its current from and requires a complete revise beyond "major revisions"*], we are willing to improve our manuscript in response to constructive and scientifically valid comments, but none of the previous ones meet these essential requirements.

We have no comment on the following sentence of the General comments [*In the following I will highlight some key problem areas (and for each one only some facets and ignore many questionable details) of the manuscript but will refrain from a detailed list of all issues*], which is the last one of the section General comments.

**Reply to the Specific comments 1: methods and assumptions behind them**

Let us consider the first sentences of the Specific comments 1 [*The authors rely on three gas equilibria to obtain "distinct equilibration temperatures and related total fluid pressures". They do not elaborate how these were computed and not even how such a "equilibration temperature" is to be interpreted (e.g.: is it the lowest temperature at which the reaction went to equilibrium along the fluid's upflow path and which was then preserved due to kinetic limitations at lower temperature?), and how such equilibrium could be demonstrated at all*]. In the sections Methods and Results of the revised manuscript, we added a few words to clarify these points. The gas equilibration temperature is the temperature at which the considered reaction or the considered key gas species (i.e., H$_2$S, CH$_4$, and CO) attain chemical equilibrium. This is assumed to be preserved at lower temperatures because reaction kinetics becomes too small. In general, in geothermometry and geobarometry, equilibrium cannot be demonstrated, it is assumed. Furthermore, in the case of Solfatara fumaroles, the computed equilibrium temperatures and pressures cannot be compared with any measured value, because the nearest deep geothermal well, called CF23/Agnano 1[1], is found at a distance of ca. 1 km from the Solfatara fumaroles. This is a problem not only for our manuscript, but also for all previous geothermometric-geobarometric studies on the Solfatara fluids such as those of Cioni et al. (1984, 1989), Tedesco and Sabroux (1987), Chiodini and Cioni (1989), Cioni and Marini (1990), Chiodini and Marini (1998), Caliro et al. (2007), Chiodini et al. (2011, 2015, 2016, 2017, 2021), Fiebig et al. (2013), Moretti et al. (2017, 2018, 2020), Buono et al. (2023) as well as the monthly INGV Bulletins. We think that there is no need to recall the general considerations and assumptions which are common practice in geothermometric and geobarometric applications.

Let us now take into account the following sentences of the Specific comments 1 [*It remains completely opaque how the computations were done. I would argue that at least some of the stated computations are currently not meaningfully possible for the lack of adequate thermodynamic data and models, e.g. for a complex gas phase in equilibrium with a 21 or 33.5 wt% aqueous NaCl brine – for the conditions stated there exists neither an adequate solubility model for the gases in such a brine*

*nor are even remotely accurate activity or fugacity coefficients available. How then could temperatures and pressures be derived? I tried to make my way through the main author's 2022 book (375 pages, the main findings of which the present manuscript is supposed to summarize and explain) but I could not identify a coherent summary of the computations and the underlying assumptions with reasonable effort. This manuscript should deliver a crystal-clear and reproducible description and openly lay out and discuss the major assumptions to become a publishable contribution*]. Since gas species are assumed to equilibrate in a single saturated vapor phase in equilibrium with a 21 or 33.5 wt% aqueous NaCl brine, there is no need for an "*adequate solubility model for the gases in such a brine*". Brine-vapor coexistence is assumed to fix $H_2O$ partial pressure. This is the only role played by the brine, in our approach, which is similar to that assumed by Giggenbach (1987). Fugacity coefficients of gas species were computed by Marini et al. (2022) following different approaches. First, the fugacity coefficients of $H_2O$ and $CO_2$ were obtained along the isenthalpic expansion path of Bocca Grande fluids of different composition: (i) by using the GERG-2008 thermodynamic module of the CONVAL Software Package and the virial EOS (see pages 102-111 of Marini et al. 2022), (ii) by using the Peng-Robinson EOS (see pages 111-118 of Marini et al. 2022), and (iii) from Table 3 in Appendix A of Gallagher et al. (1993) reporting the results of the EOS of these authors. Then, the three series of fugacity coefficients of $H_2O$ and $CO_2$ were compared (see pages 118-121 of Marini et al. 2022). After this comparison, Marini et al. (2022) decided to use the Peng-Robinson EOS to compute the fugacity coefficients of all the gas species of interest (not only $H_2O$ and $CO_2$, but also $H_2S$, $H_2$, $CH_4$, and $CO$) not only for the isenthalpic expansion path, but
* * *
[1] Well CF23/Agnano 1 was drilled in the '50s by SAFEN "on the eastern slopes of the Solfatara relief and reached a total depth of 1841 m, where a maximum temperature of 325 °C was measured (Minucci 1964)", as reported by Marini et al. (2022).

also for the linear P-T decompression path and the pressures and temperatures of coexistence of a brine containing 21 or 33.5 wt% NaCl with the related vapor phase (see pages 122-142 and pages 308-312 of Marini et al. 2022). Fugacity coefficients were then considered in the calibration of gas-geothermometers as discussed at pages 286-287, 335, and 344 of Marini et al. (2022). These explanations were synthetically added to the Introduction of the revised manuscript, to improve its clarity.

In the subsequent part of the Specific comments 1, the reviewer focuses again on the $SO_2$ disproportionation reaction [*There exists a relatively elaborate assessment of sulfur chemical behavior in the deep vs. shallow parts of magmatic-hydrothermal systems that was established some decades ago in the economic geology community (just to name the works by Einaudi and Hedenquist as an example) including the importance of "rock-buffered" vs. "direct degassing" paths. The very relevant insights presented there go largely ignored by the present authors. Namely, the important $SO_2$ disproportionation reaction to*

*generate $H_2S$ (and there are variations of the theme depending on temperature, rock, etc.) is ignored although it is considered most important by many researchers in the field. The reactions (3) and (4) involving calcite also seem odd; most people working on the deeper parts of magmatic-hydrothermal systems would see these reactions going to the left (albeit possibly rather involving $SO_2$ and using plagioclase rather than calcite) to sequester sulfur rather than creating $H_2S$, see, e.g., Henley at al., JVGR 2022. I wonder what "$H_2S$ temperatures" would come out if this was properly taken into account*

*of if "H2S temperatures" would have any meaning then.*]

Above, we have already rejected this critique on the $SO_2$ disproportionation reaction. Since the reviewer refers to the works by Einaudi and Hedenquist, we take this opportunity to consider the paper by Einaudi, M. T., Hedenquist, J. W., & Inan, E. E. (2005). *Sulfidation state of fluids in active and extinct hydrothermal systems: Transitions from porphyry to epithermal environments*. [doi:https://doi.org/10.5382/SP.10.15], in which the authors contrast the oxidation and sulfidation states of fresh igneous rocks from arc environments and the sulfidation states of sulfide assemblages in calc-alkalic porphyry copper, porphyry-related base metal veins, and epithermal gold-silver deposits with compositions of fluids from active systems by plotting vapor compositions in log $f_{S2}$ − 1,000/T, $R_H$ − 1,000/T, and $R_S$ − 1,000/T diagrams, where $R_H$ = log $(f_{H2} / f_{H2O})$ ≈ log $(X_{H2}/X_{H2O})$, $R_S$ = log $(f_{H2} / f_{H2S})$ ≈ log $(X_{H2}/X_{H2S})$, and X = mole fraction of the gas. It must be underscored that $H_2$ and $H_2S$ are apolar gas molecules, whose fugacity coefficients increase with increasing temperature and pressure, whereas $H_2O$ is a polar gas molecule, whose fugacity coefficient decreases with increasing temperature and pressure, as shown by Marini et al. (2022). Although the findings of Marini et al. (2022) refer, strictly speaking, to the Solfatara fluids, the fugacity coefficients of $H_2$, $H_2S$, and $H_2O$ are expected to vary in the same way, irrespective of chemical and physical conditions of the considered system, because in any case $H_2$ and $H_2S$ are apolar molecules and $H_2O$ is a polar molecule. Therefore, the assumption log $(f_{H2} / f_{H2S})$ ≈ log $(X_{H2}/X_{H2S})$ might still be correct to some extent, whereas the assumption log $(f_{H2} / f_{H2O})$ ≈ log

$(X_{H2}/X_{H2O})$, which was first proposed by Giggenbach (1987), becomes increasingly wrong with increasing temperature and pressure. We believe that works affected by these gross blunders do not represent the top of knowledge and those who present them in this way have a poor knowledge of gas geochemistry in magmatic-hydrothermal system. We do not understand the reason to obstacle, in a non-constructive way, new works aimed at increasing the knowledge of new geo-indicators.

Since the *"reactions (3) and (4) involving calcite also seem odd"* to the reviewer, we underscore once again that in the Campi Flegrei, above the magma chamber, there is a thick sequence of carbonate rocks, as revealed by a number of geophysical studies on the active seismic reflection data of the SERAPIS survey and the passive seismic data of the 1982-1984 bradyseismic crisis (see references in our manuscript). The thick sequence of carbonate rocks occurring above the Campi Flegrei magma chamber is the same sequence cropping out all around the Campanian Plain, which has been downthrown by the boundary faults of the plain and other NW-trending tectonic structures (called Apenninic by Italian geologists) and has been dissected by the NE-trending conjugated structures (called anti-Apenninic by Italian geologists). The occurrence of these carbonate rocks above the magma chamber prompted us to use calcite (instead of plagioclase) as partner of anhydrite in reactions (3) and (4).

For what concerns the sentence of the Specific comments 1 *"most people working on the deeper parts of magmatic-*
*hydrothermal systems would see these reactions going to the left (albeit possibly rather involving $SO_2$ and using plagioclase rather than calcite) to sequester sulfur rather than creating $H_2S$, see, e.g., Henley at al., JVGR 2022"* let us comment separately on the use of *"plagioclase rather than calcite"* and on the direction of reactions (3) and (4).

We do not agree with the reviewer that *"most people working on the deeper parts of magmatic-hydrothermal systems"* would have used the *"plagioclase rather than calcite"*, because geologists know very well that geology is the first
fundamental constraint that cannot be overlooked. We would certainly have considered reactions with plagioclase instead of calcite, if Al-silicate rocks were present above the magma chamber, but this is not the case at Campi Flegrei.

Reactions (3) and (4) are written here below for clarity:

$$CaSO_{4(s)} + CO_2 + 4\ H_2 = CaCO_{3(s)} + H_2S + 3\ H_2O \qquad\qquad (3)$$
$$CaSO_{4(s)} + CH_4 = CaCO_{3(s)} + H_2S + H_2O. \qquad\qquad (4)$$

For what concerns these two reactions, we would like to underscore that we do not care if they proceed from left to right (that is as they are written) or in the opposite direction, since, they are equilibrium reactions fixing $H_2S$ and $CH_4$ fugacities, respectively.

**Reply to the Specific comments 2: the core of the conceptual model**

Let us consider the first sentences of the Specific comments 2 [*When looking closer one has a hard time to understand reasoning behind the "conceptual model". On the one hand, three "reservoirs" at different depth, separated by aquicludes, are postulated; on the other hand – to make the fluids migrate from a magma at >8km depth to the surface – connected flow through a vertically extensive "structure" needs to be invoked.*

*Now, first and foremost, I had a hard time understanding why the different gas equilibration temperatures should be specifically connected to the individual "reservoirs" and how the latter were inferred in the first place. Is there some circular reasoning? Or does actual, drilled geology with porosity and permeability measurements come into play? There is some mentioning of wells etc. but the exact reasoning remained blurry to me, at best. So, one cannot even tell if the depth of "reservoirs" and "aquicludes" is well-constrained or just a guess.]*

As already recalled above, the three reservoirs at different depth and the interposed aquicludes are not postulated, as written by the reviewer, but are constrained by known geological data, both collected at the surface and obtained by deep geothermal boreholes, which were drilled at depths from 1.5 to 3 km approximately, as well as by known geophysical data which were processed and interpreted by a number of studies. All is all, these data provide a well-constrained conceptual model extending to ca. 8 km where the magma chamber is located. There is no circular reasoning in the integration of geological and geophysical data to reconstruct the conceptual model of the Solfatara magmatic-hydrothermal system at Campi Flegrei, that we did following the best practices used in geothermal exploration and other applied geo-sciences. To help the reader, we added a conceptual model cross-section of the system of interest, showing the shallow, intermediate, and deep reservoirs (Figure 5 of the revised manuscript).

Porosity and permeability measurements were performed on cores obtained in the deep geothermal boreholes, but obviously they do not extend at depths > 3 km approximately. These data are summarized by Rosi and Sbrana (1987). Nevertheless, they are not essential for the elaboration of the conceptual model, they would be much more important, not to say essential for the implementation of a numerical model, but this exercise is not among our aims.

The vertically permeable tectonic structure extending from the magma chamber to the surface and permitting the upward fluid flow through the three reservoirs and the interposed aquicludes "was activated during the final phase of the 1982-1984 seismic crisis along tectonic trends already active in the past (Rosi and Sbrana, 1987), as suggested by the occurrence of low-magnitudo earthquakes at depths of 0-8 km (D'Auria et al., 2011)" (see lines 320-323 of our manuscript). We take this opportunity to recall that large changes in several chemical components (e.g., $N_2$, $CO_2$, $H_2S$) of fumarolic gases occurred in the time interval May-September 1984 (e.g., Cioni et al. 1989; Caliro et al. 2014; Buono et al. 2022). They fit perfectly with the activation of this faulted-fractured zone and the transition from closed state to open state of the Solfatara magmatic-hydrothermal system, as discussed at lines 197 to 210 of our manuscript.

Although we have already explained above "*why the different gas equilibration temperatures should be specifically connected to the individual "reservoirs"*, let us return again on this point to answer the reviewer. Accepting that there is a continuous, upward fluid flow along the vertically permeable tectonic structure extending from the magma chamber to the surface and, consequently, through the three reservoirs and the interposed aquicludes, the residence time of the fluids in each reservoir and in each aquiclude is expected to be directly proportional to the volume contributing to the fumarolic discharge of each reservoir and each aquiclude; thus, the fluids will spend: (1) a relatively long time in the deep and in the intermediate reservoirs because of their large volumes, even the fraction contributing to the fumarolic discharge; (2) a not too long and not too short time in the shallow reservoir because it has a volume much smaller than the deep and the intermediate reservoirs; and (3) a short or very short time in the aquicludes, where the volume available to fluid flow is small or very small. Moreover, in a given time interval, the temperature and total fluid pressure are expected to be constant or nearly so in the three reservoirs, whereas the temperature and total fluid pressure are expected to experience large upward decreases along the faulted-fractured zone crossing the aquicludes (see below for further details). Thus, in each one of the three reservoirs and in a given time interval, gas species are expected to attain equilibrium at a given temperature, total fluid pressure condition because of both the large or relatively large residence time and the constant or relatively constant temperature and total fluid pressure. In contrast, equilibrium temperature is expected to experience small re-adjustments during the transit from one reservoir to the other because of the small residence time in the aquicludes, in spite of the large decrease in temperature and fluid pressure.

Previous considerations clarify also the following sentences of the Specific comments 2 [*Then, if there is a vertically connecting structure – what is the meaning of the temperatures then? "Equilibration" will happen (and be "frozen in" if my above speculation about the meaning of "equilibration temperature" is correct) somewhere along the flow path and why*

*should that be connected to the depth of any of the "reservoirs" then?*], whereas the following sentences of the Specific comments 2 call for some considerations [*Wouldn't it make more sense that the increase in apparent temperatures and pressures (if correct) reflects rather a change in the hydraulic regime (or the degassing rate) such that the chemical signal of deeper fluids gets better preserved rather than a specific "reservoir" getting hotter and stronger pressurized (see, e.g., the overpressure waves in a magmatic-hydrothermal systems self-developing in the simulations of Weis et al., 2012; later*

*suggested also by Lupi et al. for Campi Flegrei). What would speak for and against such different possibilities, why aren't they considered and also tested against the data?*] Weis et al. (2012) implemented a numerical model to explain porphyry-type ore deposits, which is not relevant for the Solfatara magmatic-hydrothermal system at Campi Flegrei because the geological contexts are totally different. In fact, the three porphyry-style deposits considered by Weis et al. (2012), namely, Yerington in Nevada, Bingham Canyon in Utah and Batu Hijau in Indonesia are sustained by an unspecified source pluton or an unspecified inferred source pluton, whereas the Campi Flegrei volcanic area is characterized by a nested caldera with a magma chamber, whose top is positioned at ca. 8 km depth, probably hosting a trachyandesitic magma, according to Caliro et al. (2014) and Buono et a. (2022). It is unclear to us if the numerical model implemented by Weis et al. (2012) is able to reproduce the geological complexities of (at least) one of the three porphyry-style deposits because this crucial point is not discussed. Even more important, it is unclear to us the relation between the study of Weis et al. (2012) and our work, also considering that (1) the two time-windows are totally different (the time-scale of ore formation is between 50,000 and 100,000 years, whereas at the Solfatara we are considering a time interval of 38 years only), (2) we focus on the geochemistry of Solfatara fluids, whereas no data on fluid chemistry are given by Weis et al. (2012), apart from generic sentences such as "*We assume that, on cooling through the solidus temperature of 700°C, 5 wt% of the magma are released as aqueous fluid with 10 wt% NaCl through a cupola in the roof of the magma chamber.*"

Lupi and coworkers speculate on the possible effects of large regional earthquakes on the Campi Flegrei magma chamber simulating "*the propagation of elastic waves and show that passing body waves impose high dynamic strains at the roof of the magmatic reservoir of the Campi Flegrei at about 7 km depth. This may promote a short-lived embrittlement of the magma reservoir's carapace, which is otherwise impermeable during inter-seismic times. Such failure allows magma and exsolved volatiles to be released from the magmatic reservoir. The fluids, namely exsolved volatiles and/or melts, ascent*

*through a nominally plastic zone above the magmatic reservoir.*" Although the idea might be interesting, the supporting evidence is not convincing at all. Again, the numerical model implemented by Lupi and coworkers is very generic and does not reproduce the geological reality of the Solfatara magmatic-hydrothermal system at Campi Flegrei.

Even more important, the cause-effect relationship between regional earthquakes and uplift leaves much to desire. In particular, we wonder why the numerous regional earthquakes that occurred before 1945 (the beginning of the time window considered by Lupi and coworkers) did not activate any uplift at Campi Flegrei, which begun in 1950, according to Del Gaudio et al. (2010), or in 1945, according to the pioneering investigation of Ranieri (1952). In our opinion, the analysis of the cause-effect relationship between regional earthquakes and uplift is a good example of sample with the built-in bias, similar to the cases discussed by Darrell Huff in his masterpiece "How to Lie with Statistics". Here below we report three examples taken from the book written by Darrell Huff:

(i) "Does early discovery of cancer save lives? Probably. But of the figures commonly used to prove it the best that can be said is that they don't. These, the records of the Connecticut Tumor Registry, go back to 1935 and appear to show a substantial increase in the five-year survival rate from that year till 1941. Actually those records were begun in 1941, and everything earlier was obtained by tracing back. Many patients had left Connecticut, and whether they had lived or died could not be learned. According to the medical reporter Leonard Engel, the built-in bias thus created is 'enough to account for nearly the whole of the claimed improvement in survival rate'."

(ii) "A psychiatrist reported once that practically everybody is neurotic. Aside from the fact that such use destroys any meaning in the word "neurotic," take a look at the man's sample. That is, whom has the psychiatrist been observing? It turns out that he has reached this edifying conclusion from studying his patients, who are a long, long way from being a sample of the population. If a man were normal our psychiatrist would never meet him."

(iii) "For further evidence go back to 1936 and the *Literary Digest's* famed fiasco. The ten million telephone and *Digest* subscribers who assured the editors of the doomed magazine that it would be Landon 370, Roosevelt 161 came from the list that had accurately predicted the 1932 election. How could there be bias in a list already so tested? There was a bias, of course, as college theses and other post mortems found: People who could afford telephones and magazine subscriptions in were not a cross section of voters. Economically they were a special kind of people, a sample biased because it was loaded with what turned out to be Republican voters. The sample elected Landon, but the voters thought otherwise."

[Figure]

In addition to the built-in bias in the sample considered by Lupi and coworkers, their cause-effect relationship between regional earthquakes and uplift might also be caused by chance, a common problem which is thoroughly discussed in chapter 8 of the book by Darrell Huff "Post Hoc Rides Again".

Finally, for what concerns the last question of the Specific comments 2 [*What would speak for and against such different possibilities, why aren't they considered and also tested against the data?*] we underscore that it applies also to the works of Weiss and coworkers and Lupi and coworkers.

All in all, we have chosen to ignore the works of Weiss and coworkers and Lupi and coworkers to avoid criticizing them harshly in our manuscript. After all, every author should be allowed to publish her/his results, provided that they have scientific validity and are not affected by errors of any kind. We only want to underline that it is strange, very strange that the reviewers of the works of Weiss and coworkers and Lupi and coworkers have not noticed the problems discussed above that are so evident to our eyes.

Considering the following sentences of the Specific comments 2, [*In the whole conceptual model discussion, speculative ideas (such as Fournier's "self-sealing" quartz layer for which later studies found little evidence), inferred vs. drilled geology, geochemical data with different possible interpretations etc. are just mingled without testing for plausibility etc. Personally, I think that this is quite far away from best practices; the different ideas should be formulated as hypotheses and then tested to the degree possible*] first of all, it is funny to note that the masterwork paper written by Fournier (1999) is quoted without any criticism by Weiss and coworkers and Lupi and coworkers, but here the reviewer considers a speculative idea the *Fournier's "self-sealing" quartz layer for which later studies found little evidence*. As written in the book of Marini
et al. (2022), the reduction in permeability close to an intrusive heat source caused by quartz precipitation was discussed not
only by Fournier (1999) but also in several other studies (e.g., Fournier 1985; Wells and Ghiorso 1991; Lowell et al. 1993;
Moore et al. 1994; White and Mroczek 1998; Saishu et al. 2014; Scott and Driesner 2018), which possibly are unknown to
the reviewer. Concerning the "best practices" to be followed in data interpretation, we agree with the reviewer and, as a
matter of fact, we applied these "best practices" also in our manuscript, for example at lines 151-167, where we recalled
different hypotheses to explain the gravimetric low present in the Campi Flegrei caldera, as well as, as another example, in
the book of Marini et al. (2022) where, in chapters 6, 7, and 8, we took into consideration the different geothermometric-
geobarometric approaches applicable to the $H_2O$-$CO_2$-$CH_4$-$CO$-$H_2$ system. Afterwards, in the manuscript we selected a series
of CO-, $CH_4$- and $H_2S$ equilibrium temperatures and total fluid pressures (see lines 86-92 of our manuscript) to simplify the
discussion and without pretending that these selected values are the indisputable truth. Nevertheless, if a different series of
CO-, $CH_4$- and $H_2S$ equilibrium temperatures and total fluid pressures is selected, the trend of increasing temperature and
total fluid pressure with time is still observed although with some differences. Thus, there is little doubt, on the progressive
pressurization of both the deep aquifer and the intermediate aquifer, as well as on the good correspondence between the
chronogram of the intermediate aquifer overpressure and the chronogram of ground uplift (see lines 362-387 of our
manuscript). Nevertheless, we are aware of the Karl Popper' words *"whenever a theory appears to you as the only possible
one, take this as a sign that you have neither understood the theory nor the problem which it was intended to solve"* or, in
simpler words … we are so close to perfection that it scares us.

**Reply to the Specific comments 3: drawing straight lines without reasoning**

For what concerns the first sentence of the Specific comments 3 [*The temperature and pressure profiles in Figs. 5 and 6
hinge on the – untested – assumption that the gas equilibration temperatures are representative for three "reservoirs" at
different depth*], we underscore that at lines 186-277 of our manuscript we have provided several pieces of evidence
indicating that the three gas equilibration temperatures and related total fluid pressures refer to the three "reservoirs"
positioned at different depths. We have proven that our geothermometers and geobarometers are effective tools to
understand the processes controlling the bradyseism and to monitor the evolution of the bradyseismic crisis, as indicated by
the correspondence, which is not coincidental, between the time changes in the overpressure of the intermediate reservoir
and the time changes in ground movements. This is an important step forward compared to the CO-$CO_2$ equilibrium
temperatures and pressures reported and discussed in all recent articles by Chiodini and coworkers, including the monthly
INGV Bulletins. In fact, the CO-$CO_2$ pressure refers to the shallow hydrothermal aquifer and is too low to explain ground
movements.

Our manuscript is not an academic exercise because to understand the processes controlling the bradyseism and to monitor the evolution of the bradyseismic crisis are crucial aspects for the scientific community and the Italian authorities in charge of the volcanic surveillance of Campi Flegrei to mitigate the natural risks in this densely populated area. To clarify this point, we added a few sentences in the section Conclusions.

Let us take into consideration the following sentences of the Specific comments 3 [*In Fig. 5, temperature in each reservoir is*
*taken to be constant. Unless I missed something important no reason is given why this should be the case. Rather, it is taken as granted (out of nothing) and then it is postulated that convection inside the reservoir homogenizes the temperature. Between the reservoirs – again: unless I overlooked something important – straight lines are drawn without reasoning and then it is stated that the "the heat transfer appears to be controlled by conduction". I think this is quite poor scientific practice to just draw a straight line in the absence of data and then to assume it is correct and make such a conclusion.*]

To answer these reviewer criticisms, we have reported below some relevant sentences, in blue color, from the textbook written by Malcolm A. Grant and Paul F. Bixley (2011) **Geothermal Reservoir Engineering**, Second Edition, which is "the only training tool and professional reference dedicated to advising both new and experienced geothermal reservoir engineers". "The simplest distinction made in temperature profiles is between conductive and convective profiles. When rock is impermeable, heat is transported by conduction. This produces a characteristic profile where temperature increases
linearly with depth; the gradient will change if there is a change in thermal conductivity of the rock.

[Figure]

**FIGURE 4.2** Temperature profiles in GPK-1 and GPK-2. Source: *Genter et al., 2009. © Geothermal Resources Council.*

Convection by contrast is a far more efficient means of heat transport than conduction. Once there is some permeability in the rock - and the required permeability is much less than what is needed for economic well performance - the fluid motion controls the temperature distribution. Convective profiles can take a considerable variety of forms, with isothermal sections, inversions, boiling sections, and mixtures of all of these. Figure 4.2 shows temperature profiles from two wells at the engineered geothermal systems (EGS) project at Soultz, France (Genter et al., 2009). There are three sections on the profile. The first kilometer has a high gradient and linear profile, indicating conductive transport. Then from 1 km to 3.3 km there is a much lower gradient, which is attributed to a convective system along faults and fissure zones. Finally, below 3.3 km there is again a high linear gradient, indicating conductive heat transport and consequently lower permeability in the surrounding formations.

**4.4.2. Isothermal**

An isothermal profile is a section of the well where the temperature is constant or nearly constant with depth. This can reflect circulation of fluid in a section of the wellbore or interzonal flow (without boiling), or it may be that the reservoir itself has isothermal temperatures due to convection."

For what concerns the following sentences of the Specific comments 3 [*For fluid pressure (Fig. 6) the "reservoirs" are also drawn to have constant pressure even if more than a km high. This is obviously unphysical as there would have to be a hydrostatic pressure gradient (not necessarily linear as density of the fluid may vary with depth),* the constant pressure over thicknesses of more than a kilometer is not unphysical as erroneously claimed by the reviewer. Instead, the vapor-static pressure or steam-static pressure gradient is a typical characteristic of vapor-dominated (dry-steam) systems, as discussed: (1) long ago by White et al. (1971) and Truesdell and White (1973), (2) in several later papers (e.g., Ingebritsen and Sorey 1988; Williamson 1990; Scott 2020), in the textbook of Grant and Bixley (2011, see above) and (3) summarized in the book by Marini et al. (2022) at pages 190-191.

*Ingebritsen, S. E., & Sorey, M. L. (1988). Vapor-dominated zones within hydrothermal systems: Evolution and natural state. Journal of Geophysical Research: Solid Earth, 93(B11), 13635-13655.*

*Scott, S. W. (2020). Decompression boiling and natural steam cap formation in high-enthalpy geothermal systems. Journal of Volcanology and Geothermal Research, 395, 106765.*

*Williamson, K. H. (1990). Reservoir simulation of The Geysers geothermal field. In Proceedings of the Fifteenth Workshop on Geothermal Reservoir Engineering, Stanford University, Stanford, California, January 23-25, 1990 SGP-TR-13.*

*Some other specific comments*

*I only list a few, which I think are important:*

- *1: I think that a Google Earth snapshot is a no-go and contains much less info than, e.g., a simplified line art map showing the main geologic structures and the main geographic locations. Add a square that show the location of Fig 1b. In 1b show a scale rather than an unexplained coordinate system.*

- *Avoid self-celebrating statements in the introduction.*

- *"standard deviation" in line 98 includes what? Just the effect of variable gas analysis? What's the uncertainty of the thermodynamic analysis to obtain T and P? Possibly much larger?*

- *I can't follow the reasoning in 4.3; if you assume that (1) and (2) work well separately, than (5) should work as well as it is simply a linear combination of the two, right?*

Please find our answers here below:

(i) As already recalled above, we are geologists and geologists like Google Earth maps because they are very informative. For example, the senior author of our manuscript (LM) is co-author of the book by Cioni R., Marini L. 2020. A Thermodynamic Approach to Water Geothermometry, Springer Geochemistry Series, 415 pp, containing several Google

Earth maps. In Figure 1a we added a square showing the location of Figure 1b. Since kilometric coordinates are reported on the two axes of Figure 1b, there is no need to show a scale in Figure 1b, because it would be redundant. We added this explanation to the caption of Figure 1b.

(ii) We assume that the self-celebrating statements, which are not specified by the reviewer, are the words in red in the section from line 66 to line 69, which is reported here "Recently, we proposed new gas geothermometers and geobarometers, which were suitably calibrated for different plausible expansion paths of the Solfatara fluids, also considering the deviations from the ideal gas behaviour, for the first time. Our results were presented and thoroughly discussed by Marini et al. (2022). In this work, we summarize the main findings of Marini et al. (2022), taking into account the last data produced by Buono et al. (2023) for Bocca Grande and Bocca Nuova fumaroles for October 2020 to January 2022. Moreover, in this work, we use our geothermometric and geobarometric results, as well as the information from other disciplines (e.g., surface geo- volcanological surveys, data from geothermal deep wells, and geophysical investigations) to elaborate a revised conceptual model of the Solfatara magmatic-hydrothermal system which extends at magmatic depths (~8 km) and represents a considerable step forward with respect to previous conceptual models." If so, there is nothing wrong and no self-celebration in our words and we do not see any reason to omit these sentences simply reporting the truth.

(iii) The standard deviation given in Tables A1 and A2 is just the standard deviation of the CO-, $CH_4$-, and $H_2S$-equilibrium temperatures and related total fluid pressures of each time interval (incidentally, these standard deviations can be verified using the values reported in the supplementary material). In other words, the standard deviation given in Tables A1 and A2 is "*just the effect of variable gas analysis*". The uncertainty of computed equilibrium temperatures and related total fluid pressures includes several contributions, such as the uncertainties on the third law entropy and on the enthalpy of formation from the elements which are reported in the following table (from Chase, M. 1998. NIST-JANAF Thermochemical Tables,

4th Edition. Journal of Physical and Chemical Reference Data, Monograph No. 9, 1951 pp. https://srd.nist.gov/JPCRD/jpcrdM9.pdf)

| Gas species | Error on $\Delta H_f°$ | Error on S° | Upper T |
|---|---|---|---|
| | kJ/mol | J/(mol K) | K |
| $CH_4$ | 0.34 | 0.04 | 6000 |

| | | | |
|---|---|---|---|
| CO | 0.17 | 0.04 | 6000 |
| $CO_2$ | 0.05 | 0.12 | 6000 |
| $H_2$ | 0 | 0.033 | 6000 |
| $H_2O_{(L)}$ | 0.042 | 0.079 | 500 |
| $H_2O_{(V)}$ | 0.042 | 0.042 | 6000 |
| $H_2S$ | 0.8 | n.r. | 6000 |
| $SO_2$ | 0.21 | 0.08 | 6000 |

The uncertainty of computed equilibrium temperatures and related total fluid pressures also includes the uncertainties on the Maier-Kelley coefficients (which are small for gas species because their Cp values are well constrained), on the fugacity coefficients (on the fourth or third decimal figure), as well as the uncertainties related to the regression analysis performed to obtain the different geothermometric and geobarometric equations. Nevertheless, the major effect is probably the variability of analytical data.

(iv) In case of overall (full) equilibrium, the CO-equilibrium temperature related to reaction (1), the $CH_4$-equilibrium temperature related to reaction (2) and the CO-$CH_4$-equilibrium temperature related to reaction (5) are the same, with deviations from a few degrees to a few dozen degrees. For the Solfatara fluids, this overall equilibrium was observed until 1985. Afterwards, the CO-equilibrium temperature related to reaction (1) remained nearly constant, the $CH_4$-equilibrium temperature related to reaction (2) increased more and more with time, and the CO-$CH_4$-equilibrium temperature related to reaction (5) increased but much less than the $CH_4$-equilibrium temperature. The ensuing increasing difference between the three temperatures is due to CO-$CH_4$ disequilibrium. In this case, the CO-equilibrium temperature and the $CH_4$-equilibrium temperature are still meaningful, whereas the CO-$CH_4$-equilibrium temperature is meaningless. Further details are given by Marini et al. (2022) at pages 209-215, where the CO-equilibrium temperature is called T(RWG), the $CH_4$-equilibrium temperature is called T(SS4) and the CO-$CH_4$-equilibrium temperature is called T(CCC). We decided to stop to use the SS1, SS3 and SS4 acronyms and other acronyms used by Marini et al. (2022) because we were told that they were reminiscent of Nazi SS.

---

## Author Comment (AC2)

*Reply to the second series of comments of reviewer n.1*

**PLEASE NOTE THAT THE REVIEWER'S WORDS ARE IN RED COLOR AND OUR REPLY IS IN BLACK COLOR.**

Thank you for the reply, but I must admit that I feel that the contents of my previous review rather got confirmed than refuted. In the following, after a personal introductory remark, I will provide a second, in-depth review of several aspects of the study that will unambiguously demonstrate that key calculations are deficient with the consequence that key parts of the results on pressure lack scientific validity and that likely the temperatures as well are questionable. I will conclude with an incomplete list of answers to the authors rebuttal on points where I find that it may be beneficial for clarification.

*Introductory remark*

Let me start with a personal remark on the reviewing effort. A significant part of the authors' rebuttal is in essence "we explained this in the book Marini et al., 2022". And that was my very point in my previous commentary: a lot of information that is essential to allow a competent, science-focused review (or simply: a read) of the manuscript is not in there. It was submitted as a *research* article and as such should contain sufficient information about the concept, methods and assumptions to allow the reviewer an informed preparation of checking for, quoting the authors, "scientific validity and ... errors of any kind". I estimate that a few manuscript pages on Methods with carefully chosen and explained content would have guided the reviewer (or just reader) well enough to decide whether and what to follow up or not by reading the book or parts of it and they would have much increased the value of the manuscript. The manuscript condensed ca. 150 **pages** in the book to ca. 15 **lines** for the "Methods" and made not a single statement on important underlying concepts, assumptions, limitations, etc. This would have been even more important as the authors claim to have introduced novel tools.

In the version that I reviewed I was essentially forced to explore the book in quite some depth to just get the basics of what was performed in the study, not even to then follow up on the science done. I take peer reviewing serious and I am happy to spend two or three full working days for an in-depth review of an interesting manuscript (as I did then). But, here, this was not leading me anywhere close to understanding the full line of arguments, approaches, assumptions etc. for this study. The book does not make it easy for the reader to detect that information, which is often hidden in very long technical discussions that frequently get as detailed as instructing the reader about how to copy values around in Excel spread sheets (I am not criticizing that this is documented, it is great it is, but it just makes the book a very hard read). For me this was the first time in more than three decades of reviewing that I encountered a paper that expects reading a book in-depth before it can be understood. In essence, this meant reviewing the (published!) book, not the manuscript. I hope the authors may understand that this felt like them making my reviewer job as hard as possible and that I did not feel positive about the extra, in my opinion unnecessary, time effort.

In the "Introductory remarks" section the reviewer emphasizes that the review of our paper was very difficult and time-consuming and that "*he/she was essentially forced to explore the book in quite some depth to just get the basics of what was performed in the study*" and "*it was the first time in more than three decades of reviewing that*" the reviewer "*encountered a paper that expects reading a book in-depth before it can be understood*". This is a totally unfair and unjustified criticism since our work is self-consistent. We simply did not consider necessary to rewrite the equations of geothermometers and geobarometers in the main text because they are reported in the EXCEL spreadsheet, which is part of our paper as supplementary material. If one wants to understand how we derived these equations, she/he can read the book, but it is a personal choice which, we believe, should be motivated only by scientific interest and not by other reasons. For what concerns the sentence "... the authors claim to have introduced novel tools" we want to clarify that we do not claim to have done it, we simply did it. And this is not mendacious self-praise, but it is the pure and sacrosanct truth. For what concerns our book, let us say that it was already reviewed by competent scientists

before its publication. For what concerns the critiques of the reviewer of our book, let us say that they are unsolicited, because nobody asked the reviewer to review our book, they are a petty ploy to discredit both the book and our manuscript and they are written with ethically inappropriate words and style and denote a total lack of respect towards us.

*Additional Review*

Now, although my duty was to review the manuscript and not the book, I felt challenged by the style and content of parts of the rebuttal (some more words on that in the answer list in the third part of this reply) and dived rather deeply into some parts of the "thermometry" and "barometry", which I already felt look suspicious when browsing the book before for understanding what was done in the study. After that new exercise (another two full working days including writing this reply), I make my recommendation even stronger:

**The manuscript should be rejected because (at least) the pressure calculations for the middle and deep "reservoirs" (and likely the associated thermometry as well, at least for the deep and likely also for the middle "reservoir") are based on a fundamentally invalid assumption and are, therefore, scientifically invalid in their current form. Even if the calculations were formally valid, the pressures would be based on an arbitrary, assumed (without constraints from data or theory) temperature-pressure path, i.e., again scientifically not constrained. As these invalid results form the very core of the study's main contribution – the "revised conceptual model" and the important conclusions resulting from it – the manuscript cannot be accepted.**

Notabene (after the experience how the previous review was perceived): I do acknowledge that a lot of honest effort and care were put into the underlying calculations in the book; nevertheless, I am convinced that a key problem escaped the authors' attention, which, unfortunately, renders the "saturation decompression path" invalid, no matter how much effort was put into the calculation concept and its execution and no matter how exciting the results might look to the authors. That I use frank words should not be confused with disrespect but as a way to keep the discussion clear about the problem and keep it focused on the science; and not be shifted again to a scientifically meaningless competition on how experienced and qualified either reviewer or authors might be in different subareas of magmatic-hydrothermal systems research or in applying the respective community's wording sufficiently accurately to be accepted (on that: I regret that the omission of "thermometry" in "new gas geochemical *thermometry* data" escaped my attention when proof-reading my review).

The first 25 lines of the Additional Review section [from "*Now, although my duty…*" to "*… escaped my attention when proof-reading my review).*"] are very verbose and do not concern the manuscript review. In fact, the reviewer shifts the focus from the manuscript (as it should be) to the book, looking for deficiencies in the book itself. From the unsolicited and unrespectful review of the book (see our previous comment), the reviewer seems to have realized that it is not possible to demonstrate that fluids follow the "saturation decompression path" on which geothermometers and geobarometers rely on and therefore the manuscript is not publishable. Regardless of the fact that the reviewer could have reached this conclusion based on the manuscript alone and that there was no need to read the book, we totally agree with the reviewer that it is impossible to demonstrate that the Solfatara fluids follow the saturation decompression path for a brine with a certain NaCl content rather than a different decompression path. Furthermore, we emphasize that the decompression path of the Solfatara fluids is totally unknown, as it is unknown for all fluids and all geothermometers, not only the gas geothermometers, but also the water geothermometers. Therefore, following the line of reasoning of the reviewer, one reaches the conclusion that all fluid geothermometers are based on wrong assumptions and all geothermometric results for all hydrothermal-magmatic systems

worldwide are wrong. Furthermore, since the databases of computer codes such as PHREEQC, EQ3/6, WATCH, SOLVEQ and others assume saturation conditions for pure water, also multicomponent geothermometry provides wrong results, again following the line of reasoning of the reviewer. In a nutshell, all the geochemical literature that deals with fluid geothermometry would be wrong and useless.

Alternatively, isn't it perhaps wrong to adopt the reviewer's point of view?

It is true that all geothermometers are based on hypotheses, including the decompression path during the ascent of the Solfatara fluids, but it is also true that in the book (not in the paper, for reasons of space) we have considered not only the saturation decompression paths but also other possible decompression paths, namely, the isenthalpic decompression path and the linear P-T decompression path. The CO-, CH$_4$- and H$_2$S equilibrium temperatures and total fluid pressures (obtained considering the saturation decompression paths) were selected in the paper to simplify the discussion and without pretending that the selected values are the indisputable truth. Nevertheless, if a different series of CO-, CH$_4$- and H$_2$S equilibrium temperatures and total fluid pressures (obtained considering a different decompression path) is selected, the time-trends of nearly constant CO-equilibrium temperature and total fluid pressure and increasing CH$_4$- and H$_2$S temperature and total fluid pressure are still observed with some limited differences. We will add these considerations to the revised manuscript.

**Now to the core problem**. The calculations of gas equilibration temperatures involve thermodynamics-based corrections for strong pressure effects. The pressures and resulting temperatures rely on an ***arbitrary, assumed temperature-pressure path*** along which fluids would have to flow on their way from the magmatic source to the fumaroles 8 km above where the samples were taken. The study tries to pinpoint where on that generic path individual parts of the actual Solfatara system are located.

According to the book, this assumed "saturation decompression path" is an arbitrary selection. It takes the vapor saturation pressure for a constant composition liquid (21 wt.%) in the pure H$_2$O-NaCl binary and adding a simplistic pressure correction for the presence of CO$_2$ in the vapor phase (as a side remark I wonder how one could call that "in *equilibrium* with a brine"; but see discussion of CO$_2$ solubility below). I like to emphasize again that this is an *assumed, arbitrary* path and not supported by any data; that "Giggenbach did that as well in 1987" is not a good geology- or physics-based justification. Pressures for the "reservoirs" are then "computed" by taking the computed gas equilibration temperatures and locating them on that path (I assume this was an iterative procedure).

Above we have anticipated our answers to the general introductory considerations of the reviewer and we have little to add here apart from two reviewer's sentences.

The first sentence is '*that "Giggenbach did that as well in 1987" is not a good geology- or physics-based justification*'. It is not our intention to point out that what we did is correct because Giggenbach had already done it in 1987. Our intention is to give Caesar what belongs to Caesar and to give Giggenbach what belongs to Giggenbach, who had the great merit to understand the complexities of natural systems and to develop geochemical tools easy for everyone to use. Werner Giggenbach was an unsurpassed master in this aspect.

The second sentence is "*Pressures for the "reservoirs" are then "computed" by taking the computed gas equilibration temperatures and locating them on that path (I assume this was an iterative procedure).*" There is no iterative procedure in our calculations. This is obvious to anyone who opens the EXCEL spreadsheet, which is part of our paper as supplementary material. We wonder if the reviewer has opened our EXCEL spreadsheet because we have serious doubts about it based on this sentence. If so, it would be a very serious negligence of the reviewer. But since we are not certain, we do not want to create a case without foundation.

In contrast, the key problems raised by the reviewers are undermined by several, sometimes specious, inaccuracies.

In a nutshell, the key problems of that approach are (more, illustrated detail then further below):

• A vapor-saturated 21 wt% NaCl brine that the calculations are based on simply doesn't exist above ca. 590 °C (the critical temperature for that composition in the binary $H_2O$-NaCl system[1] [footnote 1: btw., Giggenbach (1987) also stated that the validity is limited to ca. 600°C]), so the calculation is meaningless for all inferred temperatures for the deep "reservoir" and for the hottest of the middle one.

Here the reviewer reinvents the wheel. In fact, at pages 290-292 of the book we wrote that the RWG and SS4 geoindicators for the saturation decompression path of Solfatara fluids involving a brine containing 21 wt% NaCl and the related vapor phase can be used up to 600 °C. The maximum SS4 computed temperature, that is, the maximum $CH_4$-equilibrium temperature, 645°C, is not much higher. In our manuscript we wrote clearly that the $CH_4$-equilibrium temperature and total fluid pressure refer to the intermediate ("*middle*" in the reviewer's words) reservoir. We do not understand why the reviewer writes that "*the calculation is meaningless for all inferred temperatures for the deep "reservoir"* because **the $CH_4$-equilibrium temperature does not apply to the deep reservoir**. This is an unacceptable reviewer's mistake.

• Instead, above that temperature, an $H_2O$-NaCl fluid on the two-phase liquid-vapor surface in that system is a vapor itself.

• As a major consequence, and as the critical line for 21wt% NaCl (with respect to water) in the ternary $H_2O$-NaCl-$CO_2$ system is nearly isothermal at those 590 °C: if $CO_2$ is present, the fluid at such high temperature and pressure conditions would be a homogeneous, "supercritical", dense vapor-1 btw., Giggenbach (1987) also stated that the validity is limited to ca. 600°C like fluid for which no unique P-T coordinates can be constrained without serious further assumptions.

It is interesting to note that the reviewer agrees with Giggenbach (1987) when it is convenient and vice versa. Well, we all agree on the maximum temperature at which the geoindicators for the saturation decompression path of Solfatara fluids involving a brine containing 21 wt% NaCl and the related vapor phase can be used, 600 °C.

• For this dense vapor-like fluid, due to the presence of significant NaCl and even if the pressures were correct, the fugacity coefficients calculated by the authors via a Peng-Robinson equation of state for a system without NaCl likely are in error (and, as far as I recall, standard P-R equations of state are not very well suited to include electrolytes) and so would be the gas thermometry.

This critique of the reviewer is totally wrong because there are many papers dealing with the use of the Peng-Robinson EOS, with some adjustments/modifications, to model vapor + liquid equilibria in electrolyte solutions and specifically in systems including water, sodium chloride, and carbon dioxide (e.g., Nighswander et al. 1989; Kwak and Anderson 1991; Søreide and Whitson 1992; Sieder and Maurer 2004; Baseri and Lotfollahi 2011; Hou et al. 2013; Appelo 2015; Li et al. 2015; Zuo et al. 2024). In particular, Appelo (2015) used the Peng–Robinson EOS to compute reliable fugacity coefficients for $CO_2$. He noted that the ion interaction parameters given by Harvie et al. (1984) for $CO_2$ at 25 °C are valid for calculating the $CO_2$ solubility at high temperatures, pressures and salinities.

Furthermore, use of the Peng-Robinson EOS is totally justified because we obtained comparable values, for the fugacity coefficients of $CO_2$ and $H_2O$, utilizing the GERG-2008 EOS (Kunz and Wagner 2012) and the EOS of Gallagher et al. (1993). To the best of our knowledge, fugacity coefficients were not considered in gas geo-indicators proposed so far in the scientific literature, although use of fugacity coefficients is absolutely necessary at the high temperatures and pressures of $CH_4$ and $H_2S$ equilibration. To be noted also that, irrespective of the decompression path, the fugacity coefficients of non-polar gases (i.e., $CO_2$, CO, $CH_4$, $H_2$, and $H_2S$) increase with increasing P, T, deviating gradually from unity, whereas the fugacity coefficient of $H_2O$ decreases with increasing P, T, departing progressively from one. The ensuing practical implication is that the

analytical mole fraction ratios of non-polar gases (e.g., CO/CO$_2$, CH$_4$/CO$_2$, and H$_2$S/CO$_2$) may be utilized in geothermometric-geobarometric functions without incurring excessive errors, whereas use of the analytical H$_2$/H$_2$O mole fraction ratio leads to significant errors in computed equilibrium temperatures and pressures. **Summing up, the errors in fugacity coefficients and gas thermometry the reviewer refers to are totally non-existent while serious errors are committed if the fugacity coefficients are not considered**.

• These points render the temperature-pressure calculations for the deep "reservoir" in their present form invalid. How wrong they are in terms of numbers cannot be estimated in a simple way.

This critique of the reviewer is also not justified because (1) the T-P conditions of the deep reservoir are described by the H$_2$S equilibrium temperature and total fluid pressure and (2) the H$_2$S equilibrium temperature and total fluid pressure were computed considering the decompression path of Solfatara fluids involving a brine containing **33.5 wt% NaCl (not 21 wt% NaCl** as erroneously considered by the reviewer**)** and the related vapor phase. Both things are written clearly in our manuscript. These functions can be used up a maximum pressure of 3627 bar and a maximum temperature of 1000 °C (see pages 335 of the book). In practice, the maximum H$_2$S equilibrium temperature and total fluid pressure computed for the Solfatara fluids, 1087°C and 3408 bar, are slightly higher and somewhat lower of these applicability thresholds, respectively. **So, there is nothing wrong in the temperature-pressure calculations for the deep reservoir as speciously claimed by the reviewer**.

• Moreover, the approximation of a carbonic vapor phase pressure in "equilibrium" with a vapor-saturated H$_2$O-NaCl liquid is only acceptable if the mutual solubilities of CO$_2$ in the aqueous solution and of water in the carbonic vapor phase are small such that the saturation condition spreads only over a small pressure interval and remains close to the saturation line in the aqueous solution binary.

These statements of the reviewer are unclear and contain some inaccuracies. The core or what we are talking about is if it is acceptable or not to refer to the saturation line of the aqueous solution binary H$_2$O-NaCl for a ternary H$_2$O-NaCl-CO$_2$. As shown in Figure 5 of Giggenbach (1987), this approximation is acceptable if the concentration of CO$_2$ in the brine and the concentration of NaCl in the vapor phase are both sufficiently low and not when the *mutual solubilities of CO$_2$ in the aqueous solution and of water in the carbonic vapor phase are small* as erroneously claimed by the reviewer (perhaps a typo?). On this figure, in the left-hand diagram, we have reported both the logarithm of the fugacity of water for a brine containing 21 wt.% NaCl in equilibrium with a vapor phase with a mole fraction of CO$_2$ equal to 0.05 (diamonds and line of blue color) and 0.40 (squares and line of red color), corresponding to the minimum and maximum mole fraction of CO$_2$ of interest for the Solfatara fluids. In the right-hand diagram, we have reported both the logarithm of the fugacity of water for a brine containing 33.5 wt.% NaCl in equilibrium with a vapor phase with a CO$_2$ molar fraction equal to 0.05 (upward pointing triangles and line of sky-blue color) and 0.40 (downward pointing triangles and line of magenta color). In both diagrams, the two lines with symbols are close to each other and their acceptably good linearity holds true up to 600°C for the brine containing 21 wt.% NaCl and up to 1000°C for the brine containing 33.5 wt.% NaCl. These lines are not positioned exactly between the vapor and brine envelops reported in Figure 5 of Giggenbach (1987) because Giggenbach used the data of Sourirajan and Kennedy (1962) for the system H$_2$O-NaCl, whereas we used the data of Tanger and Pitzer (1989) for the brine containing 21 wt% NaCl and the data of Driesner and Heinrich (2007) for the brine containing 33.5 wt% NaCl. Moreover, Giggenbach took the fugacity coefficients, presumably of pure gases, from Ryzhenko and Volkov (1971) and Ryzhenko and Malinin (1971) and stated that "*for a separate gas phase containing water vapor as the major component, $x_{H_2O} > 0.8$, maximum pressures [not given] are controlled by the formation of an aqueous phase. Under these circumstances the isomolar ratios of activity coefficients $\gamma_{SO_2}/\gamma_{H_2S}, \gamma_{CO}/\gamma_{CO_2}, \gamma_{CO_2}/\gamma_{CH_4}, \gamma_{N_2} \cdot \gamma_{H_2O}/\gamma_{NH_3}^2, and \gamma_{H_2}/\gamma_{H_2O}$ [γ's are actually fugacity coefficients] never deviate much from unity, and uncorrected mole-ratios may be used without incurring excessive errors*".

In contrast, we computed the fugacity coefficients of gas species in gas mixtures using the Peng-Robinson EOS.

[Figure]

In spite of these minor deviations between our $f_{H2O} - 1/T(K)$ relations and the vapor and brine envelops reported in Figure 5 of Giggenbach (1987), the clear description given by Giggenbach in his 1987 paper is still valid and is reported here: "***Three fluid stability regions can be distinguished: (1) a single liquid (brine) at comparatively low temperatures and high pressures; with increasing temperature or decreasing pressure a boundary ("liquidus") is reached to a region (2) where a vapor phase will coexist with a saline brine; further increase in temperature, or reduction in pressure, will lead to complete evaporation of increasingly saline brines and to a boundary ("vaporus") delineating the stability region (3) of a single vapor phase (in the presence of solid NaCl).***" In the Solfatara hydrothermal-magmatic system, the most probable condition is the coexistence of a vapor phase with a saline brine, because the occurrence of the other possible conditions is highly unlikely. In fact:

- On the one hand, the occurrence of a single liquid (brine) at comparatively low temperatures and high pressures is at variance with the huge amount of heat released from the magma batch and transferred to the overlying hydrothermal part of the system.
- On the other hand, a single vapor phase coexisting with solid NaCl might occur in depressurized vapor-cored magmatic systems (Reyes et al. 1993), such as Vulcano Island, Italy (Cioni and D'Amore 1984), and many systems of Indonesia (Abiyudo et al. 2016) and The Philippines (Reyes et al. 1993; Ramos-Candelaria et al. 1995; Apuada and Sigurjonsson 2008), but it is at variance with the current pressurization and related ground uplift of the Solfatara hydrothermal-magmatic system.

This holds true for typical geothermal systems at less than ca. 300ish °C and has, therefore, indeed be applied to such systems. As soon as the mutual solubilities are non-negligible (starting at T > ca. 300ish °C, some people might claim 350ish °C, it really doesn't matter much) this is not possible anymore as "saturation pressure" of the liquid is not uniquely defined and phase compositions will change with phase proportions for a given bulk composition.

Rather, the liquid saturation pressure will now also be a function of the additional $CO_2$ content of liquid, i.e., there is a whole saturation surface over very wide ranges of pressure and for temperatures up to the near-isothermal (the above 590 °C) critical curve of the pseudo-binary "21 wt% aqueous NaCl solution + $CO_2$" instead of a single saturation line.

These theoretical considerations of the reviewer are correct but they do not apply to the Solfatara hydrothermal-magmatic system, in its current state, as discussed above. Things might have been different in the past and might be different in the future, as discussed in Chapter 11 of our book.

Side remark: so much about the authors' statement "there is no need for an *"adequate solubility model for the gases in such a brine"."*

**We did not write this wrong sentence, neither in the paper nor in the book.** At pages 122-123 of the book, we wrote that "... *the binary $H_2O$-NaCl system is considered here, instead of the ternary $H_2O$-NaCl-$CO_2$ system, to describe brine-vapor coexistence both: (1) to avoid complicating the calculations too much and (2) because of the incomplete knowledge of the P-V-T-X properties of the $H_2O$-NaCl-$CO_2$ system, which is largely based on the experimental work by Gehrig (1980) and the P-V-T-X data obtained through the synthetic fluid inclusion technique (e.g., Kotelnikov and Kotelnikova 1990; Frantz et al. 1992; Johnson 1992; Shmulovich and Graham 1999; Schmidt and Bodnar 2000).*

*In addition to the determinations relevant for the binary $H_2O$-$CO_2$ system (see Sect. 2.1), Gehrig (1980) measured the molar volume and defined the immiscibility boundaries for the ($H_2O$ +6 wt.% NaCl)-$CO_2$ pseudobinary system at pressures from 0 to 3000 bar and temperatures from 200 to 560 °C using a high-pressure variable volume autoclave. The data of Gehrig (1980) represent the main experimental foundations of the modified Redlich–Kwong EOS of Bowers and Helgeson (1983) as well as of the EOS of Duan et al. (1995), both for $H_2O$-$CO_2$-NaCl fluids. Bowers and Helgeson (1983) computed fugacity coefficients of $H_2O$ and $CO_2$ at temperatures of 400, 450, 500, 550, and 600 °C and pressures of 500, 1000, and 2000 bar and used these fugacity coefficients together with solubility data to establish the compositions of the coexisting immiscible phases. However, the EOS of Bowers and Helgeson (1983) was questioned by Duan et al. (1995). According to Duan et al. (1995), their EOS predicts P–V-T-X data, immiscibility/phase equilibria, solubilities, and activities with an accuracy similar to that of the experimental data at temperatures from 300 to ~1000 °C and pressures from 0 to 6000 bar for NaCl concentrations to ~30 wt% (relative to NaCl + $H_2O$) and to ~50 wt% with lower accuracy. However, the EOS of Duan et al. (1995) was considered poorly reliable by other authors (e.g., Schmidt and Bodnar 2000).*

*Given this situation, the binary $H_2O$-NaCl system was considered in this work, instead of the ternary $H_2O$-NaCl-$CO_2$ system, to describe the equilibrium coexistence of the vapor phase with a brine containing either 21 wt% NaCl (data from Tanger and Pitzer 1989) or 33.5 wt% NaCl (data from Driesner and Heinrich 2007)."*

**Our opinion on this topic is exactly the opposite of what the reviewer attributed speciously to us, in that, we believe that there is a need for an EOS able to predict P-V-T-X data, immiscibility/phase equilibria, solubilities, and activities in the ternary $H_2O$-NaCl-$CO_2$ system.**

• Given the temperature range mentioned in the previous point, also both the temperatures and pressures in the middle "reservoir" are likely highly questionable.

As explained above, there is nothing questionable about the temperatures and pressures in the intermediate (middle) reservoir we estimated for the saturation decompression paths and other possible decompression paths of Solfatara fluids.

Now this in some more detail, illustrated by published figures of the respective phase diagrams.

Let's start with the issue of 21% vapor-saturated brine with the diagram by Driesner and Heinrich (GCA, 2007). I have highlighted the 20 wt.% curve (let's stay with that as a convenient proxy for the author's choice for rest of the discussion) in yellow and one can see how it intersects the critical curve at ca. 590 °C, i.e., at lower temperatures it is on the liquid side of the twophase vapor+liquid surface, at higher temperatures on the vapor side. So, there is no such thing as a 20 wt% saturated liquid at temperatures higher than this. One might

now argue "ok, let's simply take a higher salinity such as 33.5% used elsewhere in the manuscript and the problem is gone".

[Figure]

Well, before we come to that point let's first look at a relevant P-T projection of parts of the $H_2O$-NaCl-$CO_2$ phase diagram, taken from Schmidt and Bodnar (GCA, 2000). The most relevant feature in that diagram is the near-vertical line labeled "20 wt% NaCl". This is the critical line for the pseudo-binary "20 wt% NaCl/80wt% water + $CO_2$". At temperatures higher than that curve any fluid in the ternary $H_2O$-NaCl-$CO_2$ system that has a 20:80 wt% ratio of NaCl and water and has some $CO_2$ content will exist as a single-phase, "supercritical" fluid (highlighted yellow, extent to bottom not well known and not well understood). Red added curve is the approximate location of 20 wt% NaCl on V+L surface (solid curve: liquid, dashed curve: vapor), added blue curve is threephase vapor+liquid+halite, both in the $H_2O$-NaCl binary. In that binary, below the dashed red curve, there would be liquid with >20% NaCl in coexistence with a vapor <20% NaCl but no fluid phase with 20wt% can exist there. What happens below the red curve in the ternary is not well known, in particular if and where it comes to halite saturation as that is now a divariant surface in T-P rather than a univariant curve and may actually start at much higher pressures than in the binary (see, e.g., Anovitz et al., GCA, 2004).

*I think it is obvious from the above that the authors' pressure calculation method has no foundation at temperatures above ca. 590 °C and that, therefore, **the pressure calculations for the deep reservoir are simply invalid**. An additional, very important point now is that, at the higher temperature conditions, appreciable amounts of NaCl would be present in the vapor phase (in different concentrations for different T-P) rendering the used Peng-Robinson e.o.s. version non-applicable. For example, already at 500°C and 500 bar, Anovitz et al. (GCA, 2004) found up to several mole percent NaCl in the vapor and this should be expected to increase with higher temperature for the simple thermodynamic requirement that it needs to converge with the aqueous liquid's concentration towards the upper critical end curve. **Therefore, even the thermometry is seriously questionable for temperatures of the late part of the time series for the middle reservoir and for all temperatures of the deep reservoir.***

[Figure]

The full implications of the diagram above are nontrivial to comprehend as it is a 2D projection of elements of a 4D diagram. For me, the most intuitive and plausible image emerges when taking the $H_2O$-$CO_2$ phase diagram in 3D as a reference, e.g., from Diamond (Lithos, 2001). What is labelled there "Upper critical curve of binary" is the "0 wt.% NaCl curve" in the previous diagram, i.e., this diagram could roughly be imagined as an extension of the previous diagram to the left with $X_{CO2}$ as an axis going into the diagram. The "Upper critical curve of binary" limits the chimney-like carbonic-aqueous immiscibility region to high temperatures. We could imagine this as a crude guide for how the "immiscibility chimney" expands and shifts to higher temperature when moving to the pseudo-binary "20 wt% NaCl/80wt% water + $CO_2$" (most elements in the previous diagram): also there, it'd be a "chimney" with significant mutual solubility on the steep saturation surfaces (again: not lines) on the aqueous and carbonic sides and limited to high temperatures by the "20 wt% NaCl" line in the second diagram.

Now, this also explains why choosing saturation of a 33.5% "brine" would not do the job: although the upper critical line of that pseudo-binary would be at higher temperatures (about 1000°C) the mutual solubility would still eliminate the choice of a single saturation pressure and the saturation surfaces would still be steep, i.e., saturation pressure ranges are very large. Even worse (see also Anovitz et al., 2004, their Figure 4): at a given temperature and pressure varying $X_{CO2}$ of the saturated vapor will make the NaCl content of the saturated liquid change significantly. ***So, the authors barometry approach also breaks down at temperatures relevant for the middle reservoir.***

Unfortunately, for the lack of more experimental data, we don't know to how low temperatures this may be the case but I think it would be good scientific practice to cautiously assume that Anovitz' 500°C is not a fortuitous hit of the lower temperature limit of the problem. The community has not yet fully explored and understood all the phase relations in the $H_2O$-NaCl-$CO_2$ ternary, namely on the carbonic -side or at the low pressure end. Yet, with the above (which necessarily was highly simplified and incomplete) I think we have understood enough that the negative assessment about the (in)validity of the authors' approach is robust.

In terms of the manuscript's main results this means graphically:

[Figure]

All this long discussion adds nothing to what the reviewer already pointed out above. The gist of the discussion is that the relation for a brine containing 21 wt% NaCl and the related vapor phase can be used up to 590-600°C whereas the relation for a brine containing 33.5 wt% NaCl and the related vapor phase can be used up to 1000°C, with more uncertainties. Anyhow, the $CH_4$ equilibrium temperature which refers to the intermediate reservoir are generally lower than the 600°C threshold and moderately higher in few cases while the $H_2S$ equilibrium temperature which refers to the deep reservoir are generally lower than the 1000°C threshold and moderately higher for few samples. **Again, the reviewer confuses the two temperatures and the two reservoirs.** Nothing new apart from the two diagrams defaced by childish graffiti and scribbles which are ethically questionable.

To add on this: the authors could have suspected a serious problem already from their pressure diagram. Fluid pressures as high as 2.4 times lithostatic (or, in absolute values overpressured by up to 1800 bar) and fluid pressure gradients similarly excessive, in particularly in the hot, ductile regime, would be considered unrealistic by many (if not most) people dealing with such problems for reasons laid out, e.g., by Cox (Geofluids, 2010) or discussed in the Weis et al. (2012) paper (or follow-ups on that) that the authors considered not relevant in their rebuttal. In essence, (ductile) rocks cannot sustain such high fluid overpressure, the values are very unrealistic. Since a lot of the implications of the model hinge on that, the author may want to acquaint themselves more with rock failure related to fluid pressure (the Cox paper is a good introduction).

Although in the reviewer's words we clearly perceive a certain ethically-inappropriate sarcasm and a lack of respect towards us, we appreciate the reviewer's suggestions to acquaint ourselves more with rock failure related to fluid pressure. We believe there is always something to learn. Nevertheless, we are not totally ignorant on this subject because we have gained some experience by working on the brittle-ductile transition in the Tuscan geothermal systems (e.g., Marini and Manzella 2005). As a matter of fact, the knowledge of these systems gained through multidisciplinary surface exploration and drilling of geothermal wells to depths of almost 4 km allows one to understand the rheological behavior of rocks quite realistically.

Now, to be "constructive" after all this: The phase relations discussed above also propose a possible way how the "middle reservoir" can indeed be the source of bradyseism. If the fluid released at depth were a "supercritical" $H_2O$-$NaCl$-$CO_2$ mixture then upon decompression it may hit the "immiscibility chimney" displayed in the $CO_2$-related phase diagrams above. For a suitable parameter combination this should result in sufficient pressurization.

Similarly, the overpressure problem goes back to the assumed temperature-pressure "saturation decompression path" and the fact that it is simply arbitrary, not founded on any data or defendable theory, and leads to too high pressures. Given the above discussions on the phase diagram it seems likely to me that

a plausible decompression path can be constructed that implies lower, ore [more] realistic pressures at depth. Then, "lithostatic plus a defendable overpressure" in the deep "reservoir" would be a good starting point for revising the decompression path (from the source upward rather than vice versa) of the "revised conceptual model".

We thank the reviewer for this suggestion but we prefer to keep our interpretation.

*Up to here, I consider this reply as essential information for the editor. I hope the explanations are not "too nerdy" to be evaluated for the decision process (if they are, please request an explanation in simpler words) and I hope that also the authors can accept my criticism given their own statement "every author should be allowed to publish her/his results, provided that they have scientific validity and are not affected by errors of any kind". I think I clearly showed above that the study contains serious problems of both kinds.*

We think we clearly showed above that the review contains serious problems of scientific validity and is affected by several errors.

In the following I take the freedom to share some reflections with the authors as on what other aspects I think they may want to consider and/or re-think when re-visiting their model. This will also reply to some selected parts of their rebuttal. I will not answer to all points as I think that the discussion has already reached a point of getting lost in unnecessary back-and-forth. I don't claim to own the truth here but I would expect my thoughts to be considered a bit more seriously (1) as possibly valuable when revisiting the model and (2) for what to pay attention to beyond the gas chemistry point of view.

We are open to consider respectful and scientifically valid critiques. However, point (2) written above ("*for what to pay attention to beyond the gas chemistry point of view*") is not correct because we considered different geoscientific data, whereas the previous part of the review was focused on the $H_2O$-NaCl-$CO_2$ ternary and related binaries almost exclusively, apart from a few considerations on rock rheology (see above).

*Various points*
* * *
• As an add-on to the discussion above, I would like to suggest to the authors to also think a "decompression path" indeed from the deep source upward and not top down from the surface (which is rather a "pressurization path") although the gas data were collected there. Phase changes will modify $CO_2$-$H_2O$ ratios upon ascent and the degree and cause of this modification cannot be quantified in the top down way; e.g., for your shallow isenthalpic paths, for example, when going down you hit the $H_2O$-$CO_2$ saturation on the (carbonic) vapor side (on a rather flat surface that forms the "chimney's" bottom; such vapor can be equally well result from temporary saturation and condensing out some minor liquid (due to the "bulge" in the vapor enthalpy curve as you discussed) or from the (boiling-like) exsolution of a mass-wise minor vapor phase from an $CO_2$-poor but mass-wise dominant aqueous liquid. What happens below that saturation depth is, therefore, ambiguous and cannot be constrained from surface data as presented. Also, for this problem, I missed a clear statement of what your assumptions were for the top down approach regarding the fluid phase evolution with depth. Such a statement would have allowed readers/reviewers to test your hypothesis; leaving it out – or not at least discussing this / not formulating a hypothesis – is not good practice.

Since the gas data were collected at the surface, we think that the top-down approach (using the reviewer's words) is more suitable than the opposite bottom-up approach. If one wants to consider the effects of partial water condensation (as done in some papers of Chiodini and coworkers), it is necessary to introduce an assumption which cannot be tested. So, we prefer to use the analytical data without any correction to avoid to complicate the problem rather than to simplify it. Furthermore, in the book we considered also the chemical kinetics of the reactions involving $H_2O$, $CO_2$, CO, $CH_4$ and $H_2$, which has been the subject of several studies, and we applied a simplified reaction path model, simulating the heating of the Solfatara fluids

collected at the surface. The equilibrium temperatures computed by the reaction path model compare with those given by simple gas geothermometers, within acceptable differences. Thus, gas geothermometers work well, at least for the intermediate reservoir.

• Again on top down vs. bottom up: that was also the main reason for recommending Einaudi et al. and others for sulfur or Weis et al. for thermo-hydrology. I did not assume that the former were up to date with respect to the latest in volcanic gas chemistry but these papers think the chemical (or fluid flow, resepctively) process from the source to the surface and this clearly adds value. Namely, Einaudi et al. highlight that the fluid passes through different redox and fS2 conditions along its path as exemplified by successive mineral assemblages observed, which will, among others, also alter sulfur speciation and fugacity/concentration (and therefore the values of your H2S thermometer) on the way up. To me this looks geologically and geochemically much more logically and advanced than the assumption of a single mineral-fluid reaction fixing it right away at 7 or so km depth. Whether the Weis model was inspired by porphyry deposits doesn't play a role; to me, it models a generic magmatic fluid release process that was then interpreted by those authors for its relevance for pophyry-Cu deposits.

• A bit more about the reactive path of sulfur: for the deep parts, the main window of action for $SO_2 + H_2O$ reacting are believed to be below 500ish °C or so (gas redox buffer followed by disproportionation along cooling path, if I remember correctly). This is at lower temperatures than your $H_2S$ thermometer equilibration and should therefore be assessed for a possible impact on $H_2S$ concentrations on the way to the surface.

We have nothing to add to our previous rebuttals to these reviewer's considerations apart from a comment on H2S, in that possible reactions causing significant changes of this gas species were considered in the book. The fugacity of gaseous $S_2$, $SO_2$ and COS were computed as a function of the $CH_4$ equilibrium temperature (called SS4 equilibrium temperature in the book) and resulted to be negligible. We have also considered the pyrite-pyrrhotite, pyrite-fayalite-quartz, pyrite-magnetite, and pyrite-hematite equilibria, but the computed equilibrium temperatures have no physical significance, suggesting that the $H_2S$ concentration of Solfatara fluids is controlled by other reactions. Considering the presence of carbonate rocks at depths from ∼ 4 km to ∼7.5 km, we assumed that this reaction is:

$CaSO_{4(s)} + CO_2 + 4 H_2 = CaCO_{3(s)} + H_2S + 3 H_2O$.

• Fournier vs. Weis et al.: the Weis model replaced the Fournier concept in that it explains the lithostatic to hydrostatic transition as a natural consequence of degassing magmatic fluids having the dual role of heating the rock overlying the magma to ductile temperatures and impermeable behaviour and, in turn, of transiently breaking those heated rocks due to pressure build-up to allow the temporary release of fluids; this magmatic-hydrothermal domain than naturally transitions into a classical geothermal system further up; there is no need for self-sealing by silica, just heat + fluids do the job already, matching many features of natural systems (fossil or active). Let Occam's razor do its job here.

We prefer the Fournier model also considering the occurrence of quartz in the deepest section of geothermal well San Vito-1.

• Lupi/Weis: well, that was quite a cheap trick referring to what I did not point at (the trigger mechanism, which indeed can be questioned) and then trying to make me look ridiculous by criticizing that. However, I appreciate that you didn't fully loose humor over my review. I was pointing at the overpressure waves rising in the magmatic-hydrothermal plume and these – if I remember that correctly from in-depth discussions with Weis – happen on time scales of years or 10s of years; i.e., they are highly relevant. Furthermore, you could learn from Weis et al. how $H_2O$-NaCl phase relations (unfortunately, no $CO_2$) evolve with space and time in such a system to come back to the main problem of your calculations.

It was not our intention to do cheap tricks and make ridiculous the reviewer. We are interested in scientific aspects only. We are convinced that the overpressure waves rising in the magmatic-hydrothermal plume does not provide a reasonable explanation of the on-going unrest phenomena at Campi Flegrei.

• It were these last few points that made me make the comment on "apparently from a volcanology background" as the way the manuscript is written it reads like a naive "what comes out of the fumaroles is what is at depth". There is nothing about "second class scientists" implied but rather should highlight the impression of a surface data-biased view on fluid processes in the deep parts of magmatic-hydrothermal systems (btw. understood as systems dominated by magmatic fluids). A lot of valuable information on the latter is available (mostly from the economic geology community) that could have informed the conceptual model design with quite some advantage but was not considered. In hindsight, I admit that the statement could be misinterpreted and hope these remarks clarify that.

We thank the reviewer for clarifying the comment "*the authors (apparently coming from a volcanology background)*" in the previous review.

• Convective zero temperature & zero gradient pressure profiles: another one that was apparently intended to make me look like a beginner by referring to an introductory book. My point was that your "convective" vertical temperature profiles in the reservoirs are an ad hoc invention based on no data and should, therefore be declared as such or be justified. BTW: zero pressure gradient vaporstatic columns would not convect. Regarding that zero pressure gradient, for the fun of it, let's take you own data, for example the Oct10-Jun12 $H_2S$ equilibration conditions: a pure water vapor at 830 °C and 2157 bar would have a density of 432 kg/m$^3$. In the $CH^4$ equilibration reservoir for the same period one would have 412 kg/m$^2$ [kg/m$^3$]. This is >0.4 times a cold hydrostatic gradient, far away from zero gradient. So, no point to make me look ridiculous when your own data proof you wrong. In my group it is a routine process to perform such obvious checks before adding conceptual figures to a publication.

First of all, we did not intend to make the reviewer look like a beginner by referring to the textbook by Malcolm A. Grant and Paul F. Bixley (2011) Geothermal Reservoir Engineering, Second Edition; we referred to this textbook because it is the most suitable reference in geothermal reservoir engineering. Second, the vapor-static column convects as discussed in the papers quoted in our previous reply as well as in the textbook by Malcolm A. Grant and Paul F. Bixley, which evidently the reviewer ignores. Third, the zero-pressure gradient is typical of vapor-dominated geothermal systems at temperature and pressure close to the point of maximum enthalpy of water vapor (235°C, 30.6 bar). In contrast, the intermediate and deep reservoirs of the Solfatara hydrothermal-magmatic system are generally at temperatures and pressures much higher and have densities relatively high making the pressure gradient in the intermediate and deep reservoirs higher than zero. The reviewer is absolutely right on this point. Nevertheless, it is not correct to consider the densities of pure water vapor, as done by the reviewer, due to the presence of significant gas amounts, mainly $CO_2$, in the gas mixtures of interest. We have revised Figure 6 (which will be presented in the revised version of the manuscript), showing the time changes, between September 1984 and January 2022, of the total fluid pressure vs. depth profile along a hypothetical borehole drilled in the Solfatara crater. For the vapor in equilibrium coexistence with a brine containing 21 wt% NaCl, we have considered the molar volumes reported in Tables 1, 2, 3, 4, and 5 of Chapter 9 of Marini et al. (2022), whereas for the vapor in equilibrium coexistence with a brine containing 33.5 wt% NaCl, we have considered the molar volumes reported in Tables 9, 10, 11, 12, and 13 of Chapter 9 of Marini et al. (2022). The obtained log-densities in log-kg/m$^3$ were then fitted against $X_{CO2}$ and temperature to derive simple polynomials that were used to calculate the density of the gas mixtures in the intermediate and deep reservoirs, which were assumed to be constant to avoid complicating too much the calculations. Knowing the density, $\rho$ (in kg/m$^3$), the total fluid pressure at the bottom of the two reservoirs of interest, $P_B$ (in bar), was computed, to a first approximation, by the relation:

$P_B = P_T + 1000*\rho*g*0.00001$

where g = 9.80665 m/s$^2$ is the conventional standard value of the gravity acceleration. Again, we do not intend to make the reviewer look ridiculous, but the data we used are not wrong. Our previous Figure 6 had minor inaccuracies which were adjusted as explained above. If the reviewer intends to build a case on these minor inaccuracies, we cannot stop the reviewer.

We are pleased to know that, in the reviewer's "*group, it is a routine process to perform such obvious checks before adding conceptual figures to a publication*", but these checks should be done using the proper data, not using unproperly selected data, that is, neglecting the presence of gases and referring to pure water. For instance, the density value of 431.61 kg/m$^3$ at 830 °C and 2157 bar and the density value of 412.12 kg/m$^3$ at 468°C and 582 bar pertain both to **supercritical pure water**, while we are dealing with a vapor in equilibrium with a brine containing 33.5 wt% NaCl and a vapor in equilibrium with a brine containing 21 wt% NaCl, whose density values are 430.35 and 289.39 kg/m$^3$, respectively, according to our approach. Even worse, if one considers the September 1984 – April 1987 period, with 264°C, 56.7 bar in the intermediate reservoir and 667°C, 1401 bar in the deep reservoir, the computed density values for the pure water system, 778.18 and 416.23 kg/m$^3$, respectively, refers to **pure liquid water** [sic!] and again to **supercritical pure water**, respectively. In contrast, the density values are 32.95 and 383.71 kg/m$^3$, respectively, according to our approach.

• "inferred geology". Let me cite the book, page 40: "In particular, according to Zollo et al. (2008), the **inferred** schematic stratigraphy comprises, from top to bottom (Fig. 10):". So, please don't bash me if I use your words.

In our previous rebuttal of the reviewer's critique on this point we wrote that "*There is no inferred geology in our work. Again, we considered only known geological data, both collected at the surface and obtained by deep geothermal boreholes, as well as known geophysical data to revise the "conceptual model" of the Solfatara magmatic-hydrothermal system.*" It is true that the stratigraphy was inferred by Zollo et al. (2008) based on the active seismic reflection data acquired by the SERAPIS survey. But this is how geophysics works. We did not add any inference.

• Calcite-H$_2$S: you are right, geology rules. I should have expressed much clearer that I was referring to the effect of reactive transport on sulfur content. There, I don't agree that the absolute concentrations don't matter because small concentrations may easier experience massive relative modification (reactive transport is always a competition between equilibrium constant and actual masses present) than bigger ones. Side remark, lines 69-71 of rebuttal: there is no such thing as a "strong acid" at those conditions; acids known to be strong at ambient conditions become weak in the low dielectric constant aqueous solvent at those conditions. As illustration: according to Supcrt, if one trusts it, the logK for $HSO_4^- = SO_4^{2-} + H^+$ in the temperature range of 500 to 800°C and densities from 0.4 to 0.6 g/cm3 is in the range of -8 to -11 ...

We agree that "*there is no such thing as a "strong acid at*" the conditions of the deep reservoir; "*acids known to be strong at ambient conditions become weak in the low dielectric constant aqueous solvent at*" the high temperature, high P conditions of the deep reservoir. We know very well the decreasing dissociation of acids with increasing temperature (e.g., Marini et al. 2003 a, b; Marini and Gambardella 2005). Side remark: the reviewer used again SUPCRT92, whose thermodynamic database refers to pure water, doing the same mistake done in density calculations.

• Finally, I think my doubts about the compatibility of "reservoir" and equilibration and a structural transport highway remain valid but there is no point discussing this further here. This applies also for all other points I may not have responded here, too.

We hope that our responses to the specious and unconstructive criticisms advanced by the reviewer will help to demonstrate the validity of our work, at least to those who read and comment on it with an open mind, not clouded by prejudice. Unlike the reviewer, who intends to end the discussion, we are willing to continue it as long as it is deemed necessary.

I would like to conclude with stating that I start to appreciate the egusphere discussion format. Although I was annoyed by the extra effort compared to what it could have been if the manuscript were properly prepared, I think such discussions can be very helpful and help bridging gaps between different communities, one of which became very obvious here.

We would like to emphasize once again that the reviewer used the EGUsphere public discussion format only for specious and unconstructive criticism not only on our manuscript, but also on our book. Of course, everyone is free to judge books as they prefer, but let us say that we perceived the reviewer's criticism of our book as a deliberate and unsolicited personal attack. The extra effort implicit in the review of our book is a rather obvious consequence of this choice of the reviewer, since the task assigned to the reviewer was only the scientific review of the manuscript. Finally, we agree with the reviewer on at least one thing, namely the usefulness of the EGUsphere public discussion format. Nevertheless, when one participates in a public discussion, she/he should have the good manners to reveal herself/himself and not to hide behind anonymity, even if it is permitted by the reviewer role.

**References quoted in our answers**

Abiyudo, R., Hadi, J., Cumming, W., & Marini, L. (2016). Conceptual model assessment of vapor core geothermal system for exploration. Mt. Bromo case study. In: Proceedings of the 4th Indonesia international geothermal convention and exhibition 2016, 10–12 August 2016, Cendrawasih Hall, Jakarta Convention Center, Indonesia.

Appelo, C. A. J. (2015). Principles, caveats and improvements in databases for calculating hydrogeochemical reactions in saline waters from 0 to 200 C and 1 to 1000 atm. *Applied Geochemistry*, *55*, 62-71.

Apuada N.A. & Sigurjonsson G.F. (2008). The geothermal potential of Biliran Island, Philippines. In: Proceedings of the 8th Asian geothermal symposium, 73–77.

Baseri, H., & Lotfollahi, M. N. (2011). Modification of Peng Robinson EOS for modelling (vapour+ liquid) equilibria with electrolyte solutions. *The Journal of Chemical Thermodynamics*, *43*(10), 1535-1540.

Bowers, T. S., & Helgeson, H. C. (1983). Calculation of the thermodynamic and geochemical consequences of nonideal mixing in the system H2O-CO2-NaCl on phase relations in geologic systems: Equation of state for $H_2O$-$CO_2$-NaCl fluids at high pressures and temperatures. *Geochimica et Cosmochimica Acta*, 47(7), 1247-1275.

Cioni, R., & d'Amore, F. (1984). A genetic model for the crater fumaroles of Vulcano Island (Sicily, Italy). *Geothermics*, *13*(4), 375-384.

Driesner, T., & Heinrich, C. A. (2007). The system $H_2O$–NaCl. Part I: Correlation formulae for phase relations in temperature–pressure–composition space from 0 to 1000 °C, 0 to 5000 bar, and 0 to 1 $X_{NaCl}$. *Geochimica et Cosmochimica Acta*, *71*(20), 4880-4901.

Duan, Z., Møller, N., Weare, J.H. (1995). Equation of state for the NaCl-$H_2O$-$CO_2$ system: prediction of phase equilibria and volumetric properties. *Geochimica et Cosmochimica Acta* , 59(14):2869–2882.

Frantz, J. D., Popp, R. K., & Hoering, T. C. (1992). The compositional limits of fluid immiscibility in the system $H_2O$-NaCl-$CO_2$ as determined with the use of synthetic fluid inclusions in conjunction with mass spectrometry. *Chemical Geology*, *98*(3-4), 237-255.

Gallagher, J. S., Crovetto, R., & Sengers, J. L. (1993). The thermodynamic behavior of the $CO_2$-$H_2O$ system from 400 to 1000 K, up to 100 MPa and 30% mole fraction of $CO_2$. *Journal of Physical and Chemical Reference Data*, *22*(2), 431-513.

Gehrig, M. (1980). Phasengleichgewichte und pVT-daten ternärer Mischungen aus Wasser, Kohlendioxid und Natriumchlorid bis 3 kbar und 550 °C. PhD dissertation, Univ. Karlsruhe, Hochschul Verlag, Freiburg.

Giggenbach, W. F. (1987). Redox processes governing the chemistry of fumarolic gas discharges from White Island, New Zealand. *Applied Geochemistry*, *2*(2), 143-161.

Harvie, C. E., Møller, N., & Weare, J. H. (1984). The prediction of mineral solubilities in natural waters: The Na-K-Mg-Ca-H-Cl-$SO_4$-OH-$HCO_3$-$CO_3$-$CO_2$-$H_2O$ system to high ionic strengths at 25 °C. *Geochimica et Cosmochimica Acta*, *48*(4), 723-751.

Hou, S. X., Maitland, G. C., & Trusler, J. M. (2013). Phase equilibria of ($CO_2$+ $H_2O$+ NaCl) and ($CO_2$+ $H_2O$+ KCl): Measurements and modeling. *The Journal of Supercritical Fluids*, *78*, 78-88.

Johnson, E. L. (1992). An assessment of the accuracy of isochore location techniques for $H_2O$-$CO_2$-NaCl fluids at granulite facies pressure-temperature conditions. *Geochimica et Cosmochimica Acta*, *56*(1), 295-302.

Kotelnikov, A. R., & Kotelnikova, Z. A. (1990). Experimental-study of phase state of the system $H_2O$-$CO_2$-NaCl by method of synthetic fluid inclusions in quartz. *Geokhimiya*, *1990*, 526-537.

Kunz, O., & Wagner, W. (2012). The GERG-2008 wide-range equation of state for natural gases and other mixtures: an expansion of GERG-2004. *Journal of Chemical & Engineering Data*, *57*(11), 3032-3091.

Kwak, C., & Anderson, T. F. (1991). Application of peng-Robinson equation to high-pressure aqueous systems containing gases and sodium chloride. *Korean Journal of Chemical Engineering*, *8*, 88-94.

Li, J., Wei, L., & Li, X. (2015). An improved cubic model for the mutual solubilities of $CO_2$–$CH_4$–$H_2S$–brine systems to high temperature, pressure and salinity. *Applied Geochemistry*, *54*, 1-12.

Marini, L., Vetuschi Zuccolini, M., & Saldi, G. (2003a) The bimodal pH distribution of volcanic lake waters. *Journal of Volcanology and Geothermal Research*, 121, 83-98.

Marini, L., Yock Fung, A., & Sanchez, E. (2003b) Use of reaction path modeling to identify the processes governing the generation of neutral Na-Cl and acidic Na-Cl-$SO_4$ deep geothermal liquids at Miravalles geothermal system, Costa Rica. *Journal of Volcanology and Geothermal Research*, 128, 363-387.

Marini, L., Gambardella ,B. (2005) Geochemical modeling of magmatic gas scrubbing. Annals of Geophysics, 48, 739-753.

Marini, L., & Manzella, A. (2005). Possible seismic signature of the α–β quartz transition in the lithosphere of Southern Tuscany (Italy). *Journal of Volcanology and Geothermal Research*, *148*(1-2), 81-97.

Marini, L., Principe, C., & Lelli, M. (2022). The Solfatara Magmatic-Hydrothermal System. Springer: Cham, Switzerland, 375 pp.

Nighswander, J. A., Kalogerakis, N., & Mehrotra, A. K. (1989). Solubilities of carbon dioxide in water and 1 wt.% sodium chloride solution at pressures up to 10 MPa and temperatures from 80 to 200. degree. C. *Journal of Chemical and Engineering Data*, *34*(3), 355-360.

Sieder, G., & Maurer, G. (2004). An extension of the Peng–Robinson equation of state for the correlation and prediction of high-pressure phase equilibrium in systems containing supercritical carbon dioxide and a salt. *Fluid Phase Equilibria*, *225*, 85-99.

Ramos-Candelaria, M., Sanchez, D.R., Salonga, N.D. (1995). Magmatic contributions to Philippine hydrothermal systems. In: Proceedings of the world geothermal congress, vol 2. Firenze, Italy, pp. 1337–1341.

Reyes, A. G., Giggenbach, W. F., Saleras, J. R., Salonga, N. D., & Vergara, M. C. (1993). Petrology and geochemistry of Alto Peak, a vapor-cored hydrothermal system, Leyte Province, Philippines. *Geothermics*, *22*(5-6), 479-519.

Ryzhenko, B. N., & Volkov, V.P. (1971). Fugacity coefficients of some gases in a broad range of temperatures and pressures. *Geochem. Inter.*, *8*, 468-481.

Ryzhenko, B. N., & Malinin, S. D. (1971). The fugacity rule for the systems $CO_2$-$H_2O$, $CO_2$-$CH_4$, $CO_2$-$N_2$, and $CO_2$-$H_2$. *Geochem. Int.*, *8*, 562-574.

Schmidt, C., & Bodnar, R. J. (2000). Synthetic fluid inclusions: XVI. PVTX properties in the system $H_2O$-NaCl-$CO_2$ at elevated temperatures, pressures, and salinities. *Geochimica et Cosmochimica Acta*, *64*(22), 3853-3869.

Shmulovich, K. I., & Graham, C. M. (1999). An experimental study of phase equilibria in the system $H_2O$-$CO_2$-NaCl at 800 °C and 9 kbar. *Contributions to Mineralogy and Petrology*, *136*(3), 247-257.

Sourirajan, S., & Kennedy, G.C. (1962). The system $H_2O$–NaCl at elevated temperatures and pressures: American Journal of Science, 260, 115–141.

Søreide, I., & Whitson, C. H. (1992). Peng-Robinson predictions for hydrocarbons, $CO_2$, $N_2$, and $H_2S$ with pure water and NaCl brine. *Fluid phase equilibria*, *77*, 217-240.

Tanger IV, J. C., & Pitzer, K. S. (1989). Thermodynamics of NaCl-$H_2O$: A new equation of state for the near-critical region and comparisons with other equations for adjoining regions. *Geochimica et Cosmochimica Acta*, *53*(5), 973-987.

Zollo, A., Maercklin, N., Vassallo, M., Dello Iacono, D., Virieux, J., & Gasparini, P. (2008). Seismic reflections reveal a massive melt layer feeding Campi Flegrei caldera. *Geophysical Research Letters*, *35*(12).

Zuo, Z., Lu, P., Zhu, C., & Ji, X. (2024). SAFT2 equation of state for the $CH_4$–$CO_2$–$H_2O$–NaCl quaternary system with applications to $CO_2$ storage in depleted gas reservoirs. *Chemical Geology*, *667*, 122328.

---

## Author Comment (AC3)

*Reply to the comments of reviewer n.2*

**PLEASE NOTE THAT THE REVIEWER'S WORDS ARE IN RED COLOR AND OUR REPLY IS IN BLACK COLOR.**

The research conducted by Marini et al. presents a novel interpretation of four decades of data from the Solfatara geochemical database, utilizing new geoindicators and aiming to offer potential predictions of future scenarios. The subject matter is of significant scientific relevance, particularly as the Solfatara magmatic-hydraulic system holds considerable interest, not only for the scientific community engaged in the study of this system, but also due to the concerns regarding the potential evolution of the current bradyseism towards eruptive scenarios, which may have substantial societal implications for the population residing in its vicinity. However, despite the relevance of the topic, I feel that the paper in its current state is not ready for publication, and that some concerns about the manuscript should be addressed before publication.

As written in our paper, "*The reason that pushed us to write this paper is to provide a contribution to the discussion animating, in this period, the international scientific community on the possible evolution of the unrest episode currently affecting the Campi Flegrei. The mitigation of the volcanic hazard in the Campi Flegrei is not a local issue because worldwide volcanologists look at it as an analogue of similar volcanic systems (see Chapter 11 of Marini et al., 2022).*" We are aware of the "*substantial societal implications for the population residing in its vicinity*", as written be the reviewer. In fact, our paper ends with the following considerations on societal implications: "*If we want 500,000 people to continue to live in the Campi Flegrei area affected by the bradyseism without the sword of Damocles of a hydrothermal event, it is necessary to find an appropriate solution such as to guarantee the seismic stability of a large number of buildings, in the order of 100,000, or to manage the bradyseism by zeroing the inflation of the intermediate reservoir depressurizing it. The first approach requires an investment without any economic return and does not mitigate the hazard posed by hydrothermal events. The second strategy provides a permanent solution to the problem in that it cancels the hazard posed by hydrothermal events (Lima et al. 2024). It requires a considerable initial investment to drill a suitable number of geothermal wells to ~4 km depth and to construct both a geothermal power plant and a mineral recovery plant. However, it provides a considerable economic return, thanks to the exploitation of geothermal energy for electrical production and the recovery of raw materials of utmost interest such as lithium. The feasibility of geothermal exploitation was proven by AGIP-ENEL activities carried out in the '70s and '80s (see above). The obstacles that existed at that time and caused the end of geothermal exploration no longer exist today, thanks to the improvements in drilling materials and technologies, as demonstrated by ongoing drilling activities in several supercritical geothermal systems (e.g., Reinsch et al., 2017).*"

Summing up, we are aligned with the reviewer words.

Nevertheless, in the annotated manuscript, the reviewer wrote the following sentences concerning our considerations on the societal implications reported above: "*I do not question the accuracy or validity of these statements, but note that there is no discussion of these issues in the main text, and here in the conclusion they seem off topic. That arguments should be considered and discussed in a dedicated section of the paper where the authors can provide the necessary data, cost analyses and other relevant information to support the claims made in the conclusion. Alternatively, it is recommended that these statements be removed from the concluding section.*"

Considerations on the need to drill and exploit geothermal energy, similar to our considerations, were meanwhile published by Lima, A., Bodnar, R. J., De Vivo, B., Spera, F. J., Belkin, H. E.: The "breathing" Earth (la terra che respira) at Solfatara-Pisciarelli (Campi Flegrei, southern Italy) during 2005-2024: Nature's way of attenuating the effects of bradyseism through gradual and episodic release of subsurface pressure. American Mineralogist, in press, https://doi.org/10.2138/am-2024-9516 2024, 2024. The main difference between our paper and that of Lima et al. (2024) is that they proposed to drill geothermal wells

to 5 km depth or more whereas, in our opinion, it is useless to drill so deep, because the target of geothermal drilling is the 2.7 to 4.0 km deep intermediate reservoir. No cost analysis was presented by Lima et al. (2024). Therefore, for the principle of *par condicio competitorum*, we would be inclined to maintain our considerations on societal implications in our paper. If we are asked to review our manuscript, we would prefer to move these considerations from the concluding section to section "4.8 Possible future scenarios". In fact, these societal implications are a directly linked to the hydrothermal eruption possible scenario.

In the following, I would like to point out two reasons I consider to be of great importance, and which should be reconsidered by the authors before a possible publication. Furthermore, in an attached file, I have integrated the comments and suggestions next to the relative text which are aimed at improving the readability of the text and the completeness of the information according to my personal vision.

We thank the reviewer for the constructive comments and suggestions reported in the annotated manuscript. If we are asked to review our manuscript, we are ready to change our paper according to all the reviewer's comments and suggestions apart from that on the societal implications (see above) and Figures 8c and 8b. The time scale of these two figures, referring to seismic events, begin with 2005 because no data on seismic events are reported on the INGV website (https://terremoti.ov.ingv.it/gossip/flegrei/years.html) before 2005, apart from 1983 and 1984. If needed, we will try to ameliorate Figure 8. We are sure that our manuscript will be improved thanks to the appreciated comments and suggestions of the reviewer.

In summary, the two most important aspects of the manuscript I believe should be reviewed in order to improve the quality of the paper before its possible publication are as follows:

1. The heart of the work consists in the reinterpretation of geochemical data based on the use of new geoindicators. Unfortunately, the text does not contain the information necessary to understand and interpret these geoindicators, which, as the paper is set up, must be well understood to ascertain the fields of applicability and potential limitations. This makes the paper not immediately understandable nor directly usable by the scientific community (after reading this document, can the use of these new geo-indicators be replicated in other cases?), unless one looks for the source, however this is contained in a book that precludes easy acquisition for many. I myself have not been able to find it, as I explained in the attached file. However, the authors of this manuscript are also the authors of the monograph; therefore, it is my opinion that they could easily integrate the text with a specific section describing the new geoindicators, which would certainly enhance the text. As a side effect, I was only partially able to follow the exchange between the first reviewer and the authors, due to the asymmetry of information caused by my lack of knowledge of the content of the monograph cited.

We agree with the reviewer on this point and we are ready to expand considerably the section Method of our paper, in order to provide the information needed to understand the new gas-geoindicators of Marini et al. (2022), including their applicability and limitations. It is true that, in our paper, the considerations on these new gas-geoindicators are limited to a minimum, but we thought that the book of Marini et al. (2022) was part of the scientific literature and was easily acquirable as any scientific paper. We are sorry for the difficulties encountered by the reviewer and we are ready to email to the reviewer a copy of our e-book.

1. Another significant concern relates to the capability to predict specific scenarios following the analysis and model interpretation. I acknowledge that the term "prediction" can have various meanings; however, in scientific literature, it is generally accepted to convey the notion of "the expectation of an occurrence under certain conditions". Typically, this expectation is quantified and qualified through numerical analysis (statistical or probabilistic), which lends credibility to the process of predicting future events or outcomes. Prediction can be viewed as part of hypothesis testing, where one formulates predictions as components of hypotheses. These predictions can then be empirically tested through experiments or observations to confirm or refute the

hypotheses. However, this concept is not clearly articulated in the text. Instead, the authors present reasonable scenarios that could evolve in different directions, potentially even oppositely, without providing any arguments or analyses to support specific predictions. The text offers only a description of various possible scenarios, lacking a basis for making predictions. In my opinion, this aspect of the paper needs to be reconsidered, given the significant importance of the ability to make predictions, especially for local authorities responsible for managing risk in a densely populated area like Campi Flegrei. Therefore, I suggest that the authors revise the use of the term "prediction" from the title onward, modifying the text accordingly or providing sufficient justification to support the possibility of predicting potential risk scenarios.

We have no problem to change the terminology. We are ready to use the term inference instead of prediction, the verb to infer instead of to predict, and so on.

I hope that the critical reading of the manuscript that I propose in its current state can be a constructive stimulus for the authors, regardless of the outcome of the publication.

Again, we thank the reviewer for the constructive comments and suggestions, irrespective of the fate of our paper.

**Citation**: https://doi.org/10.5194/egusphere-2024-1306-RC3

---

## Author Response (AR1)

**REFEREE N.1 – 1st review**

**Please, note that the Referee's comments are reported in red color, the Author's responses are in blue color and the Author's changes in manuscript are in black color.**

**1.1 General comments - 1st review of Referee n.1**

Marini et al. attempt to integrate new gas geochemical data and existing information into a revised "conceptual model" of the Solfatara magmatic-hydrothermal system at Campi Flegrei. I found the manuscript very difficult to review as a lot of important information, namely regarding the methods, the underlying assumptions, uncertainty assessment, known vs. inferred geology, as well as the exact lines of reasoning from data to interpretation are missing. This renders the work in its current form essentially irreproducible and very difficult to review because essentially almost every paragraph can be challenged. Even some (central) elements of the discussion seem to contradict each other; e.g., the concept of separate "reservoirs" as locations of equilibration of different gas geothermometers is obsolete with the assumption of flow through a vertically connected structure. Also, regarding the core chemistry, the authors (apparently coming from a volcanology background) seem to rely on shallow fumarole gas equilibria and ignore stablished knowledge from the magmatic-hydrothermal systems community for the deeper and higher pressured parts of the system (e.g., $H_2S$ generation through the well-known $SO_2$ disproportionation reaction). I think that for these reasons the manuscript should not be published in its current from and requires a complete revise beyond "major revisions".

In the following I will highlight some key problem areas (and for each one only some facets and ignore many questionable details) of the manuscript but will refrain from a detailed list of all issues.

**1.2 Author's response to the general comments - 1st review of Referee n.1**

For what concerns the first sentence of the General comments [*Marini et al. attempt to integrate new gas geochemical data and existing information into a revised "conceptual model" of the Solfatara magmatic-hydrothermal system at Campi Flegrei*], it must be noted that we did not produce any new gas geochemical data, whereas we used the Solfatara geochemical database, produced in forty years, first by Cioni and coworkers and second by Chiodini and coworkers, with the aim to monitor the temperatures and total fluid pressures in the three reservoirs which are present at different depths in the Campi Flegrei caldera. As part of our work, we also revised the conceptual model of the Solfatara magmatic-hydrothermal system", integrating relevant geological data, both collected at the surface and obtained by the deep geothermal wells drilled by Agip-ENEL in the late '70s early '80s, and pertinent geophysical data.

For what concerns the second sentence of the General comments [*I found the manuscript very difficult to review as a lot of important information, namely regarding the methods, the underlying assumptions, uncertainty assessment, known vs. inferred geology, as well as the exact lines of reasoning from data to interpretation are missing*], we recall that, as written in the paper, the methods are the gas-geothermometers and gas-geobarometers specifically derived and calibrated for the Solfatara fumaroles. These gas-geothermometers and gas-geobarometers as well as their underlying assumptions are thoroughly described and discussed in the book of Marini et al. (2022). In the revised manuscript, we have expanded considerably the section "2 Methods", explaining how the geothermometers and geobarometers were derived and reporting the equations used to compute the equilibrium temperatures and total fluid pressures of CO, $CH_4$, and $H_2S$. The uncertainty of gas-geothermometers and gas-geobarometers cannot be assed because this is an impossible exercise for fumarolic fluids' geo-indicators, such as those elaborated and/or used by Cioni et al. (1984, 1989), Taran (1986), Giggenbach (1987), Tedesco and Sabroux (1987), Chiodini and Cioni (1989), Cioni and Marini (1990), Chiodini and Marini (1998), Caliro et al. (2007), Chiodini et al. (2011, 2015, 2016, 2017, 2021), Fiebig et al. (2013), Moretti et al. (2017, 2018, 2020), Buono et al. (2023) and

many others. There is no inferred geology in our work. Again, we considered only known geological data, both collected at the surface and obtained by deep geothermal boreholes, as well as known geophysical data to revise the "conceptual model" of the Solfatara magmatic-hydrothermal system. Let us skip, for the moment, the sentence *exact lines of reasoning from data to interpretation are missing*; we will return on this point below.

For what concerns the following sentence of the General comments [*This renders the work in its current form essentially irreproducible and very difficult to review because essentially almost every paragraph can be challenged*]. We do not see any problem to reproduce our results, namely the temperatures and total fluid pressures in the three reservoirs which are present at depth in the Campi Flegrei caldera, in that anyone can use our gas-geothermometers and gas-geobarometers to obtain equilibrium temperature and total fluid pressure values. This is an easy task using the EXCEL file reported as supplementary material.

Let us now consider the following sentence of the General comments [*Even some (central) elements of the discussion seem to contradict each other; e.g., the concept of separate "reservoirs" as locations of equilibration of different gas geothermometers is obsolete with the assumption of flow through a vertically connected structure*]. Let us try to explain how it is possible that distinct gases (i.e., $H_2S$, $CH_4$, and $CO$) equilibrate at different temperatures and total fluid pressures in three reservoirs different but connected to each other by a fault zone along which a continuous compressible fluid flow occurs. To this purpose, let us assume that the branches of the fault zone extending from the magma chamber to the surface and connecting the three reservoirs to each other are small-diameter pipelines, the shallow reservoir volume contributing to the fumarolic discharge is a medium-diameter pipeline, and the intermediate and deep reservoirs' volumes contributing to the fumarolic discharge are large-diameter pipelines. The continuity equation written in a simple way for a compressible fluid flow is: $Q = \rho \cdot A \cdot v$, where $Q$ is the mass flow rate ($M/T$), $\rho$ is the density of the compressible fluid ($M/L^3$), $A$ is the cross-sectional area ($L^2$), and $v$ is the velocity of the fluid ($L/T$). Even if $\rho$ is a function of temperature and pressure, the continuity equation implies that the fluids will move very fast along the fault zone, very slowly in the deep reservoir (in which fluids resides long enough to permit hydrogen sulfide re-equilibration) and in the overlying intermediate reservoir (in which fluids resides long enough to allow methane re-equilibration), and not so fast but not so slowly in the shallow reservoir (in which fluids resides long enough to allow carbon monoxide re-equilibration).

Let us now take into account the subsequent sentence of the General comments [*Also, regarding the core chemistry, the authors (apparently coming from a volcanology background) seem to rely on shallow fumarole gas equilibria and ignore stablished knowledge from the magmatic-hydrothermal systems community for the deeper and higher pressured parts of the system (e.g., H2S generation through the well-known SO2 disproportionation reaction)*]. We do not rely on "*shallow fumarole gas equilibria*", because no gas species equilibrates at surface (fumarolic) conditions. Again, we used the chemistry of the Solfatara fumarolic fluids and the gas-geothermometers and gas-geobarometers of Marini et al. (2022), which were specifically derived and calibrated for the Solfatara fumarolic fluids, taking into account the deviations from the ideal gas behavior (an unprecedented exercise with respect to previous papers on gas-geothermometers and gas-geobarometers), to monitor the temperature and total fluid pressure, over a time interval of ~40 years: (i) in the shallow reservoir (0.25-0.45 km depth), where $CO$ equilibrates; (ii) in the intermediate reservoir (2.7-4.0 km depth), where $CH_4$ attains equilibrium; (iii) in the deep reservoir (6.5-7.5 km depth), where $H_2S$ achieves equilibrium.

We are aware of the "*H2S generation through the well-known SO2 disproportionation reaction*", which is actually thoroughly discussed in Chapter 9 - The Redox Potential and Sulfur Gas Species of Marini et al. (2022) where it is numbered Eqn. (10). The $SO_2$ disproportionation reaction produces both $H_2S$ and $H_2SO_4$. Sulfuric acid reacts promptly with the carbonate rocks which are present above the Campi Flegrei magma chamber, according to the following reaction:

$$CaCO_{3(s)} + H_2SO_4 = CaSO_{4(s)} + CO_{2(aq,g)} + H_2O.$$

This is the same reaction explaining the widespread occurrence of anhydrite in the geothermal systems of Central Italy, such as Latera (Cavarretta et al. 1985; Marini and Chiodini 1994), even as part of the contact-metasomatism paragenesis, as well as in the geothermal system of Nisyros (Ambrosio et al. 2010), where again underlined carbonate rocks and marbles are present above the magma chamber. The same reaction evidently occurs at the base of the deep reservoir which is present at depths of 6.5-7.5 km in the Campi Flegrei caldera. Anhydrite is the partner of calcite in the heterogeneous reaction:

$$CaSO_{4(s)} + CO_2 + 4\,H_2 = CaCO_{3(s)} + H_2S + 3\,H_2O,$$

representing the theoretical foundation of the $H_2S$ gas-geothermometer and gas-geobarometer of Marini et al. (2022).

We have no comment on the following sentences of the General comments [*I think that for these reasons the manuscript should not be published in its current from and requires a complete revise beyond "major revisions". In the following I will highlight some key problem areas (and for each one only some facets and ignore many questionable details) of the manuscript but will refrain from a detailed list of all issues*].

**1.3 Author's changes in manuscript related to the general comments - 1st review of Referee n.1**

At lines 104 to 182 of the revised manuscript we have explained how the geothermometers and geobarometers were derived and reported the equations used to compute the equilibrium temperatures and total fluid pressures of CO, $CH_4$, and $H_2S$.

**2.1 Specific comments 1: methods and assumptions behind them - 1st review of Referee n.1**

The authors rely on three gas equilibria to obtain "distinct equilibration temperatures and related total fluid pressures". They do not elaborate how these were computed and not even how such a "equilibration temperature" is to be interpreted (e.g.: is it the lowest temperature at which the reaction went to equilibrium along the fluid's upflow path and which was then preserved due to kinetic limitations at lower temperature?), and how such equilibrium could be demonstrated at all.

It remains completely opaque how the computations were done. I would argue that at least some of the stated computations are currently not meaningfully possible for the lack of adequate thermodynamic data and models, e.g. for a complex gas phase in equilibrium with a 21 or 33.5 wt% aqueous NaCl brine – for the conditions stated there exists neither an adequate solubility model for the gases in such a brine nor are even remotely accurate activity or fugacity coefficients available. How then could temperatures and pressures be derived? I tried to make my way through the main author's 2022 book (375 pages, the main findings of which the present manuscript is supposed to summarize and explain) but I could not identify a coherent summary of the computations and the underlying assumptions with reasonable effort. This manuscript should deliver a crystal-clear and reproducible description and openly lay out and discuss the major assumptions to become a publishable contribution.

There exists a relatively elaborate assessment of sulfur chemical behavior in the deep vs. shallow parts of magmatic-hydrothermal systems that was established some decades ago in the economic geology community (just to name the works by Einaudi and Hedenquist as an example) including the importance of "rock-buffered" vs. "direct degassing" paths. The very relevant insights presented there go largely ignored by the present authors. Namely, the important SO2 disproportionation reaction to generate H2S (and there are variations of the theme depending on temperature, rock, etc.) is ignored although it is considered most important by many researchers in the field. The reactions (3) and (4) involving calcite also seem odd; most people working on the deeper parts of magmatic-hydrothermal systems would see these reactions going to the left (albeit possibly rather involving SO2 and using plagioclase rather than calcite) to sequester sulfur rather than creating H2S, see, e.g., Henley at al., JVGR 2022. I wonder what "H2S temperatures" would come out if this was properly taken into account of if "H2S temperatures" would have any meaning then.

**2.2 Author's response to the Specific comments 1: methods and assumptions behind - 1st review of Referee n.1**

Let us consider the first sentences of the Specific comments 1 [*The authors rely on three gas equilibria to obtain "distinct equilibration temperatures and related total fluid pressures". They do not elaborate how these were computed and not even how such a "equilibration temperature" is to be interpreted (e.g.: is it the lowest temperature at which the reaction went to equilibrium along the fluid's upflow path and which was then preserved due to kinetic limitations at lower temperature?), and how such equilibrium could be demonstrated at all*]. Again, in the revised manuscript, we have expanded considerably the section "2 Methods", explaining how the geothermometers and geobarometers were derived and reporting the equations used to compute the equilibrium temperatures and total fluid pressures of CO, $CH_4$, and $H_2S$. The gas equilibration temperature is the temperature at which the considered reaction or the considered key gas species (i.e., $H_2S$, $CH_4$, and CO) attain chemical equilibrium, which is assumed to be preserved at lower temperatures because reaction kinetics becomes too small (explanation added in the revised manuscript). In general, in geothermometry and geobarometry, equilibrium cannot be demonstrated, it is assumed. Furthermore, in the case of Solfatara fumaroles, the computed equilibrium temperatures and pressures cannot be compared with any measured value, because the nearest deep geothermal well, called CF23/Agnano 1, is found at a distance of ca. 1 km from the Solfatara fumaroles [incidentally, Well CF23/Agnano 1 was drilled in the '50s by SAFEN "on the eastern slopes of the Solfatara relief and reached a total depth of 1841 m, where a maximum temperature of 325 °C was measured (Minucci 1964)", as reported by Marini et al. (2022)]. This is a problem not only for our manuscript, but also for all previous geothermometric-geobarometric studies on the Solfatara fluids such as those of Cioni et al. (1984, 1989), Tedesco and Sabroux (1987), Chiodini and Cioni (1989), Cioni and Marini (1990), Chiodini and Marini (1998), Caliro et al. (2007), Chiodini et al. (2011, 2015, 2016, 2017, 2021), Fiebig et al. (2013), Moretti et al. (2017, 2018, 2020), Buono et al. (2023) as well as the monthly INGV Bulletins.

Let us now take into account the following sentences of the Specific comments 1 [*It remains completely opaque how the computations were done. I would argue that at least some of the stated computations are currently not meaningfully possible for the lack of adequate thermodynamic data and models, e.g. for a complex gas phase in equilibrium with a 21 or 33.5 wt% aqueous NaCl brine – for the conditions stated there exists neither an adequate solubility model for the gases in such a brine nor are even remotely accurate activity or fugacity coefficients available. How then could temperatures and pressures be derived? I tried to make my way through the main author's 2022 book (375 pages, the main findings of which the present manuscript is supposed to summarize and explain) but I could not identify a coherent summary of the computations and the underlying assumptions with reasonable effort. This manuscript should deliver a crystal-clear and reproducible description and openly lay out and discuss the major assumptions to become a publishable contribution*]. For what concerns the saturation decompression paths of Solfatara fluids involving a vapor phase and a brine containing 21 or 33.5 wt% NaCl, brine-vapor coexistence is assumed to fix $H_2O$ partial pressure. This is the only role played by the brine in our approach, which is similar to that of Giggenbach (1987). The main difference between our approach and that of Giggenbach (1987) concerns the fugacity coefficients of each gas species in the gas mixtures of interest. Fugacity coefficients of gas species were computed by Marini et al. (2022) following different approaches. First, the fugacity coefficients of $H_2O$ and $CO_2$ were obtained along the isenthalpic expansion path of Bocca Grande fluids of different composition: (i) by using the GERG-2008 thermodynamic module of the CONVAL Software Package and the virial EOS (see pages 102-111 of Marini et al. 2022), (ii) by using the Peng-Robinson EOS (see pages 111-118 of Marini et al. 2022), and (iii) from Table 3 in Appendix A of Gallagher et al. (1993) reporting the results of the EOS of these authors. Then, the three series of fugacity coefficients of $H_2O$ and $CO_2$ were compared (see pages 118-121 of Marini et al. 2022). After this comparison, Marini et al. (2022) decided to use the Peng-Robinson EOS to compute the fugacity coefficients of all the gas species of interest (not only $H_2O$ and $CO_2$, but also $H_2S$, $H_2$, $CH_4$, and CO) for all the considered decompression (expansion) paths, namely: the isenthalpic decompression path, the linear P-T decompression path and the decompression paths involving a vapor phase and a brine containing 21 or 33.5 wt% NaCl. Fugacity coefficients were then considered in the calibration of gas-geothermometers as discussed at pages 286-287, 335, and 344 of Marini et al. (2022).

The explanation on how the fugacity coefficients of gas species in the gas mixtures of interest were computed is found at lines 114-121 of the revised manuscript.

In the subsequent part of the Specific comments 1, the referee focuses again on the $SO_2$ disproportionation reaction [*There exists a relatively elaborate assessment of sulfur chemical behavior in the deep vs. shallow parts of magmatic-hydrothermal systems that was established some decades ago in the economic geology community (just to name the works by Einaudi and Hedenquist as an example) including the importance of "rock-buffered" vs. "direct degassing" paths. The very relevant insights presented there go largely ignored by the present authors. Namely, the important $SO_2$ disproportionation reaction to generate $H_2S$ (and there are variations of the theme depending on temperature, rock, etc.) is ignored although it is considered most important by many researchers in the field. The reactions (3) and (4) involving calcite also seem odd; most people working on the deeper parts of magmatic-hydrothermal systems would see these reactions going to the left (albeit possibly rather involving $SO_2$ and using plagioclase rather than calcite) to sequester sulfur rather than creating $H_2S$, see, e.g., Henley at al., JVGR 2022. I wonder what "$H2S$ temperatures" would come out if this was properly taken into account of if "$H2S$ temperatures" would have any meaning then.*]

Above, we have already considered the comment on the $SO_2$ disproportionation reaction. For what concerns the works by Einaudi and Hedenquist, we have focused on the paper by Einaudi et al. (2005), in which the authors contrast the oxidation and sulfidation states of fresh igneous rocks from arc environments and the sulfidation states of sulfide assemblages in calc-alkalic porphyry copper, porphyry-related base metal veins, and epithermal gold-silver deposits with compositions of fluids from active systems by plotting vapor compositions in log $f_{S2}$ − 1,000/T, $R_H$ − 1,000/T, and $R_S$ − 1,000/T diagrams, where $R_H$ = log $(f_{H2} / f_{H2O})$ ≈ log $(X_{H2}/X_{H2O})$, $R_S$ = log $(f_{H2} / f_{H2S})$ ≈ log $(X_{H2}/X_{H2S})$, and X = mole fraction of the gas. It must be underscored that $H_2$ and $H_2S$ are apolar gas molecules, whose fugacity coefficients increase with increasing temperature and pressure, whereas $H_2O$ is a polar gas molecule, whose fugacity coefficient decreases with increasing temperature and pressure, as shown by Marini et al. (2022). Although the findings of Marini et al. (2022) refer, strictly speaking, to the Solfatara fluids, the fugacity coefficients of $H_2$, $H_2S$, and $H_2O$ are expected to vary in the same way, irrespective of chemical and physical conditions of the considered system, because in any case $H_2$ and $H_2S$ are apolar molecules and $H_2O$ is a polar molecule. Therefore, the assumption log $(f_{H2} / f_{H2S})$ ≈ log $(X_{H2}/X_{H2S})$ might still be correct to some extent, whereas the assumption log $(f_{H2} / f_{H2O})$ ≈ log $(X_{H2}/X_{H2O})$, which was first proposed by Giggenbach (1987), becomes increasingly wrong with increasing temperature and pressure. Consequently, the related considerations of this work should be considered with caution.

Let us consider the sentence *"The reactions (3) and (4) involving calcite also seem odd; most people working on the deeper parts of magmatic-hydrothermal systems would see these reactions going to the left (albeit possibly rather involving $SO_2$ and using plagioclase rather than calcite) to sequester sulfur rather than creating $H_2S$, see, e.g., Henley at al., JVGR 2022. I wonder what "$H2S$ temperatures" would come out if this was properly taken into account of if "$H2S$ temperatures" would have any meaning then"*. In our approach, reactions (3) and (4) are considered at equilibrium, irrespective of their direction. Furthermore, we underscore once again that in the Campi Flegrei, above the magma chamber, there is a thick sequence of carbonate rocks, as revealed by a number of geophysical studies on the active seismic reflection data of the SERAPIS survey and the passive seismic data of the 1982-1984 bradyseismic crisis (see references in our manuscript). The thick sequence of carbonate rocks occurring above the Campi Flegrei magma chamber is the same sequence cropping out all around the Campanian Plain, which has been downthrown by the boundary faults of the plain and other NW-trending (Apenninic) tectonic structures and has been dissected by the NE-trending (anti-Apenninic) conjugated structures. The occurrence of these carbonate rocks above the magma chamber prompted us to use calcite instead of plagioclase as partner of anhydrite in reactions (3) and (4). Of course, we would have done the same if Al-silicate rocks were present above the magma chamber, but this is not the case at Campi Flegrei. As the reasons for considering calcite instead of plagioclase as partner of anhydrite in reactions (3) and (4) were already discussed in the original manuscript (see section 4.5 The zone of $H_2S$ equilibration), no change was done in the revised manuscript on this matter.

**2.3 Author's changes in manuscript related to the Specific comments 1: methods and assumptions behind - 1[st] review of Referee n.1**

Again, at lines 104 to 182 of the revised manuscript we have explained how the geothermometers and geobarometers were derived and reported the equations used to compute the equilibrium temperatures and total fluid pressures of CO, CH$_4$, and H$_2$S. In particular, the approach adopted to calculate the fugacity coefficients of gas species in the gas mixtures of interest is explained at lines 114-121 of the revised manuscript, whereas the reason why geo-indicators work ("Chemical equilibrium is assumed to be preserved upon cooling because reaction kinetics becomes too small") was added to lines 139-140.

**3.1 Specific comments 2: the core of the conceptual model - 1[st] review of Referee n.1**

When looking closer one has a hard time to understand reasoning behind the "conceptual model". On the one hand, three "reservoirs" at different depth, separated by aquicludes, are postulated; on the other hand – to make the fluids migrate from a magma at >8km depth to the surface – connected flow through a vertically extensive "structure" needs to be invoked.

Now, first and foremost, I had a hard time understanding why the different gas equilibration temperatures should be specifically connected to the individual "reservoirs" and how the latter were inferred in the first place. Is there some circular reasoning? Or does actual, drilled geology with porosity and permeability measurements come into play? There is some mentioning of wells etc. but the exact reasoning remained blurry to me, at best. So, one cannot even tell if the depth of "reservoirs" and "aquicludes" is well-constrained or just a guess.

Then, if there is a vertically connecting structure – what is the meaning of the temperatures then? "Equilibration" will happen (and be "frozen in" if my above speculation about the meaning of "equilibration temperature" is correct) somewhere along the flow path and why should that be connected to the depth of any of the "reservoirs" then? Wouldn't it make more sense that the increase in apparent temperatures and pressures (if correct) reflects rather a change in the hydraulic regime (or the degassing rate) such that the chemical signal of deeper fluids gets better preserved rather than a specific "reservoir" getting hotter and stronger pressurized (see, e.g., the overpressure waves in a magmatic-hydrothermal systems self-developing in the simulations of Weis et al., 2012; later suggested also by Lupi et al. for Campi Flegrei). What would speak for and against such different possibilities, why aren't they considered and also tested against the data?

In the whole conceptual model discussion, speculative ideas (such as Fournier's "self-sealing" quartz layer for which later studies found little evidence), inferred vs. drilled geology, geochemical data with different possible interpretations etc. are just mingled without testing for plausibility etc. Personally, I think that this is quite far away from best practices; the different ideas should be formulated as hypotheses and then tested to the degree possible.

**3.2 Author's response to the Specific comments 2: the core of the conceptual model - 1[st] review of Referee n.1**

Let us consider the first sentences of the Specific comments 2 [*When looking closer one has a hard time to understand reasoning behind the "conceptual model". On the one hand, three "reservoirs" at different depth, separated by aquicludes, are postulated; on the other hand – to make the fluids migrate from a magma at >8km depth to the surface – connected flow through a vertically extensive "structure" needs to be invoked.*

*Now, first and foremost, I had a hard time understanding why the different gas equilibration temperatures should be specifically connected to the individual "reservoirs" and how the latter were inferred in the first place. Is there some circular reasoning? Or does actual, drilled geology with porosity and permeability measurements come into play? There is some mentioning of wells etc. but the exact reasoning remained blurry to me, at best. So, one cannot even tell if the depth of "reservoirs" and "aquicludes" is well-constrained or just a guess.]*

As already recalled above, the three reservoirs at different depth and the interposed aquicludes are not postulated, as written by the referee, but are constrained by known geological data, both collected at the surface and obtained by deep geothermal boreholes, which were drilled at depths from 1.5 to 3 km approximately (see sections 4.1), as well as by known geophysical data which were processed and interpreted by a number of studies (e.g., sections 4.2, 4.4, and 4.5). All is all, these data provide a well-constrained conceptual model extending to ca. 8 km where the magma chamber is located (see section 4.6). There is no circular reasoning in the integration of geological and geophysical data to reconstruct the conceptual model of the Solfatara magmatic-hydrothermal system at Campi Flegrei, that we did following the best practices used in geothermal exploration (Cumming, 2009, 2016) and other applied geo-sciences (Fournier 1999). Nevertheless, to help the reader, we added a conceptual model cross-section of the system of interest, showing the shallow, intermediate, and deep reservoirs (Figure 5 of the revised manuscript).

Porosity and permeability measurements were performed on cores obtained in the deep geothermal boreholes, but obviously they do not extend at depths > 3 km approximately. These data are summarized by Rosi and Sbrana (1987). Nevertheless, they are not essential for the elaboration of the conceptual model, they would be much more important, not to say essential for the implementation of a numerical model, but this exercise is beyond the aims of our work.

The vertically permeable tectonic structure extending from the magma chamber to the surface and permitting the upward fluid flow through the three reservoirs and the interposed aquicludes "was activated during the final phase of the 1982-1984 seismic crisis along tectonic trends already active in the past (Rosi and Sbrana, 1987), as suggested by the occurrence of low-magnitudo earthquakes at depths of 0-8 km (D'Auria et al., 2011)" (see lines 435-438 of the revised manuscript). We take this opportunity to recall that large changes in several chemical components (e.g., $N_2$, $CO_2$, $H_2S$) of fumarolic gases occurred in the time interval May-September 1984 (e.g., Cioni et al. 1989; Caliro et al. 2014; Buono et al. 2022). They fit perfectly with the activation of this faulted-fractured zone and the transition from closed state to open state of the Solfatara magmatic-hydrothermal system, as discussed in section 4.3 of our manuscript.

Although we have already explained above "*why the different gas equilibration temperatures should be specifically connected to the individual "reservoirs"*", let us return again on this point to answer the referee. Accepting that there is a continuous, upward fluid flow along the vertically permeable tectonic structure extending from the magma chamber to the surface and, consequently, through the three reservoirs and the interposed aquicludes, the residence time of the fluids in each reservoir and in each aquiclude is expected to be directly proportional to the volume contributing to the fumarolic discharge of each reservoir and each aquiclude; thus, the fluids will spend: (1) a relatively long time in the deep and in the intermediate reservoirs because of their large volumes, even the fraction contributing to the fumarolic discharge; (2) a not too long and not too short time in the shallow reservoir because it has a volume much smaller than the deep and the intermediate reservoirs; and (3) a short or very short time in the aquicludes, where the volume available to fluid flow is small or very small. Moreover, in a given time interval, the temperature is expected to be constant or nearly so in the three reservoirs, the total fluid pressure is expected to experience negligible to limited variations in the three reservoirs, whereas both parameters are expected to experience large upward decreases along the faulted-fractured zone crossing the aquicludes (see below for further details). Thus, in each one of the three reservoirs and in a given time interval, gas species are expected to attain equilibrium at a given temperature, total fluid pressure condition. In contrast, equilibrium temperature is expected to experience small re-adjustments during the transit from one reservoir to the other because of the small residence time in the aquicludes, in spite of the large decrease in temperature and fluid pressure.

Previous considerations clarify also the sentences of the Specific comments 2 "*Then, if there is a vertically connecting structure – what is the meaning of the temperatures then? "Equilibration" will happen (and be "frozen in" if my above speculation about the meaning of "equilibration temperature" is correct) somewhere along the flow path and why should that be connected to the depth of any of the "reservoirs" then?*", whereas some considerations are needed for the sentences of the Specific

comments 2 "*Wouldn't it make more sense that the increase in apparent temperatures and pressures (if correct) reflects rather a change in the hydraulic regime (or the degassing rate) such that the chemical signal of deeper fluids gets better preserved rather than a specific "reservoir" getting hotter and stronger pressurized (see, e.g., the overpressure waves in a magmatic-hydrothermal systems self-developing in the simulations of Weis et al., 2012; later suggested also by Lupi et al. for Campi Flegrei). What would speak for and against such different possibilities, why aren't they considered and also tested against the data?*" Weis et al. (2012) implemented a numerical model to explain porphyry-type ore deposits, which is not relevant for the Solfatara magmatic-hydrothermal system at Campi Flegrei because the geological contexts are totally different. In fact, the three porphyry-style deposits considered by Weis et al. (2012), namely, Yerington in Nevada, Bingham Canyon in Utah and Batu Hijau in Indonesia are sustained by an unspecified source pluton or an unspecified inferred source pluton, whereas the Campi Flegrei volcanic area is characterized by a nested caldera with a magma chamber, whose top is positioned at ca. 8 km depth, probably hosting a trachyandesitic magma, according to Caliro et al. (2014) and Buono et a. (2022). It is unclear to us if the numerical model implemented by Weis et al. (2012) is able to reproduce the geological complexities of (at least) one of the three porphyry-style deposits because this crucial point is not discussed. Even more important, it is unclear to us the relation between the study of Weis et al. (2012) and our work, also considering that (1) the two time-windows are totally different (the time-scale of ore formation is between 50,000 and 100,000 years, whereas at the Solfatara we are considering a time interval of 38 years only), (2) we focus on the geochemistry of Solfatara fluids, whereas no data on fluid chemistry are given by Weis et al. (2012), apart from generic sentences such as "*We assume that, on cooling through the solidus temperature of 700°C, 5 wt% of the magma are released as aqueous fluid with 10 wt% NaCl through a cupola in the roof of the magma chamber.*"

Lupi and coworkers speculate on the possible effects of large regional earthquakes on the Campi Flegrei magma chamber. They simulate "*the propagation of elastic waves and show that passing body waves impose high dynamic strains at the roof of the magmatic reservoir of the Campi Flegrei at about 7 km depth. This may promote a short-lived embrittlement of the magma reservoir's carapace, which is otherwise impermeable during inter-seismic times. Such failure allows magma and exsolved volatiles to be released from the magmatic reservoir. The fluids, namely exsolved volatiles and/or melts, ascent through a nominally plastic zone above the magmatic reservoir*." Although the idea might be interesting, the supporting evidence is not convincing at all. Again, the numerical model implemented by Lupi and coworkers is very generic and does not reproduce the geological reality of the Solfatara magmatic-hydrothermal system at Campi Flegrei. Furthermore, the cause-effect relationship between regional earthquakes and uplift leaves much to desire. In particular, we wonder why the numerous regional earthquakes that occurred before 1945 (the beginning of the time window considered by Lupi and coworkers) did not activate any uplift at Campi Flegrei, which begun in 1950, according to Del Gaudio et al. (2010), or in 1945, according to the pioneering investigation of Ranieri (1952).

For what concerns the last question of the Specific comments 2 [*What would speak for and against such different possibilities, why aren't they considered and also tested against the data?*], we underscore that the reason is obvious from what we wrote, in previous lines, on the works of Weiss and coworkers and Lupi and coworkers.

Let us now consider the following sentences of the Specific comments 2 [*In the whole conceptual model discussion, speculative ideas (such as Fournier's "self-sealing" quartz layer for which later studies found little evidence), inferred vs. drilled geology, geochemical data with different possible interpretations etc. are just mingled without testing for plausibility etc. Personally, I think that this is quite far away from best practices; the different ideas should be formulated as hypotheses and then tested to the degree possible*]. As written in the book of Marini et al. (2022), the reduction in permeability close to an intrusive heat source caused by quartz precipitation was discussed not only by Fournier (1999) but also in several other studies (e.g., Fournier 1985; Wells and Ghiorso 1991; Lowell et al. 1993; Moore et al. 1994; White and Mroczek 1998; Saishu et al. 2014; Scott and Driesner 2018). Concerning the "best practices" to be followed in data interpretation, we agree with the referee and, as a matter of fact, we applied these "best practices" also in our book and manuscript. For example, among the different series of $CO$-, $CH_4$- and $H_2S$ equilibrium temperatures and total fluid pressures, each one referring to a distinct decompression (expansion)

path (the isenthalpic decompression path, the linear P-T decompression path and the decompression paths involving a vapor phase and a brine containing 21 or 33.5 wt% NaCl), we selected the series of CO-, CH$_4$- and H$_2$S equilibrium temperatures and total fluid pressures for the decompression paths involving a vapor phase and a brine containing 21 or 33.5 wt% NaCl to simplify the discussion. Nevertheless, if a different series of CO- CH$_4$- and H$_2$S equilibrium temperatures and total fluid pressures is selected, the trend of constant temperature and total fluid pressure with time in the shallow reservoir and the trends of time-increasing temperature and total fluid pressure in the intermediate and deep reservoirs are still observed, although with some differences. We added this consideration at lines 212-215 of the revised manuscript. Thus, there is little doubt, on the progressive pressurization of both the deep aquifer and the intermediate aquifer, as well as on the good correspondence between the chronogram of the intermediate aquifer overpressure and the chronogram of ground uplift (see lines 479-509 of the revised manuscript).

**3.3 Author's changes in manuscript related to the Specific comments 2: the core of the conceptual model - 1$^{st}$ review of Referee n.1**

A conceptual model cross-section of the Solfatara magmatic-hydrothermal system, showing the shallow, intermediate, and deep reservoirs was added to revised manuscript at page 16 (lines 401-404). At lines 212-215 of the revised manuscript, we added a consideration on the trend of constant temperature and total fluid pressure with time in the shallow reservoir and the trends of time-increasing temperature and total fluid pressure in the intermediate and deep reservoirs, irrespective of the selected series of CO- CH$_4$- and H$_2$S equilibrium temperatures and total fluid pressures.

**4.1 Specific comments 3: drawing straight lines without reasoning - 1$^{st}$ review of Referee n.1**

The temperature and pressure profiles in Figs. 5 and 6 hinge on the – untested – assumption that the gas equilibration temperatures are representative for three "reservoirs" at different depth. In Fig. 5, temperature in each reservoir is taken to be constant. Unless I missed something important no reason is given why this should be the case. Rather, it is taken as granted (out of nothing) and then it is postulated that convection inside the reservoir homogenizes the temperature. Between the reservoirs – again: unless I overlooked something important – straight lines are drawn without reasoning and then it is stated that the "the heat transfer appears to be controlled by conduction". I think this is quite poor scientific practice to just draw a straight line in the absence of data and then to assume it is correct and make such a conclusion.

For fluid pressure (Fig. 6) the "reservoirs" are also drawn to have constant pressure even if more than a km high. This is obviously unphysical as there would have to be a hydrostatic pressure gradient (not necessarily linear as density of the fluid may vary with depth).

**4.2 Author's response to the Specific comments 3: drawing straight lines without reasoning - 1$^{st}$ review of Referee n.1**

For what concerns the first sentence of the Specific comments 3 [*The temperature and pressure profiles in Figs. 5 and 6 hinge on the – untested – assumption that the gas equilibration temperatures are representative for three "reservoirs" at different depth*], in the original manuscript, we have provided several pieces of evidence indicating that the three gas equilibration temperatures and related total fluid pressures refer to the three "reservoirs" positioned at different depths (see sections 4.1 to 4.5).

Let us take into consideration the following sentences of the Specific comments 3 [*In Fig. 5, temperature in each reservoir is taken to be constant. Unless I missed something important no reason is given why this should be the case. Rather, it is taken as granted (out of nothing) and then it is postulated that convection inside the reservoir homogenizes the temperature. Between the reservoirs – again: unless I overlooked something important – straight lines are drawn without reasoning and then it is*

*stated that the "the heat transfer appears to be controlled by conduction". I think this is quite poor scientific practice to just draw a straight line in the absence of data and then to assume it is correct and make such a conclusion.*]

To answer these criticisms of referee n.1, we have reported below some relevant sentences, in purple color, from the textbook written by Malcolm A. Grant and Paul F. Bixley (2011) **Geothermal Reservoir Engineering**, Second Edition, which is "the only training tool and professional reference dedicated to advising both new and experienced geothermal reservoir engineers". "*The simplest distinction made in temperature profiles is between conductive and convective profiles. When rock is impermeable, heat is transported by conduction. This produces a characteristic profile where temperature increases linearly with depth; the gradient will change if there is a change in thermal conductivity of the rock.*

[Figure]

**FIGURE 4.2** Temperature profiles in GPK-1 and GPK-2. Source: *Genter et al., 2009. © Geothermal Resources Council.*

Convection by contrast is a far more efficient means of heat transport than conduction. Once there is some permeability in the rock - and the required permeability is much less than what is needed for economic well performance - the fluid motion controls the temperature distribution. Convective profiles can take a considerable variety of forms, with isothermal sections, inversions, boiling sections, and mixtures of all of these. Figure 4.2 shows temperature profiles from two wells at the engineered geothermal systems (EGS) project at Soultz, France (Genter et al., 2009). There are three sections on the profile. The first kilometer has a high gradient and linear profile, indicating conductive transport. Then from 1 km to 3.3 km there is a much lower gradient, which is attributed to a convective system along faults and fissure zones. Finally, below 3.3 km there is again a high linear gradient, indicating conductive heat transport and consequently lower permeability in the surrounding formations. ….. An isothermal profile is a section of the well where the temperature is constant or nearly constant with depth. This can reflect circulation of fluid in a section of the wellbore or interzonal flow (without boiling), or it may be that the reservoir itself has isothermal temperatures due to convection."

For what concerns the following sentences of the Specific comments 3 [*For fluid pressure (Fig. 6) the "reservoirs" are also drawn to have constant pressure even if more than a km high. This is obviously unphysical as there would have to be a hydrostatic pressure gradient (not necessarily linear as density of the fluid may vary with depth),* it must be noted that the zero-pressure gradient or, in other terms, the vapor-static pressure or steam-static pressure gradient is a typical characteristic

of vapor-dominated (dry-steam) systems at temperature and pressure close to the point of maximum enthalpy of water vapor (235°C, 30.6 bar), including the shallow reservoir of the Solfatara hydrothermal-magmatic system. In contrast, the intermediate and deep reservoirs of the Solfatara hydrothermal-magmatic system are generally at temperatures and pressures much higher and their fluids have density values relatively high, making the pressure gradient in the intermediate and deep reservoirs higher than zero. The referee is absolutely right on this point. Therefore, in the revised manuscript, we have revised Figure 7 (Figure 6 in the original manuscript), showing the time changes, between September 1984 and January 2022, of the total fluid pressure vs. depth profile along a hypothetical borehole drilled in the Solfatara crater. For the vapor in equilibrium coexistence with a brine containing 21 wt% NaCl, we have considered the molar volumes reported in Tables 1, 2, 3, 4, and 5 of Chapter 9 of Marini et al. (2022), whereas for the vapor in equilibrium coexistence with a brine containing 33.5 wt% NaCl, we have considered the molar volumes reported in Tables 9, 10, 11, 12, and 13 of Chapter 9 of Marini et al. (2022). The obtained log-densities in log-kg/m$^3$ were then fitted against $X_{CO_2}$ and temperature to derive simple polynomials that were used to calculate the density of the gas mixtures in the intermediate and deep reservoirs, which were assumed to be constant to avoid complicating too much the calculations. Knowing the density, $\rho$ (in kg/m$^3$), the total fluid pressure at the bottom of the reservoir, $P_{tot,B}$ (bar), was computed by the relation:

$$P_{tot,B} = P_{tot,T} + \rho \cdot g \cdot 0.01$$

where $P_{tot,T}$ (bar) is the total fluid pressure at the top of the reservoir and $g = 9.80665$ m/s$^2$ is the conventional standard value of the gravity acceleration. These considerations were added to the revised manuscript at lines 421-427.

**4.3 Author's changes in manuscript related to the Specific comments 3: drawing straight lines without reasoning - 1$^{st}$ review of Referee n.1**

In the revised manuscript, Figure 7 (Figure 6 in the original manuscript) was modified based on the comment of referee 1 concerning the non-zero pressure gradient in the intermediate and deep reservoirs. Some considerations on this point were added to the revised manuscript at lines 421-427.

**5.1 Some other specific comments - 1$^{st}$ review of Referee n.1**

I only list a few, which I think are important:

(a) I think that a Google Earth snapshot is a no-go and contains much less info than, e.g., a simplified line art map showing the main geologic structures and the main geographic locations. Add a square that show the location of Fig 1b. In 1b show a scale rather than an unexplained coordinate system.

(b) Avoid self-celebrating statements in the introduction.

(c) "standard deviation" in line 98 includes what? Just the effect of variable gas analysis? What's the uncertainty of the thermodynamic analysis to obtain T and P? Possibly much larger?

(d) I can't follow the reasoning in 4.3; if you assume that (1) and (2) work well separately, than (5) should work as well as it is simply a linear combination of the two, right?

**5.2 Author's response to Some other specific comments - 1$^{st}$ review of Referee n.1**

Our answers are here below:

(a) We prefer to maintain the Google Earth map because it is very informative in our opinion. In Figure 1a we added a square showing the location of Figure 1b. Since kilometric coordinates are reported on the two axes of Figure 1b, there is no need to show a scale in Figure 1b, because it would be redundant. We added this explanation to the caption of Figure 1b.

(b) We could not find any self-celebrating statements in the Introduction, apart from the sentence at lines 81-85 of the revised manuscript perhaps: "Moreover, in this work, we use our geothermometric and geobarometric results, as well as the information from other disciplines (e.g., surface geo-volcanological surveys, data from geothermal deep wells, and geophysical investigations) to elaborate a new conceptual model of the Solfatara magmatic-hydrothermal system which extends at magmatic depths (~8 km) and represents a considerable step forward with respect to previous conceptual models." Since previous conceptual models extend to depths of a few hundred meters (Cioni et al., 1984) or 2.5-3.0 km (Caliro et al., 2007) and refer to the hydrothermal domain (lines 68-69 of the revised manuscript), our new model is a considerable step forward, at least in depth. Therefore, we did not change this sentence.

(c) The standard deviation mentioned at line 98 of the original manuscript, corresponding to line 188 of the revised manuscript, and given in Tables A1 and A2 is just the standard deviation of the CO-, $CH_4$-, and $H_2S$-equilibrium temperatures and related total fluid pressures of each time interval. The uncertainty of computed equilibrium temperatures and related total fluid pressures includes several contributions, such as the uncertainties on the third law entropy and on the enthalpy of formation from the elements which are reported in the following table (from Chase, M. 1998. NIST-JANAF Thermochemical Tables, 4th Edition. Journal of Physical and Chemical Reference Data, Monograph No. 9, 1951 pp. https://srd.nist.gov/JPCRD/jpcrdM9.pdf)

| Gas species | Error on $\Delta H_f°$ | Error on $S°$ | Upper T |
|---|---|---|---|
| | kJ/mol | J/(mol K) | K |
| $CH_4$ | 0.34 | 0.04 | 6000 |
| CO | 0.17 | 0.04 | 6000 |
| $CO_2$ | 0.05 | 0.12 | 6000 |
| $H_2$ | 0 | 0.033 | 6000 |
| $H_2O_{(L)}$ | 0.042 | 0.079 | 500 |
| $H_2O_{(V)}$ | 0.042 | 0.042 | 6000 |
| $H_2S$ | 0.8 | n.r. | 6000 |
| $SO_2$ | 0.21 | 0.08 | 6000 |

The uncertainty of computed equilibrium temperatures and related total fluid pressures also includes the uncertainties on the Maier-Kelley coefficients (which are small for gas species because their Cp values are well constrained), on the fugacity coefficients (on the fourth or third decimal figure), as well as the uncertainties related to the regression analysis performed to obtain the different geothermometric and geobarometric equations. Nevertheless, the major effect is probably the variability of analytical data.

(d) In case of overall (full) equilibrium, the CO-equilibrium temperature related to reaction (1), the $CH_4$-equilibrium temperature related to reaction (2) and the CO-$CH_4$-equilibrium temperature related to reaction (18) of the revised manuscript [corresponding to reaction (5) of the original manuscript] are the same, with deviations from a few degrees to a few dozen

degrees. For the Solfatara fluids, this overall equilibrium was observed until 1985. Afterwards, the CO-equilibrium temperature related to reaction (1) remained nearly constant, the $CH_4$-equilibrium temperature related to reaction (2) increased more and more with time, and the CO-$CH_4$-equilibrium temperature related to reaction (18) increased but much less than the $CH_4$-equilibrium temperature. The ensuing increasing difference between the three temperatures is due to CO-$CH_4$ disequilibrium. In this case, the CO-equilibrium temperature and the $CH_4$-equilibrium temperature are still meaningful, whereas the CO-$CH_4$-equilibrium temperature is meaningless. Further details are given by Marini et al. (2022) at pages 209-215, where the CO-equilibrium temperature is called T(RWG), the $CH_4$-equilibrium temperature is called T(SS4) and the CO-$CH_4$-equilibrium temperature is called T(CCC). We decided to stop to use the SS1, SS3 and SS4 acronyms and other acronyms used by Marini et al. (2022) because we were told that they were reminiscent of Nazi SS.

**5.3 Author's changes in manuscript related to Some other specific comments - 1st review of Referee n.1**

In Figure 1a we added a square showing the location of Figure 1b. Since kilometric coordinates are reported on the two axes of Figure 1b, there is no need to show a scale in Figure 1b, because it would be redundant. We added this explanation to the caption of Figure 1b.

**REFEREE N.1 – 2nd review**

**Please, note that the Referee's comments are reported in red color, the Author's responses are in blue color and the Author's changes in manuscript are in black color.**

Thank you for the reply, but I must admit that I feel that the contents of my previous review rather got confirmed than refuted. In the following, after a personal introductory remark, I will provide a second, in-depth review of several aspects of the study that will unambiguously demonstrate that key calculations are deficient with the consequence that key parts of the results on pressure lack scientific validity and that likely the temperatures as well are questionable. I will conclude with an incomplete list of answers to the authors rebuttal on points where I find that it may be beneficial for clarification.

**6.1 Introductory remark - 2nd review of Referee n.1**

Let me start with a personal remark on the reviewing effort. A significant part of the authors' rebuttal is in essence "we explained this in the book Marini et al., 2022". And that was my very point in my previous commentary: a lot of information that is essential to allow a competent, science-focused review (or simply: a read) of the manuscript is not in there. It was submitted as a *research* article and as such should contain sufficient information about the concept, methods and assumptions to allow the referee an informed preparation of checking for, quoting the authors, "scientific validity and ... errors of any kind". I estimate that a few manuscript pages on Methods with carefully chosen and explained content would have guided the referee (or just reader) well enough to decide whether and what to follow up or not by reading the book or parts of it and they would have much increased the value of the manuscript. The manuscript condensed ca. 150 *pages* in the book to ca. 15 *lines* for the "Methods" and made not a single statement on important underlying concepts, assumptions, limitations, etc. This would have been even more important as the authors claim to have introduced novel tools.

In the version that I reviewed I was essentially forced to explore the book in quite some depth to just get the basics of what was performed in the study, not even to then follow up on the science done. I take peer reviewing serious and I am happy to spend two or three full working days for an in-depth review of an interesting manuscript (as I did then). But, here, this was not leading me anywhere close to understanding the full line of arguments, approaches, assumptions etc. for this study. The book does not make it easy for the reader to detect that information, which is often hidden in very long technical discussions that frequently get as detailed as instructing the reader about how to copy values around in Excel spread sheets (I am not criticizing that this is documented, it is great it is, but it just makes the book a very hard read). For me this was the first time in more than three decades of reviewing that I encountered a paper that expects reading a book in-depth before it can be understood. In essence, this meant reviewing the (published!) book, not the manuscript. I hope the authors may understand that this felt like them making my referee job as hard as possible and that I did not feel positive about the extra, in my opinion unnecessary, time effort.

**6.2 Author's response to Introductory remark - 2nd review of Referee n.1**

We have no further comments on this part of the review in addition to what we wrote during the interactive discussion, apart from emphasizing that our book has been subject to review, no more and no less than any scientific article and, consequently, it is part of the scientific literature in its own right.

We agree with the referee that every work must be self-consistent and, therefore, we have amended our manuscript accordingly (see section 6.3 here below).

**6.3 Author's changes in manuscript related to the Introductory remark - 2nd review of Referee n.1**

At lines 104 to 182 of the revised manuscript we have explained how the geothermometers and geobarometers were derived and reported the equations used to compute the equilibrium temperatures and total fluid pressures of CO, $CH_4$, and $H_2S$.

**7.1 Additional Review - 2nd review of Referee n.1**

Now, although my duty was to review the manuscript and not the book, I felt challenged by the style and content of parts of the rebuttal (some more words on that in the answer list in the third part of this reply) and dived rather deeply into some parts of the "thermometry" and "barometry", which I already felt look suspicious when browsing the book before for understanding what was done in the study. After that new exercise (another two full working days including writing this reply), I make my recommendation even stronger:

*The manuscript should be rejected because (at least) the pressure calculations for the middle and deep "reservoirs" (and likely the associated thermometry as well, at least for the deep and likely also for the middle "reservoir") are based on a fundamentally invalid assumption and are, therefore, scientifically invalid in their current form. Even if the calculations were formally valid, the pressures would be based on an arbitrary, assumed (without constraints from data or theory) temperature-pressure path, i.e., again scientifically not constrained. As these invalid results form the very core of the study's main contribution – the "revised conceptual model" and the important conclusions resulting from it – the manuscript cannot be accepted.*

Notabene (after the experience how the previous review was perceived): I do acknowledge that a lot of honest effort and care were put into the underlying calculations in the book; nevertheless, I am convinced that a key problem escaped the authors' attention, which, unfortunately, renders the "saturation decompression path" invalid, no matter how much effort was put into the calculation concept and its execution and no matter how exciting the results might look to the authors. That I use frank words should not be confused with disrespect but as a way to keep the discussion clear about the problem and keep it focused on the science; and not be shifted again to a scientifically meaningless competition on how experienced and qualified either referee or authors might be in different subareas of magmatic-hydrothermal systems research or in applying the respective community's wording sufficiently accurately to be accepted (on that: I regret that the omission of "thermometry" in "new gas geochemical *thermometry* data" escaped my attention when proof-reading my review).

**7.2 Author's response to the Additional Review - 2nd review of Referee n.1**

Adopting the point of view of the referee, we totally agree that it is impossible to demonstrate that the Solfatara fluids follow the saturation decompression path for a brine with a certain NaCl content rather than a different decompression path. Furthermore, we emphasize that the decompression path of the Solfatara fluids is totally unknown, as it is unknown for all fluids and all geothermometers, not only the gas geothermometers, but also the water geothermometers. Therefore, following the line of reasoning of the referee, one reaches the conclusion that all fluid geothermometers are based on wrong assumptions and all geothermometric results for all hydrothermal-magmatic systems worldwide are wrong. Furthermore, since the databases of computer codes such as PHREEQC, EQ3/6, WATCH, SOLVEQ and others assume saturation conditions for pure water, also multicomponent geothermometry provides wrong results, again following the line of reasoning of the referee. In a nutshell, all the geochemical literature that deals with fluid geothermometry would be wrong and useless.

Adopting a different point of view, there is no doubt that all geothermometers are based on hypotheses, including the decompression path during the ascent of the Solfatara fluids, but in the book (not in the paper, for reasons of space) we have considered not only the saturation decompression paths but also other possible decompression paths, namely, the isenthalpic decompression path and the linear P-T decompression path. The CO-, $CH_4$- and $H_2S$ equilibrium temperatures and total fluid

pressures (obtained considering the saturation decompression paths) were selected in the paper to simplify the discussion and without pretending that the selected values are the indisputable truth. Nevertheless, if a different series of CO-, CH₄- and H₂S equilibrium temperatures and total fluid pressures (obtained considering a different decompression path) is selected, the time-trends of nearly constant CO-equilibrium temperature and total fluid pressure and increasing CH₄- and H₂S temperature and total fluid pressure are still observed with some limited differences. These considerations were added to the revised manuscript.

**7.3 Author's changes in manuscript related to the Additional Review - 2nd review of Referee n.1**

At lines 212-215 of the revised manuscript, we added a consideration on the trend of constant temperature and total fluid pressure with time in the shallow reservoir and the trends of time-increasing temperature and total fluid pressure in the intermediate and deep reservoirs, irrespective of the selected series of CO- CH₄- and H₂S equilibrium temperatures and total fluid pressures.

**8.1 Now to the core problem - 2nd review of Referee n.1**

The calculations of gas equilibration temperatures involve thermodynamics-based corrections for strong pressure effects. The pressures and resulting temperatures rely on an ***arbitrary, assumed temperature-pressure path*** along which fluids would have to flow on their way from the magmatic source to the fumaroles 8 km above where the samples were taken. The study tries to pinpoint where on that generic path individual parts of the actual Solfatara system are located.

According to the book, this assumed "saturation decompression path" is an arbitrary selection. It takes the vapor saturation pressure for a constant composition liquid (21 wt.%) in the pure $H_2O$-NaCl binary and adding a simplistic pressure correction for the presence of $CO_2$ in the vapor phase (as a side remark I wonder how one could call that "in *equilibrium* with a brine"; but see discussion of $CO_2$ solubility below). I like to emphasize again that this is an *assumed, arbitrary* path and not supported by any data; that "Giggenbach did that as well in 1987" is not a good geology- or physics-based justification. Pressures for the "reservoirs" are then "computed" by taking the computed gas equilibration temperatures and locating them on that path (I assume this was an iterative procedure).

In a nutshell, the key problems of that approach are (more, illustrated detail then further below):

(1) A vapor-saturated 21 wt% NaCl brine that the calculations are based on simply doesn't exist above ca. 590 °C (the critical temperature for that composition in the binary $H_2O$-NaCl system[1] [footnote 1: btw., Giggenbach (1987) also stated that the validity is limited to ca. 600°C]), so the calculation is meaningless for all inferred temperatures for the deep "reservoir" and for the hottest of the middle one.

(2) Instead, above that temperature, an $H_2O$-NaCl fluid on the two-phase liquid-vapor surface in that system is a vapor itself.

(3) As a major consequence, and as the critical line for 21wt% NaCl (with respect to water) in the ternary $H_2O$-NaCl-$CO_2$ system is nearly isothermal at those 590 °C: if $CO_2$ is present, the fluid at such high temperature and pressure conditions would be a homogeneous, "supercritical", dense vapor-1 btw., Giggenbach (1987) also stated that the validity is limited to ca. 600°C like fluid for which no unique P-T coordinates can be constrained without serious further assumptions.

(4) For this dense vapor-like fluid, due to the presence of significant NaCl and even if the pressures were correct, the fugacity coefficients calculated by the authors via a Peng-Robinson equation of state for a system without NaCl likely are in error (and, as far as I recall, standard P-R equations of state are not very well suited to include electrolytes) and so would be the gas thermometry.

(5) These points render the temperature-pressure calculations for the deep "reservoir" in their present form invalid. How wrong they are in terms of numbers cannot be estimated in a simple way.

(6) Moreover, the approximation of a carbonic vapor phase pressure in "equilibrium" with a vapor-saturated $H_2O$-NaCl liquid is only acceptable if the mutual solubilities of $CO_2$ in the aqueous solution and of water in the carbonic vapor phase are small such that the saturation condition spreads only over a small pressure interval and remains close to the saturation line in the aqueous solution binary.

(7) This holds true for typical geothermal systems at less than ca. 300ish °C and has, therefore, indeed be applied to such systems. As soon as the mutual solubilities are non-negligible (starting at T > ca. 300ish °C, some people might claim 350ish °C, it really doesn't matter much) this is not possible anymore as "saturation pressure" of the liquid is not uniquely defined and phase compositions will change with phase proportions for a given bulk composition.

(8) Rather, the liquid saturation pressure will now also be a function of the additional $CO_2$ content of liquid, i.e., there is a whole saturation surface over very wide ranges of pressure and for temperatures up to the near-isothermal (the above 590 °C) critical curve of the pseudo-binary "21 wt% aqueous NaCl solution + $CO_2$" instead of a single saturation line.

(9) Side remark: so much about the authors' statement "there is no need for an "*adequate solubility model for the gases in such a brine*"."

(10) Given the temperature range mentioned in the previous point, also both the temperatures and pressures in the middle "reservoir" are likely highly questionable.

(11) Now this in some more detail, illustrated by published figures of the respective phase diagrams.

Let's start with the issue of 21% vapor-saturated brine with the diagram by Driesner and Heinrich (GCA, 2007). I have highlighted the 20 wt.% curve (let's stay with that as a convenient proxy for the author's choice for rest of the discussion) in yellow and one can see how it intersects the critical curve at ca. 590 °C, i.e., at lower temperatures it is on the liquid side of the twophase vapor+liquid surface, at higher temperatures on the vapor side. So, there is no such thing as a 20 wt% saturated liquid at temperatures higher than this. One might now argue "ok, let's simply take a higher salinity such as 33.5% used elsewhere in the manuscript and the problem is gone".

[Figure]

Well, before we come to that point let's first look at a relevant P-T projection of parts of the $H_2O$-NaCl-$CO_2$ phase diagram, taken from Schmidt and Bodnar (GCA, 2000). The most relevant feature in that diagram is the near-vertical line labeled "20 wt% NaCl". This is the critical line for the pseudo-binary "20 wt% NaCl/80wt% water + $CO_2$". At temperatures higher than that curve any fluid in the ternary $H_2O$-NaCl-$CO_2$ system that has a 20:80 wt% ratio of NaCl and water and has some $CO_2$ content will exist as a single-phase, "supercritical" fluid (highlighted yellow, extent to bottom not well known and not well understood). Red added curve is the approximate location of 20 wt% NaCl on V+L surface (solid curve: liquid, dashed curve: vapor), added blue curve is threephase vapor+liquid+halite, both in the $H_2O$-NaCl binary. In that binary, below the dashed red curve, there would be liquid with >20% NaCl in coexistence with a vapor <20% NaCl but no fluid phase with 20wt% can exist there. What happens below the red curve in the ternary is not well known, in particular if and where it comes to halite saturation as that is now a divariant surface in T-P rather than a univariant curve and may actually start at much higher pressures than in the binary (see, e.g., Anovitz et al., GCA, 2004).

*I think it is obvious from the above that the authors' pressure calculation method has no foundation at temperatures above ca. 590 °C and that, therefore, **the pressure calculations for the deep reservoir are simply invalid**. An additional, very important point now is that, at the higher temperature conditions, appreciable amounts of NaCl would be present in the vapor phase (in different concentrations for different T-P) rendering the used Peng-Robinson e.o.s. version non-applicable. For example, already at 500°C and 500 bar, Anovitz et al. (GCA, 2004) found up to several mole percent NaCl in the vapor and this should be expected to increase with higher temperature for the simple thermodynamic requirement that it needs to converge with the aqueous liquid's concentration towards the upper critical end curve. **Therefore, even the thermometry is seriously questionable for temperatures of the late part of the time series for the middle reservoir and for all temperatures of the deep reservoir.***

[Figure]

The full implications of the diagram above are nontrivial to comprehend as it is a 2D projection of elements of a 4D diagram. For me, the most intuitive and plausible image emerges when taking the $H_2O$-$CO_2$ phase diagram in 3D as a reference, e.g., from Diamond (Lithos, 2001). What is labelled there "Upper critical curve of binary" is the "0 wt.% NaCl curve" in the previous diagram, i.e., this diagram could roughly be imagined as an extension of the previous diagram to the left with $X_{CO2}$ as an axis going into the diagram. The "Upper critical curve of binary" limits the chimney-like carbonic-aqueous immiscibility region to high temperatures. We could imagine this as a crude guide for how the "immiscibility chimney" expands and shifts to higher temperature when moving to the pseudo-binary "20 wt% NaCl/80wt% water + $CO_2$" (most elements in the previous diagram): also there, it'd be a "chimney" with significant mutual solubility on the steep saturation surfaces (again: not lines) on the aqueous and carbonic sides and limited to high temperatures by the "20 wt% NaCl" line in the second diagram.

Now, this also explains why choosing saturation of a 33.5% "brine" would not do the job: although the upper critical line of that pseudo-binary would be at higher temperatures (about 1000°C) the mutual solubility would still eliminate the choice of a single saturation pressure and the saturation surfaces would still be steep, i.e., saturation pressure ranges are very large. Even worse (see also Anovitz et al., 2004, their Figure 4): at a given temperature and pressure varying $X_{CO2}$ of the saturated vapor will make the NaCl content of the saturated liquid change significantly. *So, the authors barometry approach also breaks down at temperatures relevant for the middle reservoir.*

Unfortunately, for the lack of more experimental data, we don't know to how low temperatures this may be the case but I think it would be good scientific practice to cautiously assume that Anovitz' 500°C is not a fortuitous hit of the lower temperature limit of the problem. The community has not yet fully explored and understood all the phase relations in the $H_2O$-NaCl-$CO_2$ ternary, namely on the carbonic -side or at the low pressure end. Yet, with the above (which necessarily was highly simplified and incomplete) I think we have understood enough that the negative assessment about the (in)validity of the authors' approach is robust.

In terms of the manuscript's main results this means graphically:

[Figure]

(12) To add on this: the authors could have suspected a serious problem already from their pressure diagram. Fluid pressures as high as 2.4 times lithostatic (or, in absolute values overpressured by up to 1800 bar) and fluid pressure gradients similarly excessive, in particularly in the hot, ductile regime, would be considered unrealistic by many (if not most) people dealing with such problems for reasons laid out, e.g., by Cox (Geofluids, 2010) or discussed in the Weis et al. (2012) paper (or follow-ups on that) that the authors considered not relevant in their rebuttal. In essence, (ductile) rocks cannot sustain such high fluid overpressure, the values are very unrealistic. Since a lot of the implications of the model hinge on that, the author may want to acquaint themselves more with rock failure related to fluid pressure (the Cox paper is a good introduction).

(13) Now, to be "constructive" after all this: The phase relations discussed above also propose a possible way how the "middle reservoir" can indeed be the source of bradyseism. If the fluid released at depth were a "supercritical" $H_2O$-NaCl-$CO_2$ mixture then upon decompression it may hit the "immiscibility chimney" displayed in the $CO_2$-related phase diagrams above. For a suitable parameter combination this should result in sufficient pressurization.

Similarly, the overpressure problem goes back to the assumed temperature-pressure "saturation decompression path" and the fact that it is simply arbitrary, not founded on any data or defendable theory, and leads to too high pressures. Given the above discussions on the phase diagram it seems likely to me that a plausible decompression path can be constructed that implies lower, ore [more] realistic pressures at depth. Then, "lithostatic plus a defendable overpressure" in the deep "reservoir" would be a good starting point for revising the decompression path (from the source upward rather than vice versa) of the "revised conceptual model".

*(14) Up to here, I consider this reply as essential information for the editor. I hope the explanations are not "too nerdy" to be evaluated for the decision process (if they are, please request an explanation in simpler words) and I hope that also the authors can accept my criticism given their own statement "every author should be allowed to publish her/his results, provided that they have scientific validity and are not affected by errors of any kind". I think I clearly showed above that the study contains serious problems of both kinds.*

(15) In the following I take the freedom to share some reflections with the authors as on what other aspects I think they may want to consider and/or re-think when re-visiting their model. This will also reply to some selected parts of their rebuttal. I will not answer to all points as I think that the discussion has already reached a point of getting lost in unnecessary back-andforth. I don't claim to own the truth here but I would expect my thoughts to be considered a bit more seriously (1) as possibly valuable when revisiting the model and (2) for what to pay attention to beyond the gas chemistry point of view.

**8.2 Author's response to Now to the core problem - 2$^{nd}$ review of Referee n.1**

For what concerns the sentence '*that "Giggenbach did that as well in 1987" is not a good geology- or physics-based justification*' we underscore that it is not our intention to point out that what we did is correct because Giggenbach had already done it in 1987. Our intention is to give Caesar what belongs to Caesar and to give Giggenbach what belongs to Giggenbach, who had the great merit to understand the complexities of natural systems and to develop geochemical tools easy for everyone to use. Werner Giggenbach was an unsurpassed master in this aspect.

(1), (2), and (3) We are aware of the upper temperature limit (ca. 590-600°C) of the saturation decompression path of a vapor in equilibrium with a 21 wt% NaCl brine. In fact, at pages 290-292 of the book we wrote that the RWG and SS4 geoindicators for the saturation decompression path of Solfatara fluids involving a brine containing 21 wt% NaCl and the related vapor phase can be used up to 600 °C. The maximum SS4 computed temperature, that is, the maximum $CH_4$-equilibrium temperature, 645°C, is not much higher. To be noted that, in our manuscript we wrote clearly that the $CH_4$-equilibrium temperature and total fluid pressure refer to the intermediate ("*middle*" in the referee's words) reservoir. In contrast, the referee writes that "*the calculation is meaningless for all inferred temperatures for the deep "reservoir"* but the $CH_4$-equilibrium temperature does not apply to the deep reservoir.

(4) There are many papers dealing with the use of the Peng-Robinson EOS, with some adjustments/modifications, to model vapor + liquid equilibria in electrolyte solutions and specifically in systems including water, sodium chloride, and carbon dioxide (e.g., Nighswander et al. 1989; Kwak and Anderson 1991; Søreide and Whitson 1992; Sieder and Maurer 2004; Baseri and Lotfollahi 2011; Hou et al. 2013; Appelo 2015; Li et al. 2015; Zuo et al. 2024). In particular, Appelo (2015) used the Peng–Robinson EOS to compute reliable fugacity coefficients for $CO_2$. He noted that the ion interaction parameters given by Harvie et al. (1984) for $CO_2$ at 25 °C are valid for calculating the $CO_2$ solubility at high temperatures, pressures and salinities. Furthermore, use of the Peng-Robinson EOS is totally justified because we obtained comparable values, for the fugacity coefficients of $CO_2$ and $H_2O$, utilizing the GERG-2008 EOS (Kunz and Wagner 2012) and the EOS of Gallagher et al. (1993). To the best of our knowledge, fugacity coefficients were not considered in gas geo-indicators proposed so far in the scientific literature, although use of fugacity coefficients is absolutely necessary at the high temperatures and pressures of $CH_4$ and $H_2S$ equilibration. To be noted also that, irrespective of the decompression path, the fugacity coefficients of non-polar gases (i.e., $CO_2$, $CO$, $CH_4$, $H_2$, and $H_2S$) increase with increasing P, T, deviating gradually from unity, whereas the fugacity coefficient of $H_2O$ decreases with increasing P, T, departing progressively from one. The ensuing practical implication is that the analytical mole fraction ratios of non-polar gases (e.g., $CO/CO_2$, $CH_4/CO_2$, and $H_2S/CO_2$) may be utilized in geothermometric-geobarometric functions without incurring excessive errors, whereas use of the analytical $H_2/H_2O$ mole fraction ratio leads to significant errors in computed equilibrium temperatures and pressures. Summing up, use of the Peng-Robinson EOS is completely justified to compute fugacity coefficients, while results of gas-geothermometers are affected by serious errors if fugacity coefficients are not considered in their calibration.

(5) This critique of the referee is not justified because (a) the T-P conditions of the deep reservoir are described by the $H_2S$ equilibrium temperature and total fluid pressure and (b) the $H_2S$ equilibrium temperature and total fluid pressure were computed considering the decompression path of Solfatara fluids involving a brine containing 33.5 wt% NaCl (not 21 wt% NaCl as erroneously considered by the referee) and the related vapor phase. Both things are written clearly in our manuscript. These functions can be used up to a maximum pressure of 3627 bar and a maximum temperature of 1000 °C (see pages 335 of the book). In practice, the maximum $H_2S$ equilibrium temperature and total fluid pressure computed for the Solfatara fluids, 1087°C and 3408 bar, are slightly higher and somewhat lower of these applicability thresholds, respectively. So, there is nothing wrong in the temperature-pressure calculations for the deep reservoir.

(6) Irrespective of some inaccuracies/typos of the referee comment, the question is if it is acceptable or not to refer to the saturation line of the aqueous solution binary $H_2O$-NaCl for a ternary $H_2O$-NaCl-$CO_2$. As shown in Figure 5 of Giggenbach (1987), this approximation is acceptable if the concentration of $CO_2$ in the brine and the concentration of NaCl in the vapor phase are both sufficiently low. On this figure, in the left-hand diagram, we have reported both the logarithm of the fugacity of water for a brine containing 21 wt.% NaCl in equilibrium with a vapor phase with a mole fraction of $CO_2$ equal to 0.05 (diamonds and line of blue color) and 0.40 (squares and line of red color), corresponding to the minimum and maximum mole fraction of $CO_2$ for the Solfatara fluids.

[Figure]

In the right-hand diagram, we have reported both the logarithm of the fugacity of water for a brine containing 33.5 wt.% NaCl in equilibrium with a vapor phase with a $CO_2$ molar fraction equal to 0.05 (upward pointing triangles and line of sky-blue color) and 0.40 (downward pointing triangles and line of magenta color). In both diagrams, the two lines with symbols are close to each other and their acceptably good linearity holds true up to 600°C for the brine containing 21 wt.% NaCl and up to 1000°C for the brine containing 33.5 wt.% NaCl. These lines are not positioned exactly between the vapor and brine envelops reported in Figure 5 of Giggenbach (1987) because Giggenbach used the data of Sourirajan and Kennedy (1962) for the system $H_2O$-NaCl, whereas we used the data of Tanger and Pitzer (1989) for the brine containing 21 wt% NaCl and the data of Driesner and Heinrich (2007) for the brine containing 33.5 wt% NaCl.

Moreover, Giggenbach took the fugacity coefficients, presumably of pure gases, from Ryzhenko and Volkov (1971) and Ryzhenko and Malinin (1971) and stated that "*for a separate gas phase containing water vapor as the major component, $x_{H_2O} > 0.8$, maximum pressures [not given] are controlled by the formation of an aqueous phase. Under these circumstances the isomolar ratios of activity coefficients $\gamma_{SO_2}/\gamma_{H_2S}, \gamma_{CO}/\gamma_{CO_2}, \gamma_{CO_2}/\gamma_{CH_4}, \gamma_{N_2} \cdot \gamma_{H_2O}/\gamma_{NH_3}^2, and \gamma_{H_2}/\gamma_{H_2O}$ [γ's are actually fugacity coefficients] never deviate much from unity, and uncorrected mole-ratios may be used without incurring excessive errors*". In contrast, we computed the fugacity coefficients of gas species in gas mixtures using the Peng-Robinson EOS. In spite of these minor deviations between our $f_{H_2O} - 1/T(K)$ relations and the vapor and brine envelops reported in Figure 5 of Giggenbach (1987), the clear description given by Giggenbach in his 1987 paper is still valid and is reported here: "***Three fluid stability regions can be distinguished: (1) a single liquid (brine) at comparatively low temperatures and high pressures; with increasing temperature or decreasing pressure a boundary ("liquidus") is reached to a region (2) where a vapor phase will coexist with a saline brine; further increase in temperature, or reduction in pressure, will lead to complete evaporation***

*of increasingly saline brines and to a boundary ("vaporus") delineating the stability region (3) of a single vapor phase (in the presence of solid NaCl).*" In the Solfatara hydrothermal-magmatic system, the most probable condition is the coexistence of a vapor phase with a saline brine, because the occurrence of the other possible conditions is highly unlikely. In fact:

(a) On the one hand, the occurrence of a single liquid (brine) at comparatively low temperatures and high pressures is at variance with the huge amount of heat released from the magma batch and transferred to the overlying hydrothermal part of the system.

(b) On the other hand, a single vapor phase coexisting with solid NaCl might occur in depressurized vapor-cored magmatic systems (Reyes et al. 1993), such as Vulcano Island, Italy (Cioni and D'Amore 1984), and many systems of Indonesia (Abiyudo et al. 2016) and The Philippines (Reyes et al. 1993; Ramos-Candelaria et al. 1995; Apuada and Sigurjonsson 2008), but it is at variance with the current pressurization and related ground uplift of the Solfatara hydrothermal-magmatic system.

(7) These theoretical considerations of the referee are correct but they do not apply to the Solfatara hydrothermal-magmatic system, in its current state, as discussed here above. Things might have been different in the past and might be different in the future, as discussed in Chapter 11 of our book.

(8) We agree with the referee.

(9) We could not find the statement reported by the referee neither in the book nor in our paper. Our opinion on this topic is exactly the opposite, in that, we believe that there is a need for an EOS able to predict P-V-T-X data, immiscibility/phase equilibria, solubilities, and activities in the ternary $H_2O$-NaCl-$CO_2$ system.

In fact, at pages 122-123 of the book, we wrote that "*… the binary $H_2O$-NaCl system is considered here, instead of the ternary $H_2O$-NaCl-$CO_2$ system, to describe brine-vapor coexistence both: (1) to avoid complicating the calculations too much and (2) because of the incomplete knowledge of the P-V-T-X properties of the $H_2O$-NaCl-$CO_2$ system, which is largely based on the experimental work by Gehrig (1980) and the P-V-T-X data obtained through the synthetic fluid inclusion technique (e.g., Kotelnikov and Kotelnikova 1990; Frantz et al. 1992; Johnson 1992; Shmulovich and Graham 1999; Schmidt and Bodnar 2000).*

*In addition to the determinations relevant for the binary $H_2O$-$CO_2$ system (see Sect. 2.1), Gehrig (1980) measured the molar volume and defined the immiscibility boundaries for the ($H_2O$ +6 wt.% NaCl)-$CO_2$ pseudobinary system at pressures from 0 to 3000 bar and temperatures from 200 to 560 °C using a high-pressure variable volume autoclave. The data of Gehrig (1980) represent the main experimental foundations of the modified Redlich–Kwong EOS of Bowers and Helgeson (1983) as well as of the EOS of Duan et al. (1995), both for $H_2O$-$CO_2$-NaCl fluids. Bowers and Helgeson (1983) computed fugacity coefficients of $H_2O$ and $CO_2$ at temperatures of 400, 450, 500, 550, and 600 °C and pressures of 500, 1000, and 2000 bar and used these fugacity coefficients together with solubility data to establish the compositions of the coexisting immiscible phases. However, the EOS of Bowers and Helgeson (1983) was questioned by Duan et al. (1995). According to Duan et al. (1995), their EOS predicts P–V-T-X data, immiscibility/phase equilibria, solubilities, and activities with an accuracy similar to that of the experimental data at temperatures from 300 to ~1000 °C and pressures from 0 to 6000 bar for NaCl concentrations to ~30 wt% (relative to NaCl + $H_2O$) and to ~50 wt% with lower accuracy. However, the EOS of Duan et al. (1995) was considered poorly reliable by other authors (e.g., Schmidt and Bodnar 2000).*

*Given this situation, the binary $H_2O$-NaCl system was considered in this work, instead of the ternary $H_2O$-NaCl-$CO_2$ system, to describe the equilibrium coexistence of the vapor phase with a brine containing either 21 wt% NaCl (data from Tanger and Pitzer 1989) or 33.5 wt% NaCl (data from Driesner and Heinrich 2007).*"

(10) As explained above, there is nothing questionable about the temperatures and pressures in the intermediate (middle) reservoir that we estimated for the saturation decompression paths and other possible decompression paths of Solfatara fluids.

(11) This long discussion adds nothing to what the referee already pointed out above. The gist of the discussion is that the relation for a brine containing 21 wt% NaCl and the related vapor phase can be used up to 590-600°C whereas the relation for a brine containing 33.5 wt% NaCl and the related vapor phase can be used up to 1000°C, with more uncertainties. Anyhow, the $CH_4$ equilibrium temperature which refers to the intermediate reservoir are generally lower than the 600°C threshold and moderately higher in few cases while the $H_2S$ equilibrium temperature which refers to the deep reservoir are generally lower than the 1000°C threshold and moderately higher for few samples. Again, the referee confuses the two temperatures and the two reservoirs.

(12) The referee evidently considers the overpressure at the top of the deep reservoir (ca. 6.5 km depth), where the maximum value is 1950 bar, based on the data of Bocca Grande, and 2078 bar, based on the data of Bocca Nuova (somewhat higher than 1800 bar, as written by the referee), corresponding to 2.47 and 2.56 times the external pressure (overburden). Considering that the fluid pressure at the top of the magma chamber (ca. 8 km) is 2879 bar, we do not see why the overpressure at the top of the deep reservoir should be considered unrealistic. Actually, we are more concerned with the overpressure at the top of the intermediate reservoir (2.7 km depth), where the maximum value is 1064 bar, based on the data of Bocca Grande, and 1273 bar, based on the data of Bocca Nuova, corresponding to 4.94-5.72 times the external pressure (overburden). Nevertheless, Chiodini et al. (1992) reported that the sector of the aquifer involved in the hydrothermal explosion occurred on April 15[th], 1990 at Guagua Pichincha (Ecuador) could have reached a fluid pressure higher than 5 to 7 times the external (hydrostatic) pressure, which was sufficient to trigger the hydrothermal explosion. Therefore, even a $P_{fluid}/P_{ext}$ ratio of 4.94-5.72 should not lead to the hydrothermal explosion.

(13) We thank the referee for this suggestion but we prefer to keep our interpretation.

(14) We have no further comments on this part of the review in addition to what we wrote during the interactive discussion.

(15) We have no further comments on this part of the review in addition to what we wrote during the interactive discussion

**8.3 Author's changes in manuscript related to Now to the core problem - 2[nd] review of Referee n.1**

At lines 159-160 of the revised manuscript, we wrote that the relations derived for the saturation decompression path of a vapor in equilibrium with a 21 wt% NaCl brine, that is, Eqns. (9), (10), and (11) can be applied up to 600 °C.

At line 180 of the revised manuscript, we wrote that the relations derived for the saturation decompression path of a vapor in equilibrium with a 33.5 wt% NaCl brine, that is, Eqns. (15), (16), and (17) can be applied up to 1000 °C.

The considerations supporting the coexistence of a vapor phase with a saline brine as the most probable condition in the Solfatara hydrothermal-magmatic system were reported in the revised manuscript at lines 130-139.

**9.1 Various points - 2[nd] review of Referee n.1**

(A) As an add-on to the discussion above, I would like to suggest to the authors to also think a "decompression path" indeed from the deep source upward and not top down from the surface (which is rather a "pressurization path") although the gas data were collected there. Phase changes will modify $CO_2$-$H_2O$ ratios upon ascent and the degree and cause of this modification cannot be quantified in the top down way; e.g., for your shallow isenthalpic paths, for example, when going down you hit the $H_2O$-$CO_2$ saturation on the (carbonic) vapor side (on a rather flat surface that forms the "chimney's" bottom; such vapor can be equally well result from temporary saturation and condensing out some minor liquid (due to the "bulge" in the vapor enthalpy curve as you discussed) or from the (boiling-like) exsolution of a mass-wise minor vapor phase from an $CO_2$-poor but mass-wise dominant aqueous liquid. What happens below that saturation depth is, therefore, ambiguous and cannot be constrained from surface data as presented. Also, for this problem, I missed a clear statement of what your assumptions were

for the top down approach regarding the fluid phase evolution with depth. Such a statement would have allowed readers/referees to test your hypothesis; leaving it out – or not at least discussing this / not formulating a hypothesis – is not good practice.

(B) Again on top down vs. bottom up: that was also the main reason for recommending Einaudi et al. and others for sulfur or Weis et al. for thermo-hydrology. I did not assume that the former were up to date with respect to the latest in volcanic gas chemistry but these papers think the chemical (or fluid flow, resepctively) process from the source to the surface and this clearly adds value. Namely, Einaudi et al. highlight that the fluid passes through different redox and fS2 conditions along its path as exemplified by successive mineral assemblages observed, which will, among others, also alter sulfur speciation and fugacity/concentration (and therefore the values of your H2S thermometer) on the way up. To me this looks geologically and geochemically much more logically and advanced than the assumption of a single mineral-fluid reaction fixing it right away at 7 or so km depth. Whether the Weis model was inspired by porphyry deposits doesn't play a role; to me, it models a generic magmatic fluid release process that was then interpreted by those authors for its relevance for pophyry-Cu deposits.

A bit more about the reactive path of sulfur: for the deep parts, the main window of action for $SO_2 + H_2O$ reacting are believed to be below 500ish °C or so (gas redox buffer followed by disproportionation along cooling path, if I remember correctly). This is at lower temperatures than your $H_2S$ thermometer equilibration and should therefore be assessed for a possible impact on $H_2S$ concentrations on the way to the surface.

(C) Fournier vs. Weis et al.: the Weis model replaced the Fournier concept in that it explains the lithostatic to hydrostatic transition as a natural consequence of degassing magmatic fluids having the dual role of heating the rock overlying the magma to ductile temperatures and impermeable behaviour and, in turn, of transiently breaking those heated rocks due to pressure build-up to allow the temporary release of fluids; this magmatic-hydrothermal domain than naturally transitions into a classical geothermal system further up; there is no need for self-sealing by silica, just heat + fluids do the job already, matching many features of natural systems (fossil or active). Let Occam's razor do its job here.

(D) Lupi/Weis: well, that was quite a cheap trick referring to what I did not point at (the trigger mechanism, which indeed can be questioned) and then trying to make me look ridiculous by criticizing that. However, I appreciate that you didn't fully loose humor over my review. I was pointing at the overpressure waves rising in the magmatic-hydrothermal plume and these – if I remember that correctly from in-depth discussions with Weis – happen on time scales of years or 10s of years; i.e., they are highly relevant. Furthermore, you could learn from Weis et al. how $H_2O$-NaCl phase relations (unfortunately, no $CO_2$) evolve with space and time in such a system to come back to the main problem of your calculations.

(E) It were these last few points that made me make the comment on "apparently from a volcanology background" as the way the manuscript is written it reads like a naive "what comes out of the fumaroles is what is at depth". There is nothing about "second class scientists" implied but rather should highlight the impression of a surface data-biased view on fluid processes in the deep parts of magmatic-hydrothermal systems (btw. understood as systems dominated by magmatic fluids). A lot of valuable information on the latter is available (mostly from the economic geology community) that could have informed the conceptual model design with quite some advantage but was not considered. In hindsight, I admit that the statement could be misinterpreted and hope these remarks clarify that.

(F) Convective zero temperature & zero gradient pressure profiles: another one that was apparently intended to make me look like a beginner by referring to an introductory book. My point was that your "convective" vertical temperature profiles in the reservoirs are an ad hoc invention based on no data and should, therefore be declared as such or be justified. BTW: zero pressure gradient vaporstatic columns would not convect. Regarding that zero pressure gradient, for the fun of it, let's take you own data, for example the Oct10-Jun12 $H_2S$ equilibration conditions: a pure water vapor at 830 °C and 2157 bar would have a density of 432 kg/m$^3$. In the $CH_4$ equilibration reservoir for the same period one would have 412 kg/m$^2$ [kg/m$^3$]. This is >0.4

times a cold hydrostatic gradient, far away from zero gradient. So, no point to make me look ridiculous when your own data proof you wrong. In my group it is a routine process to perform such obvious checks before adding conceptual figures to a publication.

(G) "inferred geology". Let me cite the book, page 40: "In particular, according to Zollo et al. (2008), the **_inferred_** schematic stratigraphy comprises, from top to bottom (Fig. 10):". So, please don't bash me if I use your words.

(H) Calcite-$H_2S$: you are right, geology rules. I should have expressed much clearer that I was referring to the effect of reactive transport on sulfur content. There, I don't agree that the absolute concentrations don't matter because small concentrations may easier experience massive relative modification (reactive transport is always a competition between equilibrium constant and actual masses present) than bigger ones. Side remark, lines 69-71 of rebuttal: there is no such thing as a "strong acid" at those conditions; acids known to be strong at ambient conditions become weak in the low dielectric constant aqueous solvent at those conditions. As illustration: according to Supcrt, if one trusts it, the logK for $HSO_4^- = SO_4^{2-} + H^+$ in the temperature range of 500 to 800°C and densities from 0.4 to 0.6 g/cm3 is in the range of -8 to -11 ...

(I) Finally, I think my doubts about the compatibility of "reservoir" and equilibration and a structural transport highway remain valid but there is no point discussing this further here. This applies also for all other points I may not have responded here, too.

(J) I would like to conclude with stating that I start to appreciate the egusphere discussion format. Although I was annoyed by the extra effort compared to what it could have been if the manuscript were properly prepared, I think such discussions can be very helpful and help bridging gaps between different communities, one of which became very obvious here.

**9.2 Author's response to Various points - 2nd review of Referee n.1**

(A) Since the gas data were collected at the surface, we think that the top-down approach (using the referee's words) is more suitable than the opposite bottom-up approach. If one wants to consider the effects of partial water condensation (as done in some papers of Chiodini and coworkers), it is necessary to introduce an assumption which cannot be tested. So, we prefer to use the analytical data without any correction to avoid to complicate the problem rather than to simplify it. Furthermore, in the book we considered also the chemical kinetics of the reactions involving $H_2O$, $CO_2$, $CO$, $CH_4$ and $H_2$, which has been the subject of several studies, and we applied a simplified reaction path model, simulating the heating of the Solfatara fluids collected at the surface. The equilibrium temperatures computed by the reaction path model compare with those given by simple gas geothermometers, within acceptable differences. Thus, gas geothermometers work well, at least for the intermediate reservoir.

(B) We have nothing to add to our previous rebuttals to these referee's considerations apart from a comment on $H_2S$, in that possible reactions causing significant changes of this gas species were considered in the book. The fugacity of gaseous $S_2$, $SO_2$ and $COS$ were computed as a function of the $CH_4$ equilibrium temperature (called SS4 equilibrium temperature in the book) and resulted to be negligible. We have also considered the pyrite-pyrrhotite, pyrite-fayalite-quartz, pyrite-magnetite, and pyrite-hematite equilibria, but the computed equilibrium temperatures have no physical significance, suggesting that the $H_2S$ concentration of Solfatara fluids is controlled by other reactions. Considering the presence of carbonate rocks at depths from ~ 4 km to ~7.5 km, we assumed that this reaction is:

$CaSO_{4(s)} + CO_2 + 4 H_2 = CaCO_{3(s)} + H_2S + 3 H_2O$.

(C) In our opinion, the Fournier model is more realistic than the Weis model, also considering the occurrence of quartz in the deepest section of geothermal well San Vito-1.

(D) We have no further comments on this point of the review in addition to what we wrote during the interactive discussion.

(E) We have no further comments on this point of the review in addition to what we wrote during the interactive discussion.

(F) Please find the rebuttal to this comment in section **4.2 Author's response to the Specific comments 3: drawing straight lines without reasoning - 1st review of Referee n.1**

(G) We have no further comments on this point of the review in addition to what we wrote during the interactive discussion.

(H) We have no further comments on this point of the review in addition to what we wrote during the interactive discussion.

(I) We have no further comments on this point of the review in addition to what we wrote during the interactive discussion.

(J) We have no further comments on this point of the review in addition to what we wrote during the interactive discussion.

**9.3 Author's changes in manuscript related to Various points - 2nd review of Referee n.1**

None.

**REFEREE N.2**

**Please, note that the Referee's comments are reported in red color, the Author's responses are in blue color and the Author's changes in manuscript are in black color.**

**1.1 Review of Referee N.2**

The research conducted by Marini et al. presents a novel interpretation of four decades of data from the Solfatara geochemical database, utilizing new geoindicators and aiming to offer potential predictions of future scenarios. The subject matter is of significant scientific relevance, particularly as the Solfatara magmatic-hydraulic system holds considerable interest, not only for the scientific community engaged in the study of this system, but also due to the concerns regarding the potential evolution of the current bradyseism towards eruptive scenarios, which may have substantial societal implications for the population residing in its vicinity. However, despite the relevance of the topic, I feel that the paper in its current state is not ready for publication, and that some concerns about the manuscript should be addressed before publication.

In the following, I would like to point out two reasons I consider to be of great importance, and which should be reconsidered by the authors before a possible publication. Furthermore, in an attached file, I have integrated the comments and suggestions next to the relative text which are aimed at improving the readability of the text and the completeness of the information according to my personal vision.

In summary, the two most important aspects of the manuscript I believe should be reviewed in order to improve the quality of the paper before its possible publication are as follows:

(1) The heart of the work consists in the reinterpretation of geochemical data based on the use of new geoindicators. Unfortunately, the text does not contain the information necessary to understand and interpret these geoindicators, which, as the paper is set up, must be well understood to ascertain the fields of applicability and potential limitations. This makes the paper not immediately understandable nor directly usable by the scientific community (after reading this document, can the use of these new geo-indicators be replicated in other cases?), unless one looks for the source, however this is contained in a book that precludes easy acquisition for many. I myself have not been able to find it, as I explained in the attached file. However, the authors of this manuscript are also the authors of the monograph; therefore, it is my opinion that they could easily integrate the text with a specific section describing the new geoindicators, which would certainly enhance the text. As a side effect, I was only partially able to follow the exchange between the first reviewer and the authors, due to the asymmetry of information caused by my lack of knowledge of the content of the monograph cited.

(2) Another significant concern relates to the capability to predict specific scenarios following the analysis and model interpretation. I acknowledge that the term "prediction" can have various meanings; however, in scientific literature, it is generally accepted to convey the notion of "the expectation of an occurrence under certain conditions". Typically, this expectation is quantified and qualified through numerical analysis (statistical or probabilistic), which lends credibility to the process of predicting future events or outcomes. Prediction can be viewed as part of hypothesis testing, where one formulates predictions as components of hypotheses. These predictions can then be empirically tested through experiments or observations to confirm or refute the hypotheses. However, this concept is not clearly articulated in the text. Instead, the authors present reasonable scenarios that could evolve in different directions, potentially even oppositely, without providing any arguments or analyses to support specific predictions. The text offers only a description of various possible scenarios, lacking a basis for making predictions. In my opinion, this aspect of the paper needs to be reconsidered, given the significant importance of the ability to make predictions, especially for local authorities

responsible for managing risk in a densely populated area like Campi Flegrei. Therefore, I suggest that the authors revise the use of the term "prediction" from the title onward, modifying the text accordingly or providing sufficient justification to support the possibility of predicting potential risk scenarios.

I hope that the critical reading of the manuscript that I propose in its current state can be a constructive stimulus for the authors, regardless of the outcome of the publication.

**1.2 Author's response to the Review of Referee N.2**

We thank the referee for the constructive attitude. The Author's response to the comments and suggestions in the annotated manuscript are treated in "Section 2. Comments and suggestions posted by Referee N.2 in the annotated manuscript". Here below we report our responses to the two major comments of Referee N.2

**(1) First major comment.** We agree with the referee on this point and we have expanded considerably the section "2 Method" in the revised manuscript, in order to provide the information needed to understand the new gas-geoindicators of Marini et al. (2022), including their applicability and limitations. It is true that, in our original manuscript, the considerations on these new gas-geoindicators are limited to a minimum, but we thought that the book of Marini et al. (2022) was part of the scientific literature and was easily acquirable as any other scientific paper. We are sorry for the difficulties encountered by the referee and we are ready to email to the referee a copy of our e-book.

(2) **Second major comment.** There are two distinct aspects in the second major comment of Referee N.2, one terminological, the other substantial. For what concerns the terminological aspect, we acknowledge that the term "prediction" is generally accepted to convey the notion of "the expectation of an occurrence under certain conditions", in the scientific literature, although it might have different denotations. In the original version of the manuscript, we used the term "prediction" in a merely qualitative sense, without any statistical or probabilistic connotation, but we agree with the referee on the ambiguities of the term "prediction". Therefore, following the suggestions of the referee, in the revised version of our manuscript, we used the term "inference" instead of the term "prediction" and the verb "to infer" instead of "to predict", to avoid possible misunderstandings. For what concerns the substantial aspect, we have rewritten almost completely section 4.8. We have changed also the title of section 4.8, from "Possible future scenarios" to "Future scenarios and risk mitigation actions". We have illustrated the two possible future scenarios, assuming either (1) the decline of magmatic degassing and heat transfer from the magma to the overlying rocks or, alternatively, (2) the persistence of sustained magmatic degassing and heat transfer from the magma to the overlying rocks, in the next future. In section 4.8, we have also discussed the drilling of new geothermal wells, not only as a tool to monitor the bradyseismic evolution but, much more important, as an action to zeroing the inflation of the intermediate reservoir, depressurizing it, and consequently to cancel the hazard posed by hydrothermal explosions. In this way we have also answered the comment (7) posted by Referee N.2 in the annotated manuscript.

Final sentence of the Referee N.2. Again, we thank the referee for the constructive comments and suggestions, irrespective of the fate of our paper.

**1.3 Author's changes in manuscript related to the Review of Referee N.2**

**(1) First major comment.** At lines 104 to 182 of the revised manuscript we have explained how the geothermometers and geobarometers were derived and reported the equations used to compute the equilibrium temperatures and total fluid pressures of CO, $CH_4$, and $H_2S$, including their applicability and limitations.

**(2) Second major comment.** At lines 510 to 557 of the revised manuscript we have discussed the two possible future scenarios and the related risk mitigation actions, namely, the drilling of new geothermal wells as well as the exploitation of geothermal energy for electrical production and the recovery of raw materials of utmost interest such as lithium.

**2. Comments and suggestions posted by Referee N.2 in the annotated manuscript**

The comments and suggestions posted by Referee N.2 in the annotated manuscript are listed here below (IN RED COLOR; LINE NUMBERS REFER TO THE MANUSCRIPT ANNOTATED BY THE REFEREE N.2) together with author's response (in BLUE COLOR) and manuscript changes (IN BLACK COLOR; LINE NUMBERS REFER TO OUR REVISED MANUSCRIPT):

(1) Lines 1-3 (title): I suggest a slight change in the title. The use of the words 'revised' and 'prediction' does not seem appropriate. As for "revised", it's not clear which model(s) it refers to and may be misleading. As for "prediction", I have made some comments on section 4.8.

My own personal suggestion, which is by no means a binding one, might be as follows: "Time changes during the last 40 years in the Solfatara magmatic-hydrothermal system (Campi Flegrei, Italy): new conceptual model and possible scenarios".

(1) We have changed the title of our manuscript as suggested by the Referee N. 2. Moreover, we agree that it is more appropriate to use the words "new conceptual model" instead of "revised conceptual model" and to use the word "inference" instead of "prediction". Therefore, we revised the manuscript accordingly.

(1) Lines 4-6 (title): We wrote "Time changes during the last 40 years in the Solfatara magmatic-hydrothermal system (Campi Flegrei, Italy): new conceptual model and future scenarios" instead of "Revised conceptual model of the Solfatara magmatic-hydrothermal system (Campi Flegrei, Italy), time changes during the last 40 years, and prediction of future scenarios"
* * *
(2) Line 21: Please refer to the previous commentary on the title, as well as the comments on section 4.8.

(2) The word "to infer" was used instead of "to predict"

(2) Lines 25-26: We wrote "to infer the only two possible future scenarios" instead of "to predict possible future scenarios"
* * *
(3) Line 34: Please add more references that match the sentence in order to acknowledge the authors who have worked on these issues.

(3) All individual references were listed.

(3) Lines 40-41: We wrote "Cioni et al., 1984, 1989; Chiodini and Marini, 1998; Caliro et al., 2007, 2014; Chiodini, 2009; Chiodini et al., 2010, 2011, 2012, 2015, 2016, 2017a, b, 2021; Buono et al., 2023" instead of "Buono et al., 2023 and references therein"
* * *
(4) Line 51: Please add more references that match the sentence in order to acknowledge the authors who have worked on these issues.

(4) "Marini et al., 2022 and references therein" was deleted and all individual references were listed.

(4) Lines 60-61: We wrote "Cioni et al., 1984, 1989; Chiodini and Marini, 1998; Caliro et al., 2007, 2014; Chiodini, 2009; Chiodini et al., 2010, 2011, 2012, 2015, 2016, 2017a, b, 2021; Buono et al., 2023" instead of "Marini et al., 2022 and references therein"
* * *
(5) Lines 52-55: Please rephrase/restructure this sentence as it is not easy to understand

(5) The sentence was rephrased.

(5) Lines 62-66: We wrote "A fundamental tool to understand the behaviour of any hydrothermal-magmatic system is the conceptual model. A general conceptual model of volcano-hosted magmatic-hydrothermal systems was proposed by Fournier (1999), whereas Cumming (2009, 2016) provided the guidelines for elaborating the conceptual model of the hydrothermal domain of these systems" instead of "A fundamental tool to understand the behaviour of the system of interest and predict its future evolution and scenarios is the conceptual model elaborated merging available scientific data, in the framework of the general conceptual model of volcano-hosted magmatic-hydrothermal systems of Fournier (1999; see section 4.4), and following the guidelines of Cumming (2009, 2016) for the hydrothermal domain"
* * *
(6) Lines 57-58: Please be explicit about the reference. Is it Cioni et al 1984? - Please be explicit about the reference.

(6) All individual references were listed.

(6) Lines 70-71: We wrote "(Cioni et al., 1984, 1989; Chiodini and Marini, 1998; Caliro et al., 2007, 2014; Chiodini, 2009; Chiodini et al., 2010, 2011, 2012, 2015, 2016, 2017a, b, 2021; Buono et al., 2023)" instead of "by Cioni and coworkers and Chiodini and coworkers"
* * *
(7) Line 77: I suggest that the new geothermometers/barometers mentioned in the introduction shold be explicitly included in this section or in an added section. This should improve the readability of the text and make it easier for those who do not have access to the monograph in which the cited geoindicators are published. In fact, as a reviewer, I do not have access to the personal academic resources between libraries and subscriptions to the monograph by Marini et al 2022; the only way to have access to it is to purchase it myself, but even then, the time required to obtain it is in itself a limitation to providing my feedback in a reasonable timeframe (as well as beyond my duty). However, this lack of ability to know what is mentioned in the monograph by Marino et al 2022 significantly limits my ability to properly review this article. In this sense, I strongly recommend that this section be expanded or that another one be added that explicitly states the main themes of the geo-indicators used and the appropriateness of their choice in this Solfatara application case.

(7) See section 1.2 Author's response to the Review of Referee N.2, (1) First major comment.

(7) See section 1.3 Author's changes in manuscript related to the Review of Referee N.2 (1) First major comment.
* * *
(8) Lines 204-207: From the data presented in Figures 2 and 3, it is unclear how the authors reached the conclusion that a 'long residence time' occurred before July 1984 (the data shown in Figures 2-3 begin at the end of 1982). It would be beneficial for the authors to clarify whether they consider this time interval to be sufficiently long. Furthermore, providing additional information, data, or facts on this topic would enhance the clarity of the argument.

(8) The basic point is that the available data indicate that "CO- and CH$_4$ equilibrium temperatures and total fluid pressures were similar to each other, within uncertainties, until July 1984, while the difference between the CO- and CH$_4$ equilibrium temperatures and total fluid pressures two temperatures increased more and more in the following years". However, the referee is right, in that this basic point cannot be appreciated clearly from Figures 2 and 3, but it is clear if one takes into account the figures reported in Tables A1, A2, and S1. "Since the attainment of CO-CH$_4$ equilibrium requires a long time, the similarity between CO- and CH$_4$ equilibrium temperatures and total fluid pressures, in the period June 1983 - July 1984, indicates that

the residence time of fluids in the shallow reservoir was long enough to allow the attainment of CO-CH$_4$ equilibrium and that the inflow of deep gases from below was nil to negligible. In other words, the shallow reservoir behaved as a closed system or nearly so, at that time, as proposed in the conceptual model of Cioni et al. (1984)." Irrespective of the length of the considered time interval, from June 1983 to July 1984, it must be noted that our interpretation is based on the ten complete chemical analyses available for this time interval (see Table S1), which are more than sufficient in number.

(8) Lines 295-304: These sentences were re-phrased referring not only to the equilibrium temperatures of CO and CH$_4$ but also to the corresponding total fluid pressures. Moreover, we quoted not only Fig. 2, but also Fig. 3 and Tables A1, A2, and S1.
* * *
(9) Line 209: Please be explicit with the references

(9) All individual references were listed.

(9) Lines 306-307: We inserted "(Caliro et al., 2014; Chiodini, 2009; Chiodini et al., 2010, 2011, 2012, 2015, 2016, 2017a, b, 2021; Buono et al., 2023)"
* * *
(10) Line 220: References must be clear and complete

(10) All individual references were listed.

(10) Lines 316-320 were re-phrased as follows: "Irrespective of these issues, reaction (18), together with reaction (1), was taken into account in several studies of the Solfatara hydrothermal-magmatic system (Cioni et al., 1984, 1989; Chiodini and Marini, 1998; Caliro et al., 2007, 2014; Chiodini, 2009; Chiodini et al., 2010, 2011, 320 2012, 2015, 2016, 2017a, b, 2021; Buono et al., 2023)."

(11) Lines 299-300: What is this 'analogy'? Is it a meaningful comparison to cite? Can it be more detailed?

(11) We think that the comparison is meaningful and very important, in that the thick carbonate pile encountered by well Nisyros-1 behaves as aquiclude, somewhat in contrast with the common thinking that carbonate rocks are aquifers, which is actually true when they are affected by fracturing and dissolution. If not, carbonate rocks are impermeable. Further details were added to the text, as asked by the referee.

(11) Lines 405-408: The sentence was re-phrased as follows: "A thick carbonate pile (4-6.5 km depth) acting as aquiclude probably due to nil to negligible fracturing and dissolution, by analogy with the geothermal well Nisyros-1, which crossed an 830-m thick sequence of carbonate rocks behaving as aquiclude separating the two permeable zones and ending into the apophyses of a dioritic intrusion and related thermometamorphic rocks (Ambrosio et al., 2010)."
* * *
(12) Lines 367-369: The rationale is clear in the text, but Figure 7 has a very low resolution in panels b-d, which in turn does not make it easy to appreciate the comparison with the overpressure graph. Furthermore, it would be much better to be able to compare the different signals on the same time scale; for example, all 4 panels in vertical sequence with the same time axis.

Lines 383-388 (Figure 7): (See previous comment. I suggest improving the quality and composition of the graphs to better compare signals.

(12) Following the comments and suggestions of the referee, we modified Figure 8 of the revised manuscript (Figure 7 of the original manuscript) as much as possible, i.e. by narrowing panel (a), enlarging panel (b), positioning panel (b) over panel (a)

and adopting the same time axis (from 2000 to 2026), so as to make the chronogram of the overpressure at 2.7 km depth comparable with the chronogram of the ground elevation at station GNSS Rite, situated close to the center of the inner caldera. We also enlarged the panels (c) and (d), which now have an acceptable resolution, but the time axis of these two panels is different from that of (a) and (b), in that it starts on 01/01/2005 and ends on 30/11/2024; therefore, they are positioned one over the other, but in a column different from panels (a) and (b). This is the best we can do because with do not have access to the original data neither of ground elevation nor of magnitudo and depth of local earthquakes

(12) Lines 500-509: Figure 8 of the revised manuscript (Figure 7 of the original manuscript) has been redrawn following the indications of the referee as much as possible.
* * *
(13) Lines 397-407: Based on initial assumptions and excluding some potential constraints (lack of external factors), here and later in the text I see that the authors, describe certain scenarios. The first scenario is possible if the energy input is persistently sustained. It therefore requires a continuous, still active deep input of magmatic origin (with two different epilogues: a) decarbonatation b) renewed supply of magmatic fluid); otherwise, it would have to be exhausted by consumption of the thermal energy source. These scenarios are only possible if the energy input is persistently sustained. The next sentence (lines 407-412) shows the possibility of extinguishing the process by lowering the temperature, precipitating anhydrite and reducing permeability. So, which is the dominant scenario - the first one? Because it has a long-term history of overpressure? But from 2022 onwards, the trend seems to reverse.

This ambivalent possibility of the possible direction of the evolution of phenomena [also with regard to the intermediate reservoir] does not seem to be a real exercise of prediction, while on the contrary, it seems that some possible reasonable scenarios are considered. Furthermore, as stated in the introduction, the authors do not take into account additional external factors, which in turn lead to an increase in the number of possible scenarios.

In essence, it is not clear which of the possible scenarios identified prevails, and perhaps the authors should give further reasons for identifying one of them and provide supporting arguments. Alternatively, I would suggest not using the term 'prediction' in this context (see next cooments on section "5 Conclusion").

(13) See section 1.2 Author's response to the Review of Referee N.2, (2) Second major comment.

(13) See section 1.3 Author's changes in manuscript related to the Review of Referee N.2 (2) Second major comment.
* * *
(14) Lines 441-443: Once again, the use of the word "prediction" appears inappropriate here. Typically, "prediction" implies a deterministic or statistical/probabilistic analysis, neither of which is present in this article. While the proposed scenarios are plausible, they should be clearly presented as hypotheses of potential developments in order to avoid misinterpretation. It is imperative to distinguish between "postulated scenarios" and "predictions" that are anticipated to occur. This distinction must be explicitly defined due to the implications such communication has for national civil authorities and other stakeholders operating in the Phlegraean Fields area.

(14) Following the indications of the referee, we used the words "to infer" instead of "to make predictions".

(14) The sentence was re-phrased as follows: "(3) to infer the two possible future scenarios of the Solfatara magmatic-hydrothermal system in the lack of external factors, such as the occurrence of regional earthquakes and the input of fresh magma in the reservoir at 8 km depth, as well as the uprise of magma at shallower levels. We showed that the pressurization of the intermediate reservoir might trigger a hydrothermal explosion and we have proposed risk mitigation actions."
* * *
(15) Lines 458-467: I do not question the accuracy or validity of these statements, but note that there is no discussion of these issues in the main text, and here in the conclusion they seem off topic. That arguments should be considered and discussed in a dedicated section of the paper where the authors can provide the necessary data, cost analyses and other relevant information to support the claims made in the conclusion. Alternatively, it is recommended that these statements be removed from the concluding section.

(15) We agree with the referee that this discussion is out of place in the conclusive section, as these aspects were not discussed before.

(15) We have removed this part of the manuscript from the concluding section and inserted it into section 4.8, lines 546-557.
* * *
**References quoted in our answers**

Abiyudo, R., Hadi, J., Cumming, W., & Marini, L. (2016). Conceptual model assessment of vapor core geothermal system for exploration. Mt. Bromo case study. In: Proceedings of the 4th Indonesia international geothermal convention and exhibition 2016, 10–12 August 2016, Cendrawasih Hall, Jakarta Convention Center, Indonesia.

Ambrosio M, Doveri M, Fagioli MT., Marini L, Principe C, Raco B (2010) Water–rock interaction in the magmatic-hydrothermal system of Nisyros Island (Greece). Journal of Volcanology and Geothermal Research, 192(1-2):7-68

Appelo, C. A. J. (2015). Principles, caveats and improvements in databases for calculating hydrogeochemical reactions in saline waters from 0 to 200 C and 1 to 1000 atm. Applied Geochemistry, 55, 62-71.

Apuada N.A. & Sigurjonsson G.F. (2008). The geothermal potential of Biliran Island, Philippines. In: Proceedings of the 8th Asian geothermal symposium, 73–77.

Baseri, H., & Lotfollahi, M. N. (2011). Modification of Peng Robinson EOS for modelling (vapour+ liquid) equilibria with electrolyte solutions. The Journal of Chemical Thermodynamics, 43(10), 1535-1540.

Bowers, T. S., & Helgeson, H. C. (1983). Calculation of the thermodynamic and geochemical consequences of nonideal mixing in the system H2O-CO2-NaCl on phase relations in geologic systems: Equation of state for $H_2O$-$CO_2$-NaCl fluids at high pressures and temperatures. Geochimica et Cosmochimica Acta, 47(7), 1247-1275.

Buono, G., Paonita, A., Pappalardo, L., Caliro, S., Tramelli, A., & Chiodini, G. (2022). New insights into the recent magma dynamics under Campi Flegrei caldera (Italy) from petrological and geochemical evidence. Journal of Geophysical Research: Solid Earth, 127(3), e2021JB023773.

Buono, G., Caliro, S., Paonita, A., Pappalardo, L., & Chiodini, G. (2023). Discriminating carbon dioxide sources during volcanic unrest: The case of Campi Flegrei caldera (Italy). Geology, 51(4), 397-401.

Caliro S, Chiodini G, Moretti R, Avino R, Granieri D, Russo M, Fiebig J (2007) The origin of the fumaroles of La Solfatara (Campi Flegrei, south Italy). Geochim Cosmochim Acta 71(12):3040–3055

Caliro S, Chiodini G, Paonita A (2014) Geochemical evidences of magma dynamics at Campi Flegrei (Italy). Geochim Cosmochim Acta 132:1–15

Cavarretta G, Gianelli G, Scandiffio G, Tecce F (1985) Evolution of the Latera geothermal system II: metamorphic, hydrothermal mineral assemblages and fluid chemistry. J Volcanol Geoth Res 26(3–4):337– 364

Chase, M. (1998), NIST-JANAF Thermochemical Tables, 4th Edition, American Institute of Physics

Chiodini, G., & Cioni, R. (1989). Gas geobarometry for hydrothermal systems and its application to some Italian geothermal areas. Applied geochemistry, 4(5), 465-472.

Chiodini G, Marini L (1998) Hydrothermal gas equilibria: The H2O-H2-CO2-CO-CH4 system. Geochim Cosmochim Acta 62:2673–2687

Chiodini G., Cioni R., Guidi M., Marini L., Raco B., Taddeucci G. (1992). Gas geobarometry in boiling hydrothermal systems: a possible tool to evaluate the hazard of hydrothermal explosions. Acta Vulcanologica, Marinelli Volume, 2, 99-107.

Chiodini G, Avino R, Caliro S, Minopoli C (2011) Temperature and pressure gas geoindicators at the Solfatara fumaroles (Campi Flegrei). Ann Geophys 54 (2). https://doi.org/10.4401/ag-5002

Chiodini G, Vandemeulebrouck J, Caliro S, D'Auria L, De Martino P, Mangiacapra A, Petrillo Z (2015) Evidence of thermal-driven processes triggering the 2005–2014 unrest at Campi Flegrei caldera. Earth Planet Sci Lett 414:58–67

Chiodini G, Paonita A, Aiuppa A, Costa A, Caliro S, De Martino P, Acocella V, Vandemeulebrouck J (2016) Magmas near the critical degassing pressure drive volcanic unrest towards a critical state. Nat Commun 7 (1):1–9

Chiodini G, Giudicepietro F, Vandemeulebrouck J, Aiuppa A, Caliro S, De Cesare W, Tamburello G, Avino R, Orazi M, D'Auria L (2017) Fumarolic tremor and geochemical signals during a volcanic unrest. Geology 45(12):1131–1134

Chiodini G, Caliro S, Avino R, Bini G, Giudicepietro F, De Cesare W, Ricciolino P, Aiuppa A, Cardellini C, Petrillo Z, Selva J, Siniscalchi A, Tripaldi S (2021) Hydrothermal pressure-temperature control on CO2 emissions and seismicity at Campi Flegrei (Italy). J Volcanol Geotherm Res 414:107245

Cioni, R., D'Amore, F. (1984). A genetic model for the crater fumaroles of Vulcano Island (Sicily, Italy). Geothermics, 13(4), 375-384.

Cioni R, Marini L (1990) The determination of deep temperatures by means of the CO-CO2-H2-H2O geothermometer: an example using fumaroles in the Campi Flegrei, Italy. A comment. Bull Volcanol 53(1):67–69

Cioni R, Corazza E, Marini L (1984) The gas/steam ratio as indicator of heat transfer at the Solfatara fumaroles, Phlegraean Fields (Italy). Bull Volcanol 47(2):295–302

Cioni R, Corazza E, Fratta M, Guidi M, Magro G, Marini L (1989) Geochemical precursors at Solfatara Volcano, Pozzuoli (Italy). In: Latter JH (ed) Volcanic Hazards. IAVCEI Proceedings in Volcanology, vol 1. Springer, Berlin, Heidelberg, pp 384–398

Cumming W (2016) Resource conceptual models of volcano-hosted geothermal reservoirs for exploration well targeting and resource capacity assessment: Construction, pitfalls and challenges. Geotherm Resour Counc Trans 40:623–637

Cumming W (2009) Geothermal resource conceptual models using surface exploration data. In: Proceedings of the 34th workshop on geothermal reservoir engineering. Stanford University, Stanford, California, February 9–11, 2009 SGP-TR-187, p 6

D'Auria, L., Giudicepietro, F., Aquino, I., Borriello, G., Del Gaudio, C., Lo Bascio, D., Martini, M., Ricciardi, G.P., Ricciolino, P., and Ricco, C. (2011). Repeated fluid-transfer episodes as a mechanism for the recent dynamics of Campi Flegrei caldera (1989–2010). Journal of Geophysical Research: Solid Earth, 116(B4).

Del Gaudio C, Aquino I, Ricciardi GP, Ricco C, Scandone R (2010) Unrest episodes at Campi Flegrei: a reconstruction of vertical ground movements during 1905–2009. J Volcanol Geotherm Res 195(1):48–56

Driesner, T., & Heinrich, C. A. (2007). The system $H_2O$–NaCl. Part I: Correlation formulae for phase relations in temperature–pressure–composition space from 0 to 1000 °C, 0 to 5000 bar, and 0 to 1 $X_{NaCl}$. Geochimica et Cosmochimica Acta, 71(20), 4880-4901.

Duan, Z., Møller, N., Weare, J.H. (1995). Equation of state for the $NaCl-H_2O-CO_2$ system: prediction of phase equilibria and volumetric properties. Geochimica et Cosmochimica Acta, 59(14):2869–2882.

Einaudi, M. T., Hedenquist, J. W., & Inan, E. E. (2005). Sulfidation state of fluids in active and extinct hydrothermal systems: Transitions from porphyry to epithermal environments. [doi:https://doi.org/10.5382/SP.10.15]

Fiebig, J., Tassi, F., D'Alessandro, W., Vaselli, O., & Woodland, A. B. (2013). Carbon-bearing gas geothermometers for volcanic-hydrothermal systems. Chemical Geology, 351, 66-75.

Fournier RO (1985) The behavior of silica in hydrothermal solutions. Rev Econ Geol 2:45–72

Fournier RO (1999) Hydrothermal processes related to movement of fluid from plastic into brittle rock in the magmatic-epithermal environment. Econ Geol 94 (8):1193–1211

Frantz, J. D., Popp, R. K., & Hoering, T. C. (1992). The compositional limits of fluid immiscibility in the system $H_2O-NaCl-CO_2$ as determined with the use of synthetic fluid inclusions in conjunction with mass spectrometry. Chemical Geology, 98(3-4), 237-255.

Gallagher, J. S., Crovetto, R., Sengers, J. L. (1993). The thermodynamic behavior of the $CO_2-H_2O$ system from 400 to 1000 K, up to 100 MPa and 30% mole fraction of $CO_2$. Journal of Physical and Chemical Reference Data, 22(2), 431-513.

Gehrig, M. (1980). Phasengleichgewichte und pVT-daten ternärer Mischungen aus Wasser, Kohlendioxid und Natriumchlorid bis 3 kbar und 550 °C. PhD dissertation, Univ. Karlsruhe, Hochschul Verlag, Freiburg.

Giggenbach, W. F. (1987). Redox processes governing the chemistry of fumarolic gas discharges from White Island, New Zealand. Applied Geochemistry, 2(2), 143-161.

Grant M.A. and Bixley P.F. (2011) Geothermal Reservoir Engineering, Second Edition.

Harvie, C. E., Møller, N., & Weare, J. H. (1984). The prediction of mineral solubilities in natural waters: The Na-K-Mg-Ca-H-Cl-$SO_4$-OH-$HCO_3$-$CO_3$-$CO_2$-$H_2O$ system to high ionic strengths at 25 °C. Geochimica et Cosmochimica Acta, 48(4), 723-751.

Hou, S. X., Maitland, G. C., & Trusler, J. M. (2013). Phase equilibria of ($CO_2$+ $H_2O$+ NaCl) and ($CO_2$+ $H_2O$+ KCl): Measurements and modeling. The Journal of Supercritical Fluids, 78, 78-88.

Johnson, E. L. (1992). An assessment of the accuracy of isochore location techniques for $H_2O-CO_2-NaCl$ fluids at granulite facies pressure-temperature conditions. Geochimica et Cosmochimica Acta, 56(1), 295-302.

Kotelnikov, A. R., & Kotelnikova, Z. A. (1990). Experimental-study of phase state of the system $H_2O-CO_2-NaCl$ by method of synthetic fluid inclusions in quartz. Geokhimiya, 1990, 526-537.

Kunz, O., & Wagner, W. (2012). The GERG-2008 wide-range equation of state for natural gases and other mixtures: an expansion of GERG-2004. Journal of Chemical & Engineering Data, 57(11), 3032-3091.

Kwak, C., & Anderson, T. F. (1991). Application of peng-Robinson equation to high-pressure aqueous systems containing gases and sodium chloride. Korean Journal of Chemical Engineering, 8, 88-94.

Li, J., Wei, L., & Li, X. (2015). An improved cubic model for the mutual solubilities of $CO_2$–$CH_4$–$H_2S$–brine systems to high temperature, pressure and salinity. Applied Geochemistry, 54, 1-12.

Lowell RP, Van Cappellen P, Germanovich LN (1993) Silica precipitation in fractures and the evolution of permeability in hydrothermal upflow zones. Science 260(5105):192–194

Marini, L. and Chiodini, G. (1994) The role of carbon dioxide in the carbonate-evaporite geothermal systems of Tuscany and Latium (Italy), Acta Vulcanol., 5, 95-104.

Marini, L., Gambardella, B. (2005) Geochemical modeling of magmatic gas scrubbing. Annals of Geophysics, 48, 739-753.

Marini, L., & Manzella, A. (2005). Possible seismic signature of the α–β quartz transition in the lithosphere of Southern Tuscany (Italy). Journal of Volcanology and Geothermal Research, 148(1-2), 81-97.

Marini, L., Vetuschi Zuccolini, M., & Saldi, G. (2003a) The bimodal pH distribution of volcanic lake waters. Journal of Volcanology and Geothermal Research, 121, 83-98.

Marini, L., Yock Fung, A., & Sanchez, E. (2003b) Use of reaction path modeling to identify the processes governing the generation of neutral Na-Cl and acidic Na-Cl-$SO_4$ deep geothermal liquids at Miravalles geothermal system, Costa Rica. Journal of Volcanology and Geothermal Research, 128, 363-387.

Marini, L., Principe, C., & Lelli, M. (2022). The Solfatara Magmatic-Hydrothermal System. Springer: Cham, Switzerland, 375 pp.

Minucci L (1964) Rotary drilling for geothermal energy. In: Proceedings of the U.N. conference on new sources of energy. United Nations, New York, II.A.2 Harnessing geothermal energy—Eletricity production, G/66, pp 234–244

Moore DE, Lockner DA, Byerlee JD (1994) Reduction of permeability in granite at elevated temperatures. Science 265(5178):1558–1561

Moretti, R., De Natale, G., and Troise, C. (2017) A geochemical and geophysical reappraisal to the significance of the recent unrest at Campi Flegrei caldera (Southern Italy), Geochem. Geophy. Geosy., 18, 1244-1269.

Moretti R, Troise C, Sarno F, De Natale G (2018) Caldera unrest driven by CO2-induced drying of the deep hydrothermal system. Sci Rep 8(1):1–11

Moretti R, De Natale G, Troise C (2020) Hydrothermal versus magmatic: geochemical views and clues into the unrest dilemma at Campi Flegrei. In: De Vivo B, Belkin HE, Rolandi G (eds) Vesuvius, Campi Flegrei, and Campanian volcanism. Elsevier, Amsterdam, pp 371–406

Nighswander, J. A., Kalogerakis, N., & Mehrotra, A. K. (1989). Solubilities of carbon dioxide in water and 1 wt.% sodium chloride solution at pressures up to 10 MPa and temperatures from 80 to 200. degree. C. Journal of Chemical and Engineering Data, 34(3), 355-360.

Sieder, G., & Maurer, G. (2004). An extension of the Peng–Robinson equation of state for the correlation and prediction of high-pressure phase equilibrium in systems containing supercritical carbon dioxide and a salt. Fluid Phase Equilibria, 225, 85-99.

Søreide, I., & Whitson, C. H. (1992). Peng-Robinson predictions for hydrocarbons, CO2, N2, and H2 S with pure water and NaCI brine. Fluid phase equilibria, 77, 217-240.

Ramos-Candelaria, M., Sanchez, D.R., Salonga, N.D. (1995). Magmatic contributions to Philippine hydrothermal systems. In: Proceedings of the world geothermal congress, vol 2. Firenze, Italy, pp. 1337–1341.

Ranieri, L. (1952). Inversione del bradisisma di Pozzuoli. Bollettino della Società Geografica Italiana, 27-36.

Reyes, A. G., Giggenbach, W. F., Saleras, J. R., Salonga, N. D., & Vergara, M. C. (1993). Petrology and geochemistry of Alto Peak, a vapor-cored hydrothermal system, Leyte Province, Philippines. Geothermics, 22(5-6), 479-519.

Rosi M, Sbrana A (1987) Phlegrean fields. Quaderni de La Ricerca Scientifica 9(114)

Ryzhenko, B. N., & Volkov, V.P. (1971). Fugacity coefficients of some gases in a broad range of temperatures and pressures. Geochem. Inter., 8, 468-481.

Ryzhenko, B. N., & Malinin, S. D. (1971). The fugacity rule for the systems $CO_2$-$H_2O$, $CO_2$-$CH_4$, $CO_2$-$N_2$, and $CO_2$-$H_2$. Geochem. Int., 8, 562-574.

Saishu H, Okamoto A, Tsuchiya N (2014) The significance of silica precipitation on the formation of the permeable–impermeable boundary within Earth's crust. Terra Nova 26(4):253–259

Schmidt, C., & Bodnar, R. J. (2000). Synthetic fluid inclusions: XVI. PVTX properties in the system $H_2O$-NaCl-$CO_2$ at elevated temperatures, pressures, and salinities. Geochimica et Cosmochimica Acta, 64(22), 3853-3869.

Scott SW, Driesner T (2018) Permeability changes resulting from quartz precipitation and dissolution around upper crustal intrusions. Geofluids, Article ID 6957306

Shmulovich, K. I., & Graham, C. M. (1999). An experimental study of phase equilibria in the system $H_2O$-$CO_2$-NaCl at 800 °C and 9 kbar. Contributions to Mineralogy and Petrology, 136(3), 247-257.

Sourirajan, S., & Kennedy, G.C. (1962). The system $H_2O$–NaCl at elevated temperatures and pressures: American Journal of Science, 260, 115–141.

Søreide, I., & Whitson, C. H. (1992). Peng-Robinson predictions for hydrocarbons, $CO_2$, $N_2$, and $H_2S$ with pure water and NaCI brine. Fluid phase equilibria, 77, 217-240.

Tanger IV, J. C., & Pitzer, K. S. (1989). Thermodynamics of NaCl-$H_2O$: A new equation of state for the near-critical region and comparisons with other equations for adjoining regions. Geochimica et Cosmochimica Acta, 53(5), 973-987.

Taran Y.A. (1986) Gas geothermometers for hydrothermal systems: Geochemistry International, 23(7), 111–126.

Tedesco, D., & Sabroux, J. C. (1987). The determination of deep temperatures by means of the CO-CO2-H2-H2O geothermometer: an example using fumaroles in the Campi Flegrei, Italy. Bulletin of volcanology, 49, 381-387.

Weis, P., Driesner, T., & Heinrich, C. A. (2012). Porphyry-copper ore shells form at stable pressure-temperature fronts within dynamic fluid plumes. Science, 338(6114), 1613-1616.

Wells JT, Ghiorso MS (1991) Coupled fluid flow and reaction in mid-ocean ridge hydrothermal systems: the behavior of silica. Geochim Cosmochim Acta 55 (9):2467–2481

White SP, Mroczek EK (1998) Permeability changes during the evolution of a geothermal field due to the dissolution and precipitation of quartz. Transp Porous Media 33(1):81–101

Zollo, A., Maercklin, N., Vassallo, M., Dello Iacono, D., Virieux, J., & Gasparini, P. (2008). Seismic reflections reveal a massive melt layer feeding Campi Flegrei caldera. Geophysical Research Letters, 35(12).

Zuo, Z., Lu, P., Zhu, C., & Ji, X. (2024). SAFT2 equation of state for the CH$_4$–CO$_2$–H$_2$O–NaCl quaternary system with applications to CO$_2$ storage in depleted gas reservoirs. Chemical Geology, 667, 122328.

---

## Author Response (AR2)

**Public justification (visible to the public if the article is accepted and published)**:

Dear authors

In the following you will find my judgment on your revised version.

Marini and collaborators apply new geoindicators they have recently developed on the basis of a set of homogeneous and heterogeneous equilibria for the thermo-barometric interpretation of la Solfatara fluids (Campi Flegrei). The authors propose, after integration with independent geological and geophysical datasets, a new conceptual model for the Solfatara magmatic-hydrothermal system. The new conceptual model proposes that Solfatara fluids equilibrate at three distinct depths and T-P ranges, connected to the surface by a network of fractures. Their model stresses the role of the deepest reservoirs with the ongoing bradyseism. The paper concludes on the possible scenarios that can be deduced when embracing this new conceptual frame.

The research approach is rich, based on an extensive knowledge of the volcanic systems and specifically of the Campi Flegrei. The topic is obviously relevant and of major interest at local and global scale.

The review process was the opportunity of an in-depth exchange between the authors and the reviewers. As sometimes occur, major scientific debates are intertwined with passionate expression of each legitimate position. The role of the Editor is, besides appreciating the intense exchange in animating an important scientific debate, to propose a summary and to identify all possible constructive contributions that make science progress.

Here, I attempt to take some distance from the fiery debate (and I want to thank both the reviewers and the authors for providing abundant food for that) and I try to summarize a set of key points which I consider need to be carefully considered by the authors in order to further progress in the review process. My main advice is that the authors have extensively answered most of the remarks and criticism expressed by the reviewers, but that some key information has not been integrated yet in this last version of the manuscript.

My suggestion is to produce a new version enriched by the many elements shared or stressed online during the review process and relevant to help the reader to understand each step which has led the authors to their important conclusions. This new version must take into account all key points identified by the review process, summarized below.

**Authors' reply**: We agree with the Editor and we have amended the manuscript following his appreciated indications. We preferred to limit the changes to the main text to a minimum, in order to maintain the common thread that we believe will facilitate the reading of our article. However, we added distinct appendices both to provide the key information that was missing in the first version of our manuscript and to answer the remarks and criticisms of the reviewers, sometimes harsh, but still appreciated because they have allowed us to improve the quality of our paper. Here below, we provide a point-by-point reply to the comments of the Editor.

**Methods**

Both reviewers have stressed the importance of integrating a detailed discussion of the assumptions and limitations of the adopted method in a specific paragraph. Currently, some information is missing in the manuscript, other is disseminated between the Introduction and the

Methods paragraph. The paper builds on a book where the reader can find more detailed information. Nevertheless, the manuscript (and associated appendix and supplementary file) must provide all relevant information. That is mandatory in order to guide the readers, most important those having less experience in the modelling and interpretation of fluid geochemical datasets, along the whole path leading to the new conceptual model.

Many important points have been stressed in the review process and they have to be integrated in the manuscript (e.g. the assumptions underlying the new gas-geothermometers and gas-geobarometers and most important their P-T-X range of validity; the influence of the tested T-P saturation decompression paths on the calculated P-T values; the possible influence of correction for the presence of $CO_2$ and halides in the vapor phase; the influence of the choice of the %NaCl brine for the calculations and the properties of homogeneous supercritical fluids. Most of these topics have actually been addressed in the set of replies provided by the reviewers. I suggest they are integrated both in the manuscript and in a specific Appendix emphasizing the assumptions and limits of the adopted approach.

Assumptions have been well explained in the review process and now they need to be made fully explicit in the manuscript.

**Authors' reply**: all the assumptions and limitations of gas-geothermometers are reported in Appendix C, apart from the assumption on the equilibrium coexistence of an almost pure saturated vapor phase with a very small amount of brine, which is found in the main text at lines 134-144. In Appendix C, assumptions and limitations are discussed in the following sections:

C.1. General assumptions

C.2. The saturation decompression paths: assumptions and limitations

C.3. The linear P-T decompression path: assumptions and limitations

C.4. The isenthalpic decompression path: assumptions and limitations

Among the assumptions, I would also mention, for instance, the choice of vapor-static vs hydrostating pressure gradients, because considered typical of vapor-dominated dry steam systems.

**Authors' reply**: The increase in pressure from the top to the bottom of the intermediate and deep reservoirs was computed to a first approximation by using Eqn. (21), see lines 435-441. As mentioned at lines 476-479, due to the relatively high density of the fluids stored in both reservoirs, the pressure gradient turned out to be non-zero. This outcome is in contrast with the zero-pressure gradient typical of vapor-dominated geothermal systems (e.g., White et al., 1971; Truesdell and White, 1973; Grant and Bixley, 2011), where P, T conditions are similar to those of the shallow reservoir and significantly lower than those of the intermediate and deep reservoir.

**Uncertainty assessment**
The authors state (and that needs to be stressed in the manuscript) that absolute values of time series of T and P rely on selected/tested decompression paths, but that relative evolution is similar.

Basically, that implies that the conclusions of the manuscript do not rely on absolute values. More globally, the authors argue that uncertainty associated with the sampling+analytical procedure and fluid variability over a given time interval is larger than that related to the thermo-barometric calculations.

**Authors' reply**: In the main text, at lines 115 to 134, we clarified how the different expansion paths were considered in the calibration of the CO-, CH$_4$-, and H$_2$S-geothermometers and how many times series of equilibrium temperature and total fluid pressure were obtained for each geothermometer and geobarometer. These T, P time series are 4 for the CO-geoindicators (which gives similar results), 3 for the CH$_4$-geoindicators (which gives somewhat different results), and only 1 for the H$_2$S-geoindicators. This means that only the outcomes of the 3 CH$_4$-geoindicators have to be compared. We did this comparison in Appendix B, where we explained also why we chose the saturation (21 wt% NaCl) decompression path among the three time-series given by the CH$_4$-geoindicators.

However, the review process was the opportunity to discuss the applicability of the EOS (Peng-Robinson) chosen by the authors at temperatures >500°C and in occurrence of variable proportions of NaCl in the vapour phase together with the influence of CO2 in the saturated vapour on the NaCl content of the liquid. When discussing the uncertainties, the role of the assumptions permitting to adopt the approximation of a CO2 bearing vapor phase in equilibrium with a vapor-saturated H2O-NaCl liquid and most important the possible role at high T of the mutual solubilities on the definition of saturation pressure must be included. Both the authors and the reviewer 1 agree on the upper limits (600°C and 1000°C for the 20% and 33% NaCl brines, respectively) of applicability of the equations. The most recent datasets approach or overpass these limits. Do the authors consider that their model can still be applied to the most recent datasets?
Similarly, a general agreement exists on the need of new experiments in the ternary H2O-NaCl-CO2 system. This can be discussed and linked to the assumptions made by the authors.

**Authors' reply**: These points are discussed in appendices A and C. In appendix A, we stressed the importance of considering the deviations from the ideal gas behavior in geothermometric calculations and we recalled the contrasting behavior, with increasing P and T, of the fugacity coefficient of H$_2$O and non-polar gases (CO$_2$, CH$_4$, CO, H$_2$S, and H$_2$) and the related implication on gas geothermometers. We also discussed the effects of mutual solubilities on fugacity coefficients and we explained why these effects were disregarded by Marini et al. (2022).

In Appendix C, as already recalled above, we discussed uncertainties, assumptions, limitations and applicability of the geoindicators calibrated for the different expansion paths. In particular, we focused on the two saturation paths, because they were finally adopted in our paper. Thus, we recalled the characteristics of the unary system H$_2$O, of the H$_2$O-CO$_2$ binary system, and of the H$_2$O-NaCl binary system, explaining why we adopted the last one to link P and T. We agree that it would have been much better to use the H$_2$O-NaCl-CO$_2$ ternary system, but this is not feasible at present, because of the lack of experimental data which complicates, not to say prevents, the derivation of

reliable EOS for the $H_2O$-NaCl-$CO_2$ ternary system. We agree that new experimental data for $H_2O$-NaCl-$CO_2$ ternary system are urgently needed.

Discussion section
Again, here the assumptions or available constraints concerning the time spent by the fluids in each reservoir and/or along the flow path, the extent and depth of the re-equilibration process etc need to be made explicit for the reader. The integration of geochemical and geophysical/geological datasets is at the core of the new conceptual model.

**Authors' reply**: As discussed in Appendix E, we tried to estimate the residence time spent by the fluids in each reservoir, taking our conceptual model as reference, assuming steady-state conditions, and specifying the total volume (rocks + fluids), the effective porosity, and the T, P conditions of each reservoir, as well as the flow of fluids through the system (from the $CO_2$ flow measured at the surface and the $X_{H_2O}/X_{CO_2}$ ratio of fumarolic fluids). Unfortunately, due to the poor knowledge of the effective porosity and its possible changes with time due to different factors (e.g., ground uplift, seismicity, mineral dissolution/precipitation), it is impossible to made reliable evaluations of the residence time of fluids, even for the shallow reservoir.

For instance, Reviewer 1 proposes alternative explanation to the modelled increase in T and P, arguing that this might result from a change in the hydraulic regime or degassing rate, allowing the deep chemical signal of fluids to be better preserved.
The discussion can also remind how much room is left for alternative explanations.

**Authors' reply**: This alternative explanation proposed by reviewer 1 was discussed by Marini et al. (2022) who noted that "*the increasing SS4* [$CH_4$] *equilibrium temperature with time allows for two distinct implications, either (1) fumarolic fluids came from progressively deeper zones of the Solfatara magmatic-hydrothermal system, characterized by gradually higher fluid pressures and temperatures, or (2) fumarolic fluids came from the same deep permeable zone which experienced a progressive increase in fluid pressure and temperature with time.*" However, implication (2) explains the pressurization of the intermediate aquifer and the consequent ground uplift, on which there is a consensus in the scientific literature, whereas implication (1) does not. This is why we adopted the second implication in our paper. We added these considerations at lines 218-223.

Reviewer 1 suggests that calculated overpressures in the deepest reservoir are unrealistic and that ductile rocks cannot sustain that. This point needs to be addressed in the discussion section.

**Authors' reply**: This point was briefly recalled in the discussion section of the manuscript at lines 513-514 and was thoroughly discussed in Appendix F.

A link must thus be proposed in order to make explicit the influence of selected decompression paths on calculated pressures and the implications of estimated overpressures.

**Authors' reply**: This point is discussed in Appendix B. In particular, since the differences between distinct $CH_4$ equilibrium temperatures and pressures are <25°C and <200 bar in 2013-2021, the computed $CH_4$ equilibrium temperatures and pressures are almost independent of the considered

decompression path in the last years and the error in the estimated overpressure (Fig. 8a) is similar to or even less than short-term fluctuations.

Some important conclusions are stressed in the online replies, and can be better emphasized in the manuscript. For instance, the fact that CO-CO2 equilibrium P-T form the current basis of monitoring activity and that they rely to the shallowest reservoir and that these pressures are considered too low to explain the ongoing bradyseism is an important conclusion and can be better emphasized.

**Authors' reply**: The fact that CO-CO2 equilibrium P-T, forming the current basis of monitoring activity, refer to the shallow reservoir and that these pressures are too low to explain the ongoing bradyseism was recalled in the Conclusions.

In conclusion, the last (modified) version of the manuscript integrates only part of the answers provided to the reviewers and misses some key points (e.g. the important overpressure in the deepest reservoir; the applicability of the model for high T domains etc). If you agree to produce a new version integrating i) an appendix to summarize the main assumptions and limitations of the model and ii) the missing points in the discussion part, I will be glad to consider the manuscript for possible publication in SE.
Best regards
Massimo Coltorti

**Authors' reply**: We hope we answered all the requests by the Editor and that the manuscript is now suitable for publication in SE.

Best regards,

Matteo Lelli